# ⚖️ SCALE: Scaling up the Complexity for Advanced Language Model Evaluation

## Abstract

Recent strides in Large Language Models (LLMs) have saturated many NLP benchmarks (even professional domain-specific ones), emphasizing the need for more challenging ones to properly assess LLM capabilities. In this work, we introduce a novel NLP benchmark that poses challenges to current LLMs across four key dimensions: processing *long documents* (up to 50K tokens), using *domain-specific knowledge* (embodied in legal texts), *multilingual* understanding (covering five languages), and *multitasking* (comprising legal document-to-document Information Retrieval, Court View Generation, Leading Decision Summarization, Citation Extraction, and eight challenging Text Classification tasks). Our benchmark contains diverse legal NLP datasets from the Swiss legal system, allowing for a comprehensive study of the underlying non-English, inherently multilingual, federal legal system. Despite recent advances, efficient processing of long documents for intense review/analysis tasks remains an open challenge for LLMs. In addition, comprehensive, domain-specific benchmarks requiring high expertise to develop are rare, as are multilingual benchmarks. This scarcity underscores our contribution's value, considering that most public models are trained predominantly on English corpora, while other languages remain understudied, particularly for practical domain-specific NLP tasks. Our benchmark allows for testing and advancing the state-of-the-art LLMs. As part of our study, we evaluate several pre-trained multilingual language models on our benchmark to establish strong baselines as a point of reference. Despite the large size of our datasets (tens to hundreds of thousands of examples), existing publicly available models struggle with most tasks, even after extensive in-domain pre-training. We publish all resources (benchmark suite, pre-trained models, code) under a fully permissive open CC BY-SA license.

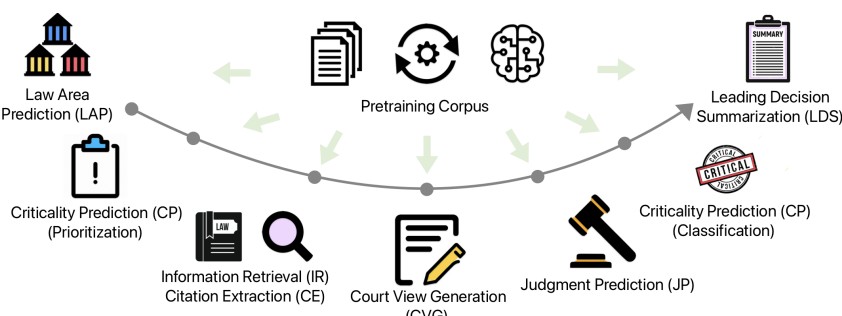

Figure 1: Sequence of tasks for support in the judicial system.

## 1 Introduction

The history of legal Natural Language Processing (NLP) is extensive (Ashley, 2017), with remarkable progress recently (Katz et al., 2023a). Notably, the introduction of datasets containing legal data from various jurisdictions worldwide (Paul et al., 2021), (Chalkidis et al., 2019b), as well as the development of more domain-specific tasks and benchmarks (Hendrycks et al., 2021b; Li & Zhang, 2021a; Semo et al., 2022; Brugger et al., 2023; Hwang et al., 2022; Niklaus et al., 2023a; Thakur et al., 2021; Chen et al., 2022; Guha et al., 2022) have significantly contributed to the progress in the field. General benchmarks such as SuperGLUE (Wang et al., 2019) are saturated and ineffective at

differentiating Large Language Models (LLMs). Hence, larger, challenging benchmarks are urgently needed, especially in the domain-specific context. In the context of Switzerland, the availability of only one dataset for evaluating LLMs hampers the assessment of their performance and effectiveness within the country's diverse linguistic and legal landscape (Niklaus et al., 2021; 2022). In this paper, we introduce seven related datasets covering a range of tasks and spanning across five languages within the same overarching jurisdiction. These datasets are derived from 26 cantons and the Swiss Federal Supreme Court (SFCS), each with distinct legal frameworks, in the uniquely multilingual and multi-jurisdictional context of Switzerland. The country's multiple official languages and a wealth of data for its size, position Switzerland as an exemplary testbed for assessing LLMs in a multilingual and multi-jurisdictional environment. Our assessment concentrates on three classification tasks – Criticality Prediction (CP), Judgment Prediction (JP), and Law Area Prediction (LAP) – an Information Retrieval (IR) task and two generative tasks – Court View Generation (CVG) and Leading Decision Summarization (LDS). To facilitate a comprehensive analysis and provide baselines for future research, we evaluate an array of models on our datasets similar to Hwang et al. (2022) or Niklaus et al. (2023a). Furthermore, we have pretrained our own Swiss legal models, Legal Swiss RoBERTa$_{Base/Large}$ and Legal Swiss Longformer$_{Base}$. Our tasks challenge current models significantly, with the best performing model only achieving an aggregated Macro F1 score of 48.4. ChatGPT was not able to solve the Text Classification (TC) tasks well, considerably lagging behind fine-tuned models. The results for CP, IR and CVG are particularly underwhelming, seeming rather arbitrary. We invite the research community to develop new methods to tackle these hard tasks. All data employed in this study is in the public domain (see *https://entscheidsuche.ch/dataUsage* and *https://www.fedlex.admin.ch/en/legal-information* and is available on the HuggingFace Hub under a CC BY-SA license (URL available upon acceptance).

This paper makes three contributions. First, we present seven public multilingual datasets containing Swiss legal documents. Second, we release two large, in-domain pretraining datasets, and pretrain three new models - Legal-Swiss-RoBERTa$_{Base/Large}$ and Legal-Swiss-LongFormer$_{base}$. Third, we evaluate multilingual baselines on our datasets and compare them to our models. Although in-domain pretraining improves performance, significant room for improvement remains in most tasks.

## 2 RELATED WORK

We briefly discuss prior work on benchmarks for long documents, domain specificity, multilinguality, and multitasking. Additional task-specific related work is presented in Appendix E.

**Long Documents** SCROLLS consists of summarization, Question Answering (QA), and Natural Language Inference (NLI) tasks with example inputs typically in the thousands of English words (Shaham et al., 2022). MULD is a set of six tasks (twice QA, style change detection, classification, summarization, and translation) where each input is at least 10K tokens, with some up to almost 500K tokens (Hudson & Moubayed, 2022).

**Domain Specificity** The BLUE benchmark (Peng et al., 2019a) contains five tasks over ten datasets for biomedical and clinical texts. CBLUE (Zhang et al., 2022) is a Chinese biomedical benchmark with eight NLU tasks including Named Entity Recognition (NER), information extraction, diagnosis normalization, TC, QA, intent classification, and semantic similarity. LEXGLUE covers six predictive tasks over five datasets made of English documents from the US, EU, and Council of Europe (Chalkidis et al., 2022). LEXTREME is a multi-lingual and multi-task benchmark for the legal domain (Niklaus et al., 2023a). LegalBench (Guha et al., 2022) covers zero-shot and few-shot Language Model (LM) evaluation for diverse realistic legal tasks in English. LBOX OPEN (Hwang et al., 2022) consists of five legal tasks from South Korea.

**Multilinguality** XTREME (Hu et al., 2020) evaluates cross-lingual generalization across six tasks and ten datasets, covering 40 languages. Some datasets were cross-lingual, others were extended via professional and automatic translations. XTREME-UP expands XTREME, focusing on few-shot evaluation of multilingual models for user-centric tasks (Ruder et al., 2023). It includes 88 underrepresented languages like Swahili, Burmese, and Telugu.

**Multitasking** GLUE (Wang et al., 2018), an early benchmark of sentence NLU tasks evaluating general-purpose neural LMs, quickly became obsolete due to advanced models like BERT (Devlin et al., 2019). Its updated version, SUPERGLUE (Wang et al., 2019), introduced new tasks challenging for machines yet solvable by humans. MMLU features only zero-shot and few-shot learning tasks (Hendrycks et al., 2021a), containing 16K multiple-choice questions across 57 subtasks, in subjects such as humanities, social and hard sciences. CLUE (Xu et al., 2020) is the first Chinese language multitask benchmark that includes single sentence classification, sentence pair classification, and machine reading comprehension. BIG-Bench (Srivastava et al., 2022) consists of 204 language tasks created by 450 authors from 132 institutions. The tasks cover topics such as linguistics, childhood development, math, common-sense reasoning, biology, physics, social bias, software development. HELM (Liang et al., 2022) is a multi-metric benchmark covering seven metrics and seven targeted evaluations and involves 42 test scenarios with a large-scale evaluation of 30 LMs.

## 3 BACKGROUND ON THE SWISS LEGAL SYSTEM

Switzerland comprises 26 cantons, each with unique jurisdiction and court organization. The Swiss Federal Supreme Court (SFCS) is Switzerland's highest legal authority and final arbiter for federal criminal, administrative, patent, and cantonal courts. Its decisions make gaps in legislation explicit, and shape the development of the law and its adaptation to changing circumstances. The SFCS has seven divisions, specializing in public, penal, and civil law (Bundesgericht, 2019). Some SFCS cases are designated as Leading Decisions and are published separately, significantly influencing future jurisprudence. Cantonal court proceedings begin at the lowest instance and may be appealed higher, with appeal stages varying by canton and legal area. Swiss court decisions typically consist of four major sections: 1) the *rubrum* (introduction) contains the date and chamber, mentions the involved judge(s) and parties and finally states the topic of the decision. 2) the *facts* describe what happened in the case and form the basis for the considerations of the court. The higher the level of appeal, the more general and summarized the facts. 3) the *considerations* reflect the formal legal reasoning, citing laws and other influential rulings, and forming the basis for the final ruling. 4) the *rulings*, the final section, are an enumeration of the binding decisions made by the court. This section is normally rather short and summarizes the considerations.

## 4 SCALE: THE DATASETS

Table 1 presents our eleven datasets, collected from 26 cantons (in addition to federal decisions), 184 courts, 456 chambers, four main law areas, and five languages as seen in Table 18. Document availability varies significantly across cantons and courts.[1] Most courts are monolingual, but there are cantons where multiple languages are used in documents. Besides decision-based datasets, we also provide a collection of approx. 35K laws from cantonal and federal jurisdictions in Switzerland.

While most SFCS cases are written in German, French is more common for cantonal cases. We partitioned downstream datasets into training (until 2015), validation (2016-2017), and test (2018-2022) sets. We opted for a relatively large test split because LLMs seem to need relatively little training data Brown et al. (2020). A large test set allows longitudinal studies, including COVID-19 pandemic years. This large temporal gap between the newest training (2015) and test (2022) samples might contribute to model difficulties (see section 6). Source data undergo rigorous

Table 1: Overview over all datasets and their multilingualism: Abbreviations: **Cant**onal, **Fed**eral, **Fac**ts, **Cons**iderations Column Fac and Cons report the mean number of tokens. Sections Facts and Considerations are not available for Ruling Summarization, Legislation and Rulings due to different format, thus mean number of tokens for the full text is reported and marked with *.

| Name | Level | Total | DE | FR | IT | RM | EN | Fac | Cons |
|---|---|---|---|---|---|---|---|---|---|
| Rulings | Cant + Fed | 638K | 320K | 247K | 71K | - | 180 | - | *7K |
| Leading Decisions | Fed | 21K | 14K | 6K | 1K | - | - | 689 | 3K |
| Legislation | Cant + Fed | 36K | 18K | 11K | 6K | 534 | 207 | - | *7K |
| Doc2Doc IR | Fed | 141K | 87K | 46K | 8K | - | - | 847 | 3K |
| Citation Extraction | Fed | 131K | 85K | 38K | 8K | - | - | - | 204 |
| Criticality | Fed | 139K | 85K | 45K | 8K | - | - | 828 | 3K |
| Law Area | Cant + Fed | 329K | 127K | 156K | 46K | - | - | 2K | 4K |
| Judgment Prediction | Cant + Fed | 329K | 160K | 128K | 41K | - | - | 2K | 4K |
| Court View | Cant + Fed | 404K | 197K | 163K | 44K | - | - | 2K | 5K |
| Court View Origin | Fed | 270 | 49 | 221 | - | - | - | 1K | 6K |
| Leading Decision Summarization | Fed | 18K | 12K | 5K | 835 | - | - | - | *3K |

---

[1] The SFCS is the only court where we have complete data, since all decisions since 2007 have been published.

curation by established Swiss institutions such as courts and administrative bodies, including manual anonymization at an approximate cost of 45 minutes per case (Niklaus et al., 2021).

## 4.1 DATABASE CREATION PIPELINE

Every day, new cases are published on Entscheidsuche.ch, allowing for daily document retrieval (see Figure 2). **(1)** We scrape all files from Entscheidsuche.ch, including each court's folder metadata. Only new case documents are sent through the pipeline. **(2)** We used BeautifulSoup / tika-python library to extract text from HTML / PDF. **(3)** Language is identified using fastText Grave et al. (2018) for subsequent tasks. **(4)** A cleaner removes irregular patterns or redundant text to avoid extraction errors. **(5)** Cases are segmented into header, facts, considerations, rulings, and footer via regex patterns. **(6)** To extract the judgement outcome, a word set is defined for each outcome. As these indicators are not context-exclusive, considering only the

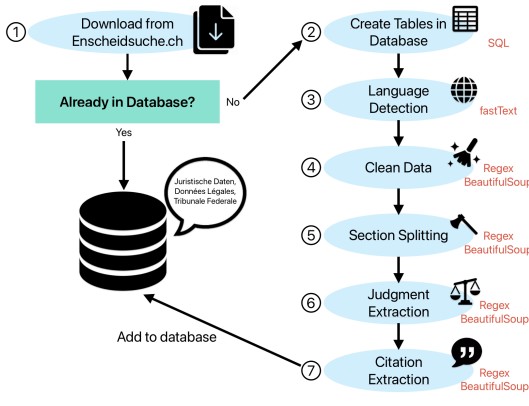

Figure 2: Database Creation Pipeline

ruling section is crucial to avoid false positives. Therefore, accurate judgment outcome extraction relies on precise section splitting. **(7)** Leading Decision (LD) and law citations are obtained through Regex (cantonal) or BeautifulSoup (federal). The SFCS labels citations with HTML tags, ensuring a high quality of citations for federal cases.

Providing an objective metric for quality is hard and expensive to obtain. Multiple people repeated quality checks over multiple months during this process to ensure the highest quality. The parsers and regexes were double-checked by senior people before integration. We wrote a series of tests to make sure that the pipeline is robust to changes (*test_utils.py*). Finally, we wrote code to easily inspect samples at various stages of the pipeline to ensure quality (*debug_utils.py*).

## 4.2 PRETRAINING

**Legislation**   The Swiss Legislation dataset comprises 35.7K legislative texts (182M tokens) in five languages: German, French, Italian, Romansh, and English (see Table 7). Table 9 details its coverage of federal, cantonal, and inter-cantonal legislation on a broad array of legal topics including public health, education, civil rights, societal matters, energy, environment, infrastructure, and visa regulations. It also includes instances of the same legislation texts across different languages, useful for enhancing the multilingual capabilities of legal LMs.

**Rulings**   The Swiss Rulings dataset is a comprehensive collection of Swiss court rulings designed for pretraining purposes. It consists of 638K cases (3.3B tokens) distributed across three languages: German (319K), French (247K) and Italian (71K). Spanning several decades and covering multiple areas of law, this dataset provides an extensive representation of Swiss law practice.

## 4.3 TEXT CLASSIFICATION

We use eight configurations from the LAP, CP, and JP datasets. Law Area and Judgment Prediction datasets include federal and cantonal cases, while the Criticality dataset focuses on SFCS cases. All tasks presented in this section involve Single Label Text Classification (SLTC), which required either extracting or defining labels. For each of the tasks, we consider the facts and the considerations as input. The facts represent the most similar available proxy to the complaints, useful for predictive tasks. The considerations as input make the tasks considerably easier, since they include the legal reasoning. These tasks can be used as post-hoc analyses for verification (e.g., in judgment prediction whether the made judgment is congruent with the given reasoning).

**Law Area Prediction**   The **Law Area** label was established by associating a law-area to each chamber where a case was adjudicated. Using metadata from Entscheidsuche.ch, a lawyer helped

Table 2: Task Configurations. Label names are *Critical* (C), *Non-critical* (NC), *Critical-1* (C1) to *Critical-4* (C4), *Approval* (A), *Dismissal* (D). For Law-Sub-Area we reported only the two most common labels *Substantive Criminal* (SC), *Criminal Procedure* (CP), and the two least common *Intellectual Property* (IP), *Other Fiscal* (OF). Abbreviations: **Val**idation, **Con**siderations, **Fac**ts

| Task Name | Train | Labels Train | | | | Val | Labels Val | | | | Test | Labels Test | | | |
|---|---|---|---|---|---|---|---|---|---|---|---|---|---|---|---|
| | | **C** | **NC** | | | | **C** | **NC** | | | | **C** | **NC** | | |
| LD-Fac | **75K** | 3K | 72K | - | - | **12K** | 580 | 13K | - | - | **26K** | 950 | 25K | - | - |
| LD-Con | **91K** | 3K | 85K | - | - | **15K** | 580 | 13K | - | - | **32K** | 948 | 29K | - | - |
| | | **C-1** | **C-2** | **C-3** | **C-4** | | **C-1** | **C-2** | **C-3** | **C-4** | | **C-1** | **C-2** | **C-3** | **C-4** |
| Citation-Fac | **2.5K** | 782 | 626 | 585 | 513 | **563** | 186 | 152 | 131 | 94 | **725** | 137 | 177 | 224 | 187 |
| Citation-Con | **2.5K** | 779 | 624 | 586 | 520 | **563** | 186 | 154 | 131 | 92 | **723** | 137 | 177 | 224 | 185 |
| | | **D** | **A** | | | | **D** | **A** | | | | **D** | **A** | | |
| Judgment-Fac | **197K** | 135K | 62K | - | - | **37K** | 27K | 11K | - | - | **94K** | 67K | 27K | - | - |
| Judgment-Con | **188K** | 130K | 59K | - | - | **37K** | 26K | 11K | - | - | **92K** | 66K | 26K | - | - |
| | | **SC** | **CP** | **IP** | **OF** | | **SC** | **CP** | **IP** | **OF** | | **SC** | **CP** | **IP** | **OF** |
| Law-Sub-Area-Fac | **10K** | 3K | 3K | 6 | 2 | **9K** | 2K | 1K | 11 | 1 | **3K** | 1K | 509 | 5 | 1 |
| Law-Sub-Area-Con | **8K** | 2K | 1K | 6 | 2 | **7K** | 2K | 750 | 11 | 1 | **3K** | 885 | 401 | 3 | 1 |

define the law areas for each chamber, resulting in chambers being classified into one of four main law-area categories (civil, public, criminal and social law) and 12 sub-law-areas. Due to many chambers operating in various law areas, it was not always feasible to assign a single law area label to each chamber. Particularly for the more detailed sub-law areas, where several chambers could not be uniquely linked, this resulted in a small subset of cases with the subset label. Initial results on the full dataset including the four main law areas showed that current models achieve near perfect accuracy, which is why we only consider the smaller filtered dataset of sub areas for this benchmark.

**Judgment Prediction**  We generated the **Judgment** label by using regex patterns to extract the judgment outcome, assigning it a binary label: approval or dismissal, akin to Niklaus et al. (2021). Partially approved or dismissed judgments were labeled as either approval or dismissal.

**Criticality Prediction**  We quantified **Criticality** in two ways: First, the **LD-Label** is binary: *critical* and *non-critical*. SFCS cases are labeled as *critical* if additionally published as Leading Decision (see 3). We extracted SFCS file names from LD headers using regex. Cases absent in LD headers are labeled *non-critical* (missing *critical* cases exist as we have all LD but not all SFCS cases). Second, to create a more precise adaptation of the LD-Label, we developed the **Citation-Label**, which involved counting all citations of LD in all SFCS cases. The LD frequency was weighted based on recency, with older citations receiving a smaller weight: $score = count * \frac{year - 2002 + 1}{2023 - 2002 + 1}$. This resulted in a ranking of LDs, which were then divided into four categories of criticality *critical-1* to *critical-4*. We used the 25, 50 and 75% quartiles as separation for our four classes.

## 4.4 TEXT GENERATION

**Court View Generation**  Clerks and judges spend about 50% of their time on writing considerations in penal law and an estimated 85% in other legal areas (Niklaus et al., 2021). Crafting considerations is central to a judge's role, requiring deep knowledge of legislation, caselaw, legal analysis, and advanced reasoning to synthesize this information. The task's complexity, particularly in the Supreme Court, manifests in the average appointment age of judges being 50.[2], highlighting the length and difficulty of their professional journey. Given the time and expertise demands, the need for the CVG task arises, targeting the generation of case considerations from facts. Generating court views is challenging due to the length and complexity of both facts (input) and considerations (output). Current models, limited in handling long context, frequently fail to process this extensive input. This limitation, combined with the input's complexity, highlights the current models' deficiencies. To benchmark models in long-context complex reasoning, we introduce a novel CVG dataset with over 400K cases spanning various legal scenarios. Averaging 1522 tokens for facts and 4673 for considerations, the dataset serves as a challenging benchmark for generating coherent, accurate case considerations. Additionally, we offer a court view origin dataset with federal rulings, augmented by lower court data, including facts and considerations from both levels. This adds a multilevel judicial perspective, enhancing understanding of case progression and increasing the CVG challenge.

---

[2]*Mit 28 Jahren Mitglied des Bundesgerichts*, SonntagsZeitung, December 19, 2019, p. 24

**Leading Decision Summarization** LD are crucial in the Swiss legal system, often cited to clarify legislative gaps. Access to their summaries simplifies searching and understanding key concepts, the most important citations, and main themes. In the LDS dataset, we include 18K LDs with their summaries, written by SFCS clerks and judges.

## 4.5 INFORMATION RETRIEVAL

Our IR task is organized into queries, qrels, and corpus (see Figure 3). The corpus includes all Swiss legislation and leading decisions; queries come from SFCS cases in German, French, and Italian. The mean token count for our queries significantly exceeds that in other IR benchmarks, due to the use of entire documents as queries (see Table 1). The goal is to find laws and decisions cited in a given case. We use the facts as a proxy for a lawyer-drafted appeal. Ground truth is based on citations from the considerations. We extract relevant cited laws and decisions from the Swiss legislation and leading decisions datasets, respectively. Document lengths resemble tasks like EU2UK (Chalkidis et al., 2021a). Laws in all three official languages yield cross-lingual query-corpus pairs, logged as qrels. Long documents and cross-lingual factors challenge retrieval models. We have 10K documents, 101K queries, and 2K qrels, averaging 19 relevant documents per query.

## 4.6 CITATION EXTRACTION

The SFCS annotates citations with special HTML tags, which we used to create a token classification dataset for Citation Extraction (CE). Solving the task with regexes is complicated due to extensive citation rules, but the transformer-based model MiniLM (Wang et al., 2020b) achieves over 95 macro F1. For brevity, we omit experiments on this dataset, but release the dataset and the trained model as a resource to the community under a CC BY-SA license (*URLs available upon acceptance*).

## 4.7 THE BIG PICTURE

The pretraining corpus and our seven datasets JP, LAP, CP, IR, CVG, LDS, and CE form a unified framework that resembles an artificial judicial system (see Figure 1). Pre-training serves as the foundation, equipping models with the ability to specialize in the respective tasks and thereby enhancing their performance. The remaining tasks, all interconnected, focus on the output of the judicial system. Superior performance in one task can bolster the effectiveness of the others. LAP facilitates routing decisions to the correct chambers inside a court. CP enables courts allocating resources and setting priorities. IR identifies the relevant documents for a case, facilitating the JP and CVG tasks, which predicts the case's outcome and synthesizes a coherent text to explain the decision's rationale. CE automatically extracts citations to enrich the final decision before publication. LDS condenses the reasoning into a short summary. Together, these tasks model (albeit still primitively) the flow of the judicial system end-to-end, the first of this kind, to the best of our knowledge.

# 5 EXPERIMENTS

In this section, we detail the pretraining of our legal models and the experimental setup for each task. Besides BLOOM Scao et al. (2022) and mT5 Xue et al. (2021), few open multilingual LLMs (> 500M parameters) exist, with most recent work pretraining on English only. Only 4.5% of LLaMA-2 Touvron et al. (2023) uses only 4.5% non-English data, while XLM-R Conneau et al. (2020b) includes 87% non-English text. Table 7 provides an overview of models evaluated.

## 5.1 PRETRAINING LEGAL MODELS

We release two multi-lingual legal PLMs, Legal-Swiss-RoBERTa and Legal-Swiss-LF$_{Base}$ (see Table 7 for details), trained on Swiss rulings, legislation, and EUR-LEX data (Niklaus et al., 2023b). We adhered to best-practices in LM development detailed in Appendix F. We will release all models on the HuggingFace Hub under a CC BY-SA license including intermediate checkpoints (every 50K/10K training steps for RoBERTa/Longformer models) upon acceptance. Limited resources prevented us from pretraining a large generative model, so we leave this to future work.

## 5.2 TEXT CLASSIFICATION

For our TC tasks, namely LAP, JP, and CP, we adopted the LEXTREME benchmark setup (Niklaus et al., 2023a), namely hierarchical aggregation of macro-averaged F1 scores using harmonic mean for fairness (the harmonic mean is biased more towards lower scores than the geometric or arithmetic mean). We averaged in order over random seeds, languages (de, fr, it), configurations (e.g., JP-F and JP-C), and datasets (LAP, JP, and CP). This setup punishes models with outlier low scores in certain languages, or configurations, thus promoting fairer models. We fine-tuned all models below 2B parameters per task on our training datasets with early stopping on the validation dataset. We evaluated closed models 0-shot as per the setup of Chalkidis (2023), using one instruction and example as input. Samples were randomly selected from the validation set to prevent test set leakage for future evaluations. For each sample, we checked whether it exceeded the model's maximum token limit of 4096 and truncated if necessary. To manage costs, we limited the validation set to 1000 samples. Our experiments focused solely on zero-shot classification due to the long input lengths. We show an overview of the prompts used in Appendix J. We used the ChatCompletion API for GPT-3.5 (gpt3.5-turbo as of June 7, 2023), the Anthropic Claude API for Claude-2, the Vertex AI API for PaLM-2 (text-bison@001), and ran LLaMA-2 locally with 4-bit quantization.

## 5.3 TEXT GENERATION

We evaluated using **BERT**Score (Zhang et al., 2020b), **BLEU** (Papineni et al., 2002), **MET**EOR (Banerjee & Lavie, 2005), and **R**OUGE (Lin, 2004). Each individual metric has inherent weaknesses (Zhang et al., 2020b), so it is necessary to employ multiple metrics for a more comprehensive assessment. We suggest future work to evaluate predictions with trained lawyers.

**Court View Generation**  Due to lengthy input (avg. 1522 tokens) and output (avg. 4673 tokens) for CVG, we truncated input facts to 2048 tokens and output considerations to 512 tokens. In 90% of cases, complete facts were retained. This truncation was driven by resource constraints and the task's complexity, which remained challenging with only 512 output tokens. Owing to test data volume and compute limits, the evaluation was limited to a subset of 1K instances. For the origin dataset, input was evenly divided between origin facts and considerations.

**Leading Decision Summarization**  In our LDS experiments, we faced large input text (avg. 3081 tokens) but shorter output text (avg. 168 tokens). To manage this, we truncated input to 4096 tokens and output to 256 tokens, preserving full output in over 80% of cases.

## 5.4 INFORMATION RETRIEVAL

Finding relevant legal references for SFCS cases is challenging due to legal language complexity, multilinguality, and long documents. We explore multilingual Doc2Doc IR in the legal domain using our new dataset, which includes SFCS cases with unique identifiers for law citations and LD. Performance is expected to decline as document count increases. We conducted an ablation study to assess minor dataset adjustments on performance (see Appendix G). We employ BM25 for its scalability to long documents but note its limitations in contextual processing and multilingual handling (Robertson & Zaragoza, 2009). Neural methods like Sentence-Bert (SBERT) show promise but degrade on long texts due to truncation-induced context loss. We also investigate training with hard negatives, using the distiluse-base-multilingual-cased-v1 SBERT model (Zhan et al., 2021). This 135M parameter model employs DistilmBERT as the student and Multilingual Universal Sentence Encoder (mUSE) as the teacher, trained on 15 languages. It has a 128 token maximum sequence length and outputs 512-dimensional embeddings via mean pooling, suitable for cosine similarity scoring. We excluded cross-encoder models (Wang et al., 2020a), due to their high computational cost.[3] In addition to Normalized Discounted Cumulative Gain (NDCG) (Wang et al., 2013) we use Capped Recall@k (Thakur et al., 2021).[4]

---

[3] BEIR datasets (Thakur et al., 2021) feature query lengths of 3-192 words and document lengths of 11-635 words. Our dataset surpasses these by 4-282x for queries (847 words on average) and approx. 8-500x for documents (approx. 4K/7K words on average for rulings/legislation documents), further increasing computational costs.

[4] The Capped Recall@k is computed as the proportion of relevant documents for a specific query, retrieved from the top k scored list of documents generated by the model. This is a good representation of model success in our specific task, as each query has multiple relevant documents without the need for intra-document ranking.

Table 3: Results on the Text Classification datasets. Macro F1 score is reported. The highest values are in bold. The 'F' or 'C' following the dash represents input based on 'Facts' or 'Considerations' respectively. 'CPB' and 'CPC' refer to the CP task using LD and Citation labels, respectively, while 'SLAP' denotes Sub Law Area Prediction. Note: Seeds that yielded very high evaluation losses were considered failed and, therefore, excluded from the analysis. The models marked with an asterisk (*) are LLMs that generated zero-shot predictions (based on prompts) on a maximum of samples from the validation dataset, as described in section 5.2.

| Model | CPB-F | CPB-C | CPC-F | CPC-C | SLAP-F | SLAP-C | JP-F | JP-C | Agg. |
|---|---|---|---|---|---|---|---|---|---|
| MiniLM | 54.7 | 65.8 | 9.8 | 20.8 | 59.7 | 61.1 | 58.1 | 78.5 | 32.4 |
| DistilmBERT | 56.2 | 65.4 | 19.6 | 22.1 | 63.7 | 65.9 | 59.9 | 75.5 | 42.1 |
| mDeBERTa-v3 | 55.1 | 69.8 | 21.0 | 17.5 | 63.8 | 59.3 | 60.6 | 77.9 | 40.2 |
| XLM-R$_{Base}$ | 57.2 | 65.9 | 21.3 | 23.7 | 67.2 | 73.4 | 60.9 | 79.7 | 44.6 |
| XLM-R$_{Large}$ | 56.4 | 67.9 | 24.4 | **29.1** | 65.1 | 78.9 | 60.8 | 80.9 | **48.6** |
| X-MOD$_{Base}$ | 56.6 | 67.8 | 20.0 | 20.6 | 63.9 | 64.4 | 60.5 | 79.1 | 41.9 |
| SwissBERT$_{(xlm-vocab)}$ | 56.9 | 67.3 | 25.7 | 23.0 | 61.5 | 73.2 | 61.4 | 79.4 | 46.1 |
| mT5$_{Small}$ | 52.2 | 62.1 | 13.2 | 17.9 | 53.1 | 60.9 | 58.9 | 74.2 | 34.4 |
| mT5$_{Base}$ | 52.1 | 61.5 | 14.0 | 19.7 | 58.4 | 61.8 | 54.5 | 72.0 | 35.9 |
| BLOOM$_{560M}$ | 53.0 | 61.7 | 10.7 | 8.0 | 52.6 | 53.2 | 60.5 | 73.4 | 24.9 |
| Legal-Swiss-RoBERTa$_{Base}$ | 57.7 | 70.5 | 16.2 | 20.1 | 77.0 | 79.7 | 64.0 | 86.4 | 40.9 |
| Legal-Swiss-RoBERTa$_{Large}$ | 55.9 | 68.9 | **25.8** | 16.3 | 76.9 | **84.9** | 62.8 | **87.1** | 43.3 |
| Legal-Swiss-LF$_{Base}$ | **58.1** | **70.8** | 21.4 | 17.4 | **80.1** | 77.1 | **65.4** | 86.4 | 42.5 |
| GPT-3.5* | 46.6 | 44.8 | 25.7 | 16.7 | 67.9 | 69.5 | 51.3 | 61.9 | 38.6 |
| Claude-2* | 38.4 | 40.5 | 16.6 | 19.7 | 60.2 | 60.5 | 48.9 | 48.1 | 33.9 |
| LLaMA-2* | 45.2 | 26.6 | 7.0 | 8.5 | 58.7 | 55.6 | 40.3 | 37.8 | 19.7 |
| PaLM-2* | 40.6 | 38.5 | 16.6 | 14.3 | 57.6 | 67.8 | 52.6 | 65.3 | 32.3 |

## 6 RESULTS

### 6.1 TEXT CLASSIFICATION

We present results in Table 3, with detailed information including standard deviations in Table 15. Language-specific scores are in Table 16. Scores on the validation dataset are in Table 17. As expected, larger models generally perform better, with XLM-R$_{Large}$ emerging on top. Our pre-trained model Legal-ch R$_{Base}$ outperformed XLM-R$_{Base}$, indicating that domain-specific pre-training enhances performance. Overall, our pre-trained models showed better aggregated results compared to other models. However, unexpectedly, Legal Swiss RoBERTa$_{Large}$ underperformed compared to its base model XLM$_{Large}$. Due to the high weight to outliers allotted by the harmonic mean, Legal-ch-R$_{Large}$ is severely penalized by its relatively low performance in CPC-C compared to XLM-R$_{Large}$. Despite extra training on longer texts up to 4096 tokens, Legal-ch-LF did not surpass the hierarchical Legal-ch-R-$_{Base}$ model. Large models such as GPT-3.5, Claude-2, and LLaMA-2 underperform fine-tuned models, underlining the need for specialized models for these tasks. The difference is largest in the JP and Sub Law Area Prediction (SLAP) tasks where the fine-tuned models are best.

### 6.2 TEXT GENERATION

**Court View Generation** We present CVG results in Table 4. Fine-tuning models generally leads to higher scores, with even small models like mT5$_{Small}$ outperforming 1-shot GPT-4. 1-shot prompting offers marginal gains over 0-shot for both GPT-4 and Claude-2 (LLaMA-2 1-shot was limited by context width). The generated text generally showed stylistic authenticity, resembling typical legal language, but often lacked logical coherence, highlighting

Table 4: Results on the Court View Generation task. The input is truncated to 2048 tokens. **Bold**: best within setup; underlined: best overall. (*) These models were fine-tuned on only 1'000 samples for 3 epochs. All models, except the mT5 models, were evaluated on the validation set.

| Model | Setup | BERT ↑ | BLEU ↑ | MET ↑ | R1 / R2 / RL ↑ |
|---|---|---|---|---|---|
| mT5$_{Large}$ | Fine-tuned | **75.74** | **66.92** | **34.44** | **34.91** / 15.58 / **33.53** |
| mT5$_{Base}$ | Fine-tuned | 75.01 | 65.48 | 32.89 | 33.23 / 13.57 / 31.89 |
| mT5$_{Small}$ | Fine-tuned | 74.13 | 63.97 | 30.96 | 31.29 / 11.01 / 29.90 |
| GPT-3.5-Turbo | Fine-tuned* | 72.31 | 62.23 | 28.08 | 26.06 / 7.19 / 24.54 |
| LLaMA-2-13B Chat | Fine-tuned* | **74.22** | **63.51** | **33.33** | **34.36** / 16.68 / **33.20** |
| GPT-4 | 1-shot | **70.39** | 59.69 | 24.63 | 23.87 / 4.64 / 22.32 |
| Claude-2 | 1-shot | 69.45 | **61.26** | **24.85** | **24.98 / 5.39 / 23.61** |
| GPT-4 | 0-shot | 69.41 | 58.16 | 23.25 | 22.61 / 3.95 / 21.10 |
| Claude-2 | 0-shot | **69.46** | **61.36** | **24.90** | **24.66 / 5.30 / 23.24** |
| LLaMA-2-13B Chat | 0-shot | 67.23 | 55.01 | 20.18 | 19.76 / 3.26 / 18.57 |

current models' limitations in generating coherent court views. In multiple court cases, target considerations contained similar paragraphs, which were generally well-predicted (see examples in Table 38). While fine-tuned models proficiently predict specific textual patterns, LLaMA-2-13B-Chat in the zero-shot setup struggles, often reverting from German to English and introducing linguistic errors, probably due to a highly English dominant training corpus. Despite their challenges, zero-shot

models focus more on the main content, while fine-tuned models mirror target formalities. We provide a more detailed error analysis in Appendix I.1. Larger mT5 models consistently outperformed smaller ones, but performance increase with longer input was minimal, sometimes counterproductive (see Table 11). The results from the origin dataset were less conclusive (see Table 10), likely due to the smaller dataset size.

**Leading Decision Summarization** In contrast to the very specific CVG task, requiring long-form output, the closed large models perform very well on LDS, at least in BLEU and ME-TEOR (see Table 5). According to ROUGE, fine-tuned mT5 models are still better, while BERT-Score does not discriminate clearly. We assume that summarization is a much larger portion of internal instruction tuning

Table 5: Results on the Leading Decision Summarization task. The input is truncated to 4096 tokens. **Bold**: best within setup; underlined: best overall.

| Model | Setup | BERT ↑ | BLEU ↑ | MET ↑ | R1 / R2 / RL ↑ |
|---|---|---|---|---|---|
| mT5$_{Base}$ | Fine-tuned | **73.33** | 30.81 | **23.50** | **32.43** / **12.78** / **30.87** |
| mT5$_{Small}$ | Fine-tuned | 72.04 | 28.68 | 21.29 | 29.61 / 10.31 / 28.12 |
| GPT-4 | 1-shot | **73.55** | **47.75** | **34.72** | **30.82** / **9.68** / **28.89** |
| GPT-3.5-Turbo-16K | 1-shot | 72.89 | 45.21 | 32.76 | 29.69 / 9.25 / 27.94 |
| Claude 2 | 1-shot | 72.91 | 47.55 | 33.57 | 30.28 / 9.12 / 28.58 |
| Claude Instant | 1-shot | 72.44 | 44.80 | 30.29 | 27.89 / 8.56 / 26.18 |
| GPT-4 | 0-shot | **71.56** | 48.35 | **32.97** | 26.52 / **8.93** / 24.51 |
| GPT-3.5-Turbo-16K | 0-shot | 70.28 | 46.08 | 30.60 | 25.18 / 7.58 / 23.59 |
| Claude 2 | 0-shot | 71.13 | **49.20** | 32.54 | **27.70** / 8.39 / **25.90** |
| Claude Instant | 0-shot | 71.33 | 45.65 | 29.22 | 26.13 / 8.16 / 24.15 |

datasets used for optimizing these models. The quality of the generated text demonstrated a good stylistic imitation of legal language and more consistent logical coherence compared to the CVG task (see examples in Table 39 and Table 29). GPT-3.5-Turbo (zero-shot) offered a narrative-style summary, while others adhered to the traditional 'Regeste' format. Notably, GPT-3.5-Turbo made a factual error by negating a crucial element, and Claude-2 referenced an outdated legal provision. See Appendix I.2 for more detailed error analysis. Table 13 shows two trends for fine-tuned mT5 models. First, longer input generally improved scores across models. Second, larger models outperformed smaller ones, although the differences between base and large models were subtle.

## 6.3 INFORMATION RETRIEVAL

Table 6 shows that most models failed to retrieve relevant documents, even with k=100. Lexical models out-performed others even without hy-perparameter optimization for BM25 (Chalkidis et al., 2021b). Surpris-ingly, despite German prevalence in our dataset, a French language ana-lyzer (used for stemming and stop-word removal) demonstrated superior performance. For SBERT, truncation

Table 6: Results on Information Retrieval with best scores per section in **bold**. Abbreviations: **distil**use$_{Base}$-multilingual-cased-v1, **swiss**-legal-roberta$_{Base}$

| Model | RCap@ 1 / 10 / 100 ↑ | NDCG@ 1 / 10 / 100 ↑ |
|---|---|---|
| BM25 (fr lang analyzer) | **11.37** / **7.74** / **16.54** | **11.37** / **8.34** / **11.51** |
| SBERT distil | 0.90 / 0.75 / 2.64 | 2.06 / 1.70 / 3.31 |
| SBERT distil + pos | **4.40** / 3.92 / 12.64 | **10.11** / 8.76 / 16.16 |
| SBERT distil + pos + h-neg | 3.97 / **4.46** / **13.36** | 9.12 / **9.21** / **16.87** |
| SBERT swiss + pos | 3.97 / 3.47 / 12.28 | 9.12 / 7.76 / 15.16 |
| SBERT distil eval on de queries | **4.22** / **4.49** / **15.21** | **8.21** / **8.15** / **15.86** |
| SBERT distil eval on fr queries | 1.88 / 2.20 / 9.19 | 5.77 / 6.22 / 13.94 |
| SBERT distil eval on it queries | 0.22 / 0.24 / 0.79 | 5.43 / 5.74 / 11.44 |

led to context loss, negatively affecting scores, a problem absent in lexical models. Training SBERT models using Multiple Negative Ranking Loss (Henderson et al., 2017) significantly improved per-formance, with hard negative examples beneficial. SBERT evaluation on single languages, denoted as DE, FR, and IT, revealed its inability to perform consistently across all languages, which could be caused by the training set consisting of more German than French or Italian documents. More experiments can be found in Tables 8 and 9. Overall, our study exposes limitations of models in handling multilingualism, long documents, and legal texts, areas relatively underexplored in previous research. These findings offer a foundation for the IR community to address these challenges.

## 7 CONCLUSIONS

We present SCALE, an end-to-end benchmark of seven datasets for the Swiss legal system, a world-wide unique possibility to study crosslinguality within the same jurisdiction. Our tasks require legal reasoning abilities and challenge models on four key aspects: long documents, domain-specificity, multilinguality, and multitasking. We evaluate 14 open and five closed multilingual models, including three in-domain pretrained, as a reference point. These models, including ChatGPT, Claude-2, LLaMA-2 and PaLM-2, show low performance, particularly in challenging tasks like CVG and IR. Our results highlight opportunities for improving models and set the stage for next-generation LLM evaluations in domain-specific, multilingual contexts.

ETHICS STATEMENT

While our research has several positive applications, it is important to acknowledge potential negative societal impacts. Large Language Models (LLMs) and their applications in the legal domain could potentially automate certain tasks traditionally performed by legal professionals, such as legal IR and LDS. While our goal is to support lawyers, it could impact the job market for legal professionals.

Recent literature has identified potential ethical problems within legal NLP research. The study on legal judgement prediction in China regarding prison term duration demonstrated the criticalness of legal NLP datasets and analysis Chen et al. (2019). As a response to this publication at EMNLP 2019, concerned researchers pointed out important questions to respond when working with ethically delicate data and NLP tasks Leins et al. (2020). They suggest asking a series of ethical questions to assess the potential societal risks associated with a publication.

For example, they asked: "Does the dataset contain information that might be considered sensitive or confidential?" Leins et al. (2020). In our case, we only used publicly available court decisions that are anonymized. Therefore, the person should not be identifiable. Another aspect is concerned with the possibility of future updates of the court decision due to new facts or an appeal (going to a higher court): "Will the dataset be updated?" We could update our data anytime. However, this maintenance would not happen automatically. We would need to be informed that a new decision was made regarding a certain case.

Another ethical concern concerns precision: Thus, LLMs can occasionally deliver results that are not entirely precise. This can have severe implications when it comes to the legal domain, where precision and factual accuracy are paramount. This could potentially lead to misinformation or misinterpretation of legal texts, impacting legal proceedings and decisions.

While the benchmark focuses on the Swiss legal system, it is important to recognize that law systems are highly culturally and contextually dependent. The understanding and interpretations of legal texts by these models, especially in a multilingual context, might not accurately reflect the nuanced cultural aspects of different regions. This could potentially lead to misrepresentations or misinterpretations, particularly when applied to other legal systems.

Finally, like any AI model, LLMs could be misused to create misinformation or misleading content at scale, especially in languages and domains where automated content generation is still a novel concept. It is crucial to develop and implement robust ethical guidelines and policies to mitigate these risks.

Therefore, while the new benchmark presents exciting opportunities to improve LLMs, it is essential to carefully consider the implications of its use and manage the associated risks effectively. The developers and users of such technology should adhere to ethical guidelines to ensure its responsible use.

REPRODUCIBILITY STATEMENT

Datasets, models, and source code for both dataset curation and experiments will be publicly released upon acceptance. The exact links will be provided in Appendix B. We detail experimental setup in section 5 and in more detail in Appendix F. All prompts used for evaluation of models larger than 2B parameters are listed in Appendix J.

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

## A    APPENDIX TABLE OF CONTENTS

We include the following supplementary sections in addition to the main paper:

## B  ACCESS TO THE PROVIDED RESOURCES

In this section, we provide the URLs to the data, models, and code upon acceptance.

## C    LIMITATIONS

### C.1    GENERAL

The research area for language models and benchmarks continues to evolve, and while there is palpable enthusiasm in the field, it is critical to maintain a balanced perspective. Studies, including **?**, have shed light on the limitations of language models and benchmarks, stressing that language models do not truly "learn" meaning and that communities often focus on limited datasets, some of which are borrowed from other fields.

### C.2    MODELS

Even though English models are plentiful, LLMs pre-trained multilingually are very rare. To the best of our knowledge, mT5 is the only multilingual model with variants over 1B parameters, covering German, French, and Italian (BLOOM does not contain German and Italian). Additionally, we were limited by very large sequence lengths. We did not have the resources available to run $\text{mT5}_{\text{XL}}$ or $\text{mT5}_{\text{XXL}}$ with sequence lengths greater than 1K.

### C.3    DATA

The process of selecting tasks for benchmarks is typically influenced by the interests of the community or the convenience of available resources, rather than being informed by all-encompassing theories. These constraints present difficulties when trying to explore a model's broader applicability or its capacity for understanding. The data employed in benchmarks is often tied to a specific context and is naturally susceptible to inherent biases. Furthermore, the content of such data may vary significantly from real-world data, is de-contextualized and the uniformity of the task formats may not adequately reflect the diversity of human activities. Regarding our specific context, it is crucial to acknowledge that we cannot generalize Swiss legal data to other countries or different legal systems.

Figures 6 and 8 show the language and cantonal distributions over rulings and Figures 7 and 9 over legislation. Note that the distribution is imbalanced for both rulings (50% German, 39% French, and 11% Italian) and legislation (49% German, 31% French, and 17% Italian). However, compared to (Swiss Speaker Distribution (63% German, 23% French, and 8% Italian), the legal text is actually more balanced. It looks similar in the cantonal distribution, with many sparsely inhabited cantons being represented above their weight, especially the ones in the non-German speaking regions such as Vaud, Geneva, and Ticino.

### C.4    LABELS

Annotating high-quality datasets is very expensive, especially when experts are needed, such as in the legal domain. Because of our limited budget and in order to arrive at a large amount of labels, we algorithmically generated labels based on metadata information present in the corpus. These metadata are of high quality, being provided by the courts themselves.

### C.5    TEXT CLASSIFICATION

**Judgment Prediction**   While the judgment prediction task is arguably very interesting and also very challenging, it is unlikely to be deployed in practice anytime soon. Ideally, we would want the complaints as input (similar to Semo et al. (2022)) instead of the facts description, since this is written by the court itself in part to justify its reasoning. Unfortunately, the complaints are not public in Switzerland, making us rely on the widely available facts description as a close proxy.

**Law Area Prediction**   We used information about the chambers at the courts to determine the law areas. Predicting the main law area is not challenging for current models, leading to very high results and thus rendering this task unsuitable for a benchmark. Unfortunately, most chambers cover multiple sub areas, thus ruling them out for the sub area prediction task and considerably reducing dataset size. In conclusion, while this task is very useful in practice for routing requests to the different chambers inside a court, it relatively unsuitable for a challenging benchmark.

**Criticality Prediction**    It is very difficult to estimate the importance of a case. By relying on proxies such as whether the case was converted to a leading decision (LD-label) and how often this leading decision was cited (Citation-label), we were able to create labels semi-automatically. While we discussed this with lawyers at length and implemented the solution we agreed on finally, this task remains somewhat artificial.

## C.6    TEXT GENERATION

**Court View Generation**    Court View Generation is an extremely challenging task and thus very well suitable for a benchmark. Current multilingual transformer-based models do not allow processing text in the tens of thousands of tokens. As a consequence, we were forced to look at a simplified version of this task, only considering the facts as input and ignoring relevant case law and legislation. Additionally, we were only able to generate the first 512 tokens of the considerations. We thus invite the community to develop new methods potentially capable of tackling harder versions of this task.

**Leading Decision Summarization**    Due to limited resources, we limited our evaluation to mT5$_{\text{Small/Base/Large}}$. Future work may investigate large multilingually pretrained generative models on this task. Additionally, one may want to conduct human evaluation on the generated summaries. Finally, we only considered the simple version of this task where we only generate a text-based output. Future work may treat the first and second parts of the summary as extreme multilabel classification problems of relevant citations and relevant keywords from the thesaurus Jurivoc respectively, possibly increasing performance.

## C.7    CITATION EXTRACTION

Even though the citations are annotated by the SFCS, we encountered citations that were not marked. However, models achieved very high scores anyway, leading us to exclude it from the benchmark. Future work may investigate this in more detail.

## C.8    INFORMATION RETRIEVAL

The labels are constructed with the citations from the considerations. Due to most legal analyses being private, our corpus is restricted to case law and legislation. Constructing a ranking of relevant documents is challenging due to missing information, and thus probably requiring extensive human annotation. Additionally, S-BERT models are usually limited to 512 tokens, being a constraint for this task due to our long documents.

# D    DIRECTIONS FOR FUTURE RESEARCH

The political parties of the judges in the ruling determine in what direction the ruling will go. In future work we would like to enrich the dataset with this information to make models more accurate in the judgment prediction task.

For simplicity, we treat the summary (regeste) of the leading decisions as just one string. Actually, it is composed of important citations in the first part, keywords from a Thesaurus in the second part and a text-based summary in the third part. The first two tasks could be framed as classification or retrieval tasks, possibly improving model performance.

Due to limited context width, we only considered the facts as input to the court view generation task. However, judges and clerks do not only look at the facts when drafting a decision. They consider a myriad of information including possible lower court decisions and relevant case law, legislation and legal analyses. This information is available in our dataset. In the future, we would like to develop systems that are capable of integrating all this information to write the legal reasoning.

So far, to our knowledge, the largest model pretrained on legal data specifically is Legal-XLM-R$_{Large}$ (435M parameters) Niklaus et al. (2023b). Future work should look at pretraining larger generative models in the billions and tens of billions of parameters.

Future work may investigate the more difficult Citation Prediction task in addition to the Citation Extraction task. In Citation Prediction, the model only gets the context up to the citation as input and is tasked to predict the citation. This may help lawyers in drafting their texts.

We strongly suggest future work include relevant external information, like caselaw or legislation for solving these challenging tasks. Augmenting models with retrieval Lewis et al. (2021) and models using tools Schick et al. (2023) seem to be promising avenues.

Finally, to provide a better perspective on the results, we suggest future work to collect human performance as an additional reference point. This may be done at several levels, i.e., laypeople, law students, early career lawyers, expert lawyers in the respective field.

# E ADDITIONAL RELATED WORK

## E.1 DOMAIN-SPECIFIC PRETRAINING

General-purpose language models are trained on generic text corpora such as Wikipedia and evaluated on widely used benchmarks such as GLUE (Wang et al., 2018). However, domain-specific models need focused datasets for training and specialized benchmarks for assessing the quality of the model. The following examples illustrate the increase in performance when using domain-specific datasets and benchmarks.

In the biomedical area of natural language processing (BioNLP), Lee et al. (2019) created for the first time a domain-specific LM based on BERT (Devlin et al., 2019) by pre-training it on biomedical text corpora. They used PubMed abstracts (4.5B words) and PubMed Central (PMC) full-text articles (13.5B words). The resulting domain-specific LM BioBERT achieved higher F1 scores than BERT in the biomedical NLP tasks named entity recognition (0.62) and relation extraction (2.80), and a higher mean reciprocal rank (MRR) score (12.24) in the biomedical question-answering task. In 2022, those scores were outperformed. Naseem et al. (2022) conducted a domain-specific pre-training of ALBERT Lan et al. (2020) using only text from the biomedical field (PudMed) and from the "Medical Information Mart for Intensive Care" (MIMIC-III), a large, de-identified and publicly-available collection of medical records (Johnson et al., 2016). One domain-specific benchmark applied to test BioALBERT originates from Gu et al. (2021) who created BLURB, the Biomedical Language Understanding and Reasoning Benchmark. Naseem et al. (2022) found that BioALBERT exceeded the state-of-the-art models by 11.09% in terms of micro averaged F1-score (BLURB score). Another biomedical NLP benchmark is BLUE, the "Biomedical Language Understanding Evaluation" (Peng et al., 2019b). It covers five tasks (sentence similarity, named entity recognition, relation extraction, document classification, inference) with ten datasets from the biomedical and clinical area. BioALBERT also includes all datasets and tests from BLUE thus presenting the most comprehensive domain-specific model and benchmark in the biomedical area at the moment.

In the financial domain, FinBERT was pretrained 2020 by Yang et al. (2020) using financial data. The text corpora consisted of 203'112 corporate reports (annual and quarterly reports from the Securities Exchange Commission SEC), 136'578 earnings call transcripts (conference call transcripts from CEOs and CFOs), and 488'494 analyst reports (textual analysis of the company) resulting in 3.3B tokens. For testing FinBERT, Yang et al. (2020) used the Financial Phrase Bank dataset with 4'840 sentiment classifications (Malo et al., 2014), the AnalystTone Dataset with 10'000 sentences (Huang et al., 2014), and FiQA Dataset with 1'111 sentences from an open challenge dataset for financial sentiment analysis (Financial Opinion Mining and Question Answering). The results show that the domain-specific FinBERT outperforms the generic BERT models in all of these financial datasets. An improved financial domain LM was released 2022 by Shah et al. (2022) by introducing FLANG-BERT, the Financial LANGuage Model. They also created a domain-specific benchmark, Financial Language Understanding Evaluation (FLUE). Recently in May 2023, Bloomberg announced the BloombergGPT model, a Large Language Model (LLM) for the financial domain (Wu et al., 2023). However, next to some experience on the training process no datasets, benchmarks, or weights have been released publicly.

Numerous other domain-specific LMs have been created since the rise of BERT. They all outperform general-purpose LMs. For instance, SciBERT is a pretrained LM based on scientific publications and evaluated on a suite of tasks in difference scientific domains (Beltagy et al., 2019). ConfliBERT is built to improve monitoring of political violence and conflicts (Hu et al., 2022) and PoliBERTweet is used to analyze political content on Twitter (Kawintiranon & Singh, 2022). Cybersecurity is another important area thus Aghaei et al. (2023) pretrained a M on a large corpus of cybersecurity text. To improve IR tasks in the architecture, engineering, and construction (AEC) industry, Zheng et al. (2022) pretrained BERT on a corpus of regulatory text. Also, the domain-specific model BlueBERT (Peng et al., 2019b) from the biomedical domain has been further pretrained and evaluated on more narrow, cancer-related vocabularies, resulting in CancerBERT (Zhou et al., 2022).

In the legal domain Chalkidis et al. (2020) pretrained LegalBERT on EU and UK legislation, ECHR and US cases and US contracts. Zheng et al. (2021) pretrained CaseHoldBERT on US caselaw. Henderson et al. (2022) trained PoL-BERT on their 256 GB diverse Pile of Law corpus. Niklaus & Giofré (2022) pretrained longformer models using the Replaced Token Detection (RTD) task Clark et al. (2020) on the Pile of Law. Hua et al. (2022) pretrained reformer models with RTD on 6 GB

of US caselaw. Finally, Niklaus et al. (2023b) released a large multilingual legal corpus and trained various legal models on it.

## E.2 JUDGMENT PREDICTION

The domain of Legal Judgment Prediction (LJP) centers around the crucial task of predicting legal case outcomes given the provided facts. In the landscape of LJP research, there have been significant advances focusing on diverse languages, jurisdictions, and input types. Researchers have utilized a variety of datasets, each with their unique characteristics and annotations, to analyze and predict case outcomes (Feng et al., 2022; Aletras et al., 2016; Şulea et al., 2017; Medvedeva et al., 2018; Chalkidis et al., 2019a).

In the context of Chinese criminal cases, notable efforts have been made by Xiao et al. (2018; 2021), where they utilized the CAIL2018 dataset, which consists of over 2.6M cases and provides annotations for Law Article, Charge, and Prison Term, among others.

Focusing on the Indian and Swiss jurisdictions, Malik et al. (2021) and Niklaus et al. (2021; 2022) employed the ILDC and SJP datasets respectively, both using binary labels. The ILDC dataset, with over 34K Indian Supreme Court cases, offers sentence-level explanations along with Court Decision annotations, while the SJP corpus is trilingual, containing judgments from Switzerland in German, French, and Italian, and provides annotations like the publication year, legal area, and the canton of origin.

European jurisdictions have been explored using the ECHR2019 and ECHR2021 datasets (Chalkidis et al., 2019a; 2021c). These corpora feature cases from the European Court of Human Rights, annotated for Violation, Law Article, and Alleged Law Article, among others, with the latter also providing paragraph-level rationales.

The FCCR dataset, containing over 126K cases from France, has been used to predict Court Decisions with different setups, offering additional annotations such as the date of the court ruling and the law area (Şulea et al., 2017).

Recently, Semo et al. (2022) introduced a new perspective on LJP, applying it to US class action cases. The proposed task involves predicting the judgment outcome based on the plaintiff's pleas, further expanding the scope of LJP research and making the task more realistic.

These efforts underscore the breadth of LJP research, demonstrating its applicability across multiple jurisdictions, languages, and legal systems, and its potential in assisting legal professionals and enhancing access to justice.

## E.3 CRITICALITY PREDICTION

Chalkidis et al. (2019a) introduced the Importance Prediction task, which predicts the importance of a ECtHR case on a scale from 1 (key case) to 4 (unimportant). Legal experts defined and assigned these labels for each case, representing a significant contrast to our approach where labels were algorithmically determined. This is to our knowledge the only comparable task to Criticality Prediction.

## E.4 LAW AREA PREDICTION

Although not widely studied, several notable works have focused on LAP. Şulea et al. (2017) worked on the Law Judgment Prediction (LJP) task, using a dataset of over 126K cases from the French Supreme Court. The study used Linear Support Vector Machines (SVMs) to classify cases into one of eight law areas, using the entire case description as input. This approach yielded an F1 score of 90%. Soh et al. (2019) conducted a similar study using a dataset of 6K judgments in English from the Singapore Supreme Court. These judgments were mapped into 30 law areas. Several text classifiers were used in the study, achieving a macro F1 score of up to 63.2%.

### E.5 Court View Generation

Over the past decade, text generation in the field of Legal NLP has been underexplored (Katz et al., 2023b), especially in comparison to tasks such as classification and information extraction. Li & Zhang (2021b) utilize Chinese case facts, as well as charge (formal accusations of crimes) and law article information, to generate court opinions. A key difference from our task is the shortness of their opinions (avg. 31/34 tokens), while ours span approximately four thousand tokens on average. With the emergence of powerful generative models, we expect a surge in research activity in this area, necessitating challenging benchmarks to assess progress effectively.

### E.6 Leading Decisions Summarization

In the field of legal text summarization, several noteworthy contributions have been made (?Hachey & Grover, 2006; Kim et al., 2013; Jain et al., 2021), with the BillSum (Kornilova & Eidelman, 2019) and Multi-LexSum (Shen et al., 2022) datasets being particularly significant. The creators of the BillSum dataset focused on summarizing 22K bills from the US Congress and the state of California. They also applied transfer learning in summarization from federal to state laws. Models based on BERT and TF-IDF, as well as a combination of both, have been evaluated on this dataset. The BillSum dataset focuses on English language documents related to the US legislative environment. The Multi-LexSum dataset is another significant development in the area of legal text summarization. It targets long civil rights lawsuits, with an average length of over 75K words. This 9K-document dataset allows for in-depth study at different summary lengths: short (25 words), medium (130 words), and long (650 words), a unique feature of the Multi-LexSum dataset. Models based on BART Lewis et al. (2020) and PEGASUS Zhang et al. (2020a) were evaluated on this dataset. Like BillSum, the Multi-LexSum dataset is primarily for the English language and is relevant to the US legal setting.

### E.7 Citation Extraction

Early work from Martínez-González et al. (2005) extract citations from legal text with patterns. Nambanoor Kunnath et al. (2022) studied the effect of differing context size for citation classification in scientific text. Taylor et al. (2022) considered the more difficult Citation Prediction task on scientific text and found that larger models are more true to the real citation distribution, whereas smaller models tend to output the most frequent citations most of the times.

### E.8 Information Retrieval

Lawrie et al. (2023) revisited the challenges of multilingual IR and proposed neural approaches to address this issue. They demonstrated that combining neural document translation with neural ranking resulted in the best performance in their experiments conducted on the MS MARCO dataset Bajaj et al. (2018). However, this approach is computationally expensive. To mitigate this issue, they showed that using a pre-trained XLM-R multilingual model to index documents in their native language resulted in only a two percent difference in effectiveness. XLM-R is a transformer-based masked language model that employs self-supervised training techniques for cross-lingual understanding Conneau et al. (2020a). Lawrie et al. (2023) crucially utilized mixed-language batches from the neural translation of MS MARCO passages.

A widely used technique is BM-25, which is an improved retrieval method that considers the term frequencies and takes into account the saturation effect and document length Robertson & Zaragoza (2009). The saturation effect refers to the point where the relevance of a term stops increasing, even if it appears many times in a document. This issue is mitigated through the use of an additional parameter, k. Additionally, longer documents are more likely to contain a higher number of occurrences of a term simply because they contain more words, not necessarily because the term is more relevant to the document, which is why parameter b is used. The BM-25 score is calculated using the Inverse Document Frequency (IDF), Term Frequency (TF), queries Q, documents d, and term t.

$$BM25(d, Q, b, k) = \sum_{t \in Q} IDF(t) \frac{(k+1)TF(t,d)}{(1-b)(b*A)+TF(t,d)}$$

Chalkidis et al. (2021b) proposed a new IR task called REG-IR, which deals with longer documents in the corpus and entire documents as queries. This task is an adaptation of Document-to-Document (Doc2Doc) IR, which aims to identify a relevant document for a given document. The authors observed that neural re-rankers underperformed due to contradicting supervision, where similar query-document pairs were labeled with opposite relevance. Additionally, they demonstrated for long documents that using BM25 as a document retriever in a two-stage approach often results in underperformance since the parameters k and b are often not optimal when using standard values. The problem of noise filtering of long documents was also addressed by using techniques like stopwords removal. However, as seen in Leveling (2012), this approach can have a negative effect on performance. The best pre-fetcher for long documents was found to be C-BERTs Chalkidis et al. (2021a), which are trained on classifying documents using predefined labels.

Thakur et al. (2021) proposed a novel evaluation benchmark for IR that encompasses a wide range of approaches, including BM25, dense, and re-ranking models. They found that while BM25 is computationally expensive, it provides a robust baseline, whereas other models failed to achieve comparable performance. Their findings suggest that there is still much room for improvement in this area of NLP. Efficient retrieval of relevant information is crucial for many NLP tasks, and these results highlight the need for continued research in this area.

# F    MORE DETAILED EXPERIMENTAL SETUP

Table 7: Models: InLen is the maximum input length the model has seen during pretraining. # Paramaters is the total parameter count (including embedding). Our models were built upon the pre-trained RoBERTa/Longformer. SwissBERT was further trained from X-MOD. Utilizing three language adapters with X-MOD and SwissBERT led to fewer parameters and languages. (%de/%fr/%it) shows the percentages of the Swiss languages in the corpus. A question mark in brackets (?) indicates that the we could not find reliable sources.

| Model | Source | InLen | # Parameters | Vocab | # Steps | BS | Corpus (%de/%fr/%it) | # Langs |
|---|---|---|---|---|---|---|---|---|
| MiniLM | Wang et al. (2020b) | 512 | 118M | 250K | 1M | 256 | 2.5TB CC100 (2.9/2.5/1.3) | 100 |
| DistilmBERT | Sanh et al. (2020) | 512 | 135M | 120K | n/a | < 4000 | Wikipedia (na/na/na) | 104 |
| mDeBERTa-v3 | He et al. (2021b;a) | 512 | 278M | 128K | 500K | 8192 | 2.5TB CC100 (2.9/2.5/1.3) | 100 |
| XLM-R$_{Base/Large}$ | Conneau et al. (2020b) | 512 | 278M/560M | 250K | 1.5M | 8192 | 2.5TB CC100 (2.9/2.5/1.3) | 100 |
| X-MOD$_{Base}$ | Pfeiffer et al. (2022) | 512 | 299M | 250K | 1M | 2048 | 2.5TB CC100 (2.9/2.5/1.3) | 3 (81) |
| SwissBERT (XLM vocab) | Vamvas et al. (2023) | 512 | 299M | 250K | 364K | 768 | Swissdox (80/18/1) | 3 (4) |
| mT5$_{Small/Base/Large}$ | Xue et al. (2021) | 1K | 300M/580M/1.2B | 250K | 1M | 1024 | mC4 (CC) (3.1/2.9/2.4) | 101 |
| BLOOM$_{560M}$ | Scao et al. (2022) | 2K | 560M | 250K | 1.3M | 256 | ROOTS (0/5/0) | 59 |
| Legal-Swiss-R$_{Base}$ | ours | 512 | 184M | 128K | 1M | 512 | CH Legal (50/27/23) | 3 |
| Legal-Swiss-R$_{Large}$ | ours | 512 | 435M | 128K | 500K | 512 | CH Legal (50/27/23) | 3 |
| Legal-Swiss-LF$_{Base}$ | ours | 4096 | 208M | 128K | 50K | 512 | CH Legal (50/27/23) | 3 |
| Claude-2 | Anthropic | 100K | 137B (?) | na | na | na | na | na |
| Claude-Instant | Anthropic | 100K | 52B (?) | na | na | na | na | na |
| GPT-3.5 | Brown et al. (2020) | 16K | 175B | na | na | na | na | na |
| GPT-4 | OpenAI (2023) | 32K | 1.8T (?) | na | na | na | na | na |
| PaLM-2 | Anil et al. (2023) | 8K | 340B (?) | na | na | na | na | na |
| LLaMA-2 | Touvron et al. (2023) | 4K | 7B/13B/70B | 32K | na | na | LLaMA-2 (0.2/0.2/0.1) | 27 |

## F.1    PRETRAINING LEGAL MODELS

(a) We warm-start (initialize) our models from the original XLM-R checkpoints (base or large) of Conneau & Lample (2019). Model recycling is a standard process followed by many Wei et al. (2021); Ouyang et al. (2022) to benefit from starting from an available "well-trained" PLM, rather from scratch (random). XLM-R was trained on 2.5TB of cleaned CommonCrawl data in 100 languages.

(b) We train a new tokenizer of 128K BPEs on the training subsets to better cover legal language across languages. However, we reuse the original XLM-R embeddings for all lexically overlapping tokens **?**, i.e., we warm-start word embeddings for tokens that already exist in the original XLM-R vocabulary, and use random ones for the rest.

(c) We continue pretraining our models on our pretraining corpus with batches of 512 samples for an additional 1M/500K steps for the base/large model. We do initial warm-up steps for the first 5% of the total training steps with a linearly increasing learning rate up to $1e-4$, and then follow a cosine decay scheduling, following recent trends. For half of the warm-up phase (2.5%), the Transformer encoder is frozen, and only the embeddings, shared between input and output (MLM), are updated. We also use an increased 20/30% masking rate for base/large models respectively, where also 100% of the predictions are based on masked tokens, compared to **?**[5], based on the findings of Wettig et al. (2023).

(d) For both training the tokenizer and our legal models, we use a sentence sampler with exponential smoothing of the sub-corpora sampling rate following Conneau & Lample (2019) and Raffel et al. (2020), since there is a disparate proportion of tokens across languages (Figure 7) and we aim to preserve per-language capacity, i.e., avoid overfitting to the majority (almost 50% of the total number of texts) German texts.

(e) We consider mixed cased models, i.e., both upper- and lowercase letters covered, similar to all recently developed large PLMs Conneau & Lample (2019); Raffel et al. (2020); Brown et al. (2020).

(f) To better account for long contexts often found in legal documents, we continue training the base-size multilingual model on long contexts (4096 tokens) with windowed attention (128 tokens window size) Beltagy et al. (2020) for 50K steps, dubbing it Legal-Swiss-LF-base. We use the standard 15% masking probability and increase the learning rate to $3e-5$ before decaying but otherwise use the same settings as for training the small-context models.

---

[5]**?** – and many other follow-up work – used a 15% masking ratio, and a recipe of 80/10/10% of predictions made across masked/randomly-replaced/original tokens.

## F.2 RESOURCES USED

The experiments were performed on internal university clusters on NVIDIA GPUs with the following specifications: 24GB RTX3090, 32GB V100, 48GB A6000, and 80GB A100. We used an approximate total of 160, 20, and 2 GPU days for the text classification, text generation and information retrieval experiments.

**Text Generation** For inference and fine-tuning LLaMA-2 in the text generation task, we used the Together API.

## F.3 HYPERPARAMETERS

**Text Classification** For all models and datasets, a learning rate of 1e-5 was used without any tuning. Each experiment was executed with three random seeds (1-3), and the batch size was tailored for each task and corresponding computational resource. If the GPU memory was inadequate, gradient accumulation was employed as a workaround to arrive at a final batch size of 64. The training was conducted with early stopping based on validation loss, maintaining a patience level of 5 epochs. Due to the considerable size of the judgment prediction dataset and the extended duration of the experiment, training was limited to a single epoch with evaluations after every 1000th step. To reduce costs, we utilized AMP mixed precision during the training and evaluation phases whenever it did not lead to overflows (e.g., mDeBERTa-v3). We established the max-sequence-length (determined by the product of max-segment-length and max-segments in the hierarchical setup Aletras et al. (2016); Niklaus et al. (2021; 2022)) based on whether we used Facts: 2048 (128 X 16), or Considerations: 4096 (128 X 32).

**Text Generation** For the main CVG dataset, we trained our mT5 models for only one epoch (because of the large training set) with a final batch size of 16, using gradient accumulation as needed. We performed evaluations every 1000 steps. For the smaller origin dataset, we increased the number of epochs to 100 and evaluated every 100 steps. For the LDS task, we adjusted the training to 10 epochs.

**Information Retrieval** For the BM25 model, we used the same parameters as used in the BEIR paper Thakur et al. (2021), chosen were k = 0.9 and b = 0.4. For the SBERT model training, we employed the BEIR toolkit Thakur et al. (2021). Our training process was constrained by a maximum sequence length of 512 tokens. During the training phase, we completed a single epoch, comprising 5000 evaluation steps.

In the context of training with hard negative examples, we incorporated 5 negative examples for every query. The selection of these examples was based on the 5 highest-ranked erroneous predictions generated by the BM25 model. To facilitate training with these challenging negatives, we followed the guidelines provided by Thakur et al. (2021), utilizing the Hardnegs template.

```
"corpus": {
    "decision_id_bge": {
        "title": "file number",
        "text": "facts and considerations of a case"
    },
    "law_id": {
        "title": "",
        "text": "text of a law"
    }
},
"queries": {
    "decision_id_bger_1": "facts of a case",
    "decision_id_bger_2": "facts"
},
"qrels": {
    "decision_id_bger_1": {"law-id": 1, "decision-id-bge": 1},
    "decision_id_bger_2": {"law-id": 1}
}
```

Figure 3: Structure of corpus, queries and qrels for IR task

# G  ADDITIONAL RESULTS

## G.1  INFORMATION RETRIEVAL

Table 8: Results IR: using a subset of 100 queries and only relevant documents in the corpus resulting in an easier task

| Model | Additional | $Rcap@1\uparrow$ | $Rcap@10\uparrow$ | $Rcap@100\uparrow$ | $NDCG@1\uparrow$ | $NDCG@10\uparrow$ | $NDCG@100\uparrow$ |
|---|---|---|---|---|---|---|---|
| Train + Evaluate S-BERT | sbert-legal-xlm-roberta-base | 32.32 | 32.34 | 81.77 | 32.32 | 30.89 | 49.11 |
| Train + Evaluate S-BERT | sbert-legal-swiss-roberta-base | **36.36** | **35.68** | 76.03 | **36.36** | **34.54** | 49.90 |
| Train + Evaluate S-BERT | distiluse-base-multilingual-cased-v1 | 22.22 | 30.35 | 84.38 | 22.22 | 25.72 | 48.66 |
| Evaluate S-BERT | distiluse-base-multilingual-cased-v1 | 8.08 | 11.83 | 43.35 | 8.08 | 10.55 | 21.56 |
| Train(HN) + Evaluate S-BERT | distiluse-base-multilingual-cased-v1 | 27.27 | 33.94 | **86.81** | 27.27 | 30.09 | **52.03** |
| Dim Reduction | distiluse-base-multilingual-cased-v1 | 0.00 | 1.59 | 5.43 | 0.00 | 1.17 | 2.41 |
| Cross Encoder | distiluse-base-multilingual-cased-v1 | 5.94 | 8.04 | 14.20 | 2.97 | 1.84 | 7.35 |
| Lexical | | 5.94 | 8.04 | 14.20 | 5.94 | 8.52 | 10.64 |
| ML Lexical | 'German' | 9.90 | 8.41 | 15.19 | 9.90 | 9.14 | 11.58 |

Table 9: Results IR Additional: Results IR Abbreviations: Capped **R**ecall, NDCG, **dist**iluse-base-multilingual-cased-v1, swiss-legal-**rob**erta-base, legal-**xlm**-roberta-base, **T**rain, **H**ard **N**egative, **E**valuate, **S**-**B**ert, **Dim** Reduction

| Model | Adaption | | $R@1\uparrow$ | $R@10\uparrow$ | $R@100\uparrow$ | $N@1\uparrow$ | $N@10\uparrow$ | $N@100\uparrow$ |
|---|---|---|---|---|---|---|---|---|
| LR | | | 8.38 | 6.43 | 15.76 | 8.38 | 6.66 | 10.23 |
| LR | | S | **10.64** | 7.57 | 16.47 | **10.64** | 8.04 | 11.33 |
| LR | | SL | 7.91 | **9.99** | **32.46** | 9.13 | **9.65** | **18.03** |
| MLR | 'German' | | 8.69 | 6.54 | 15.99 | 8.69 | 6.82 | 10.43 |
| MLR | 'German' | S | **10.88** | 7.65 | 16.80 | **10.88** | 8.14 | 11.53 |
| MLR | 'German' | SL | 8.05 | 9.94 | 32.63 | 9.30 | **9.70** | **18.17** |
| MLR | 'French' | | **11.37** | 7.74 | 16.54 | **11.37** | **8.34** | 11.51 |
| MLR | 'French' | S | 10.97 | 7.60 | 16.52 | 10.97 | 8.14 | 11.41 |
| MLR | 'Italian' | | 10.08 | 7.118 | 16.294 | 10.08 | 7.582 | 11.021 |
| MLR | 'English' | | 8.38 | 6.43 | 15.76 | 8.38 | 6.66 | 10.23 |
| T+E SB | xlm | | 2.77 | 2.58 | 10.17 | 6.36 | 5.66 | 12.03 |
| T+E SB | rob | | 3.97 | 3.47 | 12.28 | 9.12 | 7.76 | 15.16 |
| E SB | dist | | 0.90 | 0.75 | 2.64 | 2.06 | 1.70 | 3.31 |
| T+E SB | dist | | 4.4 | 3.92 | 12.64 | 10.11 | 8.76 | 16.16 |
| T+E SB | dist | S | 4.69 | 4.14 | 13.39 | 10.77 | 9.27 | 17.05 |
| T+E SB | dist | SL | 1.79 | 3.92 | 14.17 | 4.03 | 6.17 | 12.91 |
| SB T(HN)+E | dist | | 3.97 | 4.46 | 13.36 | 9.12 | 9.21 | 16.87 |
| SB T(HN)+E | dist | S | 3.76 | 4.75 | 12.80 | 8.64 | 9.66 | 16.57 |
| SB T(HN)+E | dist | SL | 2.34 | 4.37 | 14.43 | 5.27 | 6.99 | 13.75 |
| T+E SB | dist | DE | 4.22 | 4.49 | 15.21 | 8.21 | 8.15 | 15.86 |
| T+E SB | dist | DE SL | 4.06 | 8.47 | 29.43 | 4.51 | 6.73 | 13.78 |
| T+E SB | dist | FR | 1.88 | 2.2 | 9.19 | 5.77 | 6.22 | 13.94 |
| T+E SB | dist | FR SL | 2.69 | 5.68 | 27.28 | 3.0 | 4.59 | 11.11 |
| T+E SB | dist | IT | 0.22 | 0.24 | 0.79 | 5.43 | 5.74 | 11.44 |
| T+E SB | dist | IT SL | 1.71 | 4.54 | 16.24 | 1.91 | 3.38 | 6.83 |
| Dim | dist | | 0.71 | 0.62 | 2.42 | 1.64 | 1.4 | 2.95 |

For the ML Lexical Retrieval model a main language must be chosen, indicated with German, French and Italian. Dataset adaptions are indicated with: (S) stopword removal, (SL) using only single language links, (DE/FR/IT) using only queries in one language. Table 8 shows the results of the IR task on a subset of 100 queries and with only relevant documents while Table 9 shows more detailed results using all queries.

## G.2  COURT VIEW GENERATION

Table 10 shows the results of the CVG task from both, the main and the origin dataset. Table 12 presents the CVG task results split by language, detailing scores for German, French, and Italian.

## G.3  LEADING DECISION SUMMARIZATION (LDS)

Table 13 shows all results of the LDS task. Table 14 presents the LDS task results split by language, detailing scores for German, French, and Italian.

Table 10: Results of Court View Generation task. 'In Len' denotes input length in tokens. **Bold**: best within model; underlined: best overall.

| Model | In Len ↑ | Main Scores ↑ | | | | Origin Scores ↑ | | | |
|---|---|---|---|---|---|---|---|---|---|
| | | BERT | BLEU | MET | R1 / R2 / RL | BERT | BLEU | MET | R1 / R2 / RL |
| mT5$_{Large}$ | 2048 | **75.74** | **66.92** | **34.44** | **34.91** / **15.58** / **33.53** | 76.24 | 62.59 | 32.25 | 34.80 / 16.11 / 33.58 |
| mT5$_{Large}$ | 1024 | 75.56 | 66.68 | 34.02 | 34.26 / 14.72 / 32.87 | 74.99 | 58.35 | 31.06 | 33.35 / 14.80 / 32.16 |
| mT5$_{Large}$ | 512 | 75.27 | 66.12 | 33.48 | 33.61 / 14.26 / 32.21 | **76.33** | 62.08 | **32.92** | **36.61** / **18.17** / **34.84** |
| mT5$_{Base}$ | 2048 | 75.01 | 65.48 | 32.89 | 33.23 / 13.57 / 31.89 | 75.99 | **63.39** | **34.15** | 36.48 / **18.81** / 35.58 |
| mT5$_{Base}$ | 1024 | **75.15** | **65.73** | **33.15** | **33.49** / **13.96** / **32.18** | 76.07 | 60.99 | 33.50 | **37.68** / 18.79 / **36.58** |
| mT5$_{Base}$ | 512 | 74.89 | 65.55 | 32.66 | 32.66 / 13.16 / 31.35 | **76.08** | 62.21 | 32.80 | 36.40 / 17.58 / 34.98 |
| mT5$_{Small}$ | 2048 | **74.13** | **63.97** | **30.96** | **31.29** / **11.01** / **29.90** | 75.23 | 56.59 | 30.71 | 34.68 / 13.64 / 33.24 |
| mT5$_{Small}$ | 1024 | 74.00 | 63.70 | 30.68 | 31.05 / 10.77 / 29.64 | **75.75** | 58.99 | 31.17 | 34.62 / 14.25 / **33.91** |
| mT5$_{Small}$ | 512 | 73.92 | 63.83 | 30.57 | 30.58 / 10.35 / 29.20 | 75.63 | **61.12** | **32.33** | **35.16** / **14.45** / 33.72 |

Table 11: Results of Court View Generation task. 'In Len' denotes input length in tokens. **Bold**: best within setup; underlined: best overall. (*) These models were fine-tuned on only 1'000 samples for 3 epochs. All models, except for the mT5 models, were evaluated on the validation set.

| Model | Setup | In Len ↑ | BERT ↑ | BLEU ↑ | MET ↑ | R1 / R2 / RL ↑ |
|---|---|---|---|---|---|---|
| mT5$_{Large}$ | Fine-tuned | 2048 | **75.74** | **66.92** | **34.44** | **34.91** / 15.58 / **33.53** |
| mT5$_{Large}$ | Fine-tuned | 1024 | 75.56 | 66.68 | 34.02 | 34.26 / 14.72 / 32.87 |
| mT5$_{Large}$ | Fine-tuned | 512 | 75.27 | 66.12 | 33.48 | 33.61 / 14.26 / 32.21 |
| mT5$_{Base}$ | Fine-tuned | 2048 | 75.01 | 65.48 | 32.89 | 33.23 / 13.57 / 31.89 |
| mT5$_{Base}$ | Fine-tuned | 1024 | 75.15 | 65.73 | 33.15 | 33.49 / 13.96 / 32.18 |
| mT5$_{Base}$ | Fine-tuned | 512 | 74.89 | 65.55 | 32.66 | 32.66 / 13.16 / 31.35 |
| mT5$_{Small}$ | Fine-tuned | 2048 | 74.13 | 63.97 | 30.96 | 31.29 / 11.01 / 29.90 |
| mT5$_{Small}$ | Fine-tuned | 1024 | 74.00 | 63.70 | 30.68 | 31.05 / 10.77 / 29.64 |
| mT5$_{Small}$ | Fine-tuned | 512 | 73.92 | 63.83 | 30.57 | 30.58 / 10.35 / 29.20 |
| GPT-3.5-Turbo | Fine-tuned* | 2048 | 72.31 | 62.23 | 28.08 | 26.06 / 7.19 / 24.54 |
| LLaMA-2-13B Chat | Fine-tuned* | 2048 | **74.22** | **63.51** | **33.33** | **34.36** / **16.68** / **33.20** |
| LLaMA-2-7B Chat | Fine-tuned* | 2048 | 73.27 | 62.34 | 31.6 | 32.31 / 14.47 / 31.38 |
| GPT-4 | 1-shot | 2048 | 70.39 | 59.69 | 24.63 | 23.87 / 4.64 / 22.32 |
| GPT-3.5-Turbo-16K | 1-shot | 8192 | **70.86** | 59.89 | **25.63** | 24.97 / 5.44 / 23.50 |
| GPT-3.5-Turbo-16K | 1-shot | 2048 | 70.73 | 59.92 | 25.55 | 24.95 / 5.43 / 23.49 |
| Claude 2 | 1-shot | 8192 | 69.71 | 60.92 | 24.88 | 24.92 / **5.55** / **23.61** |
| Claude 2 | 1-shot | 2048 | 69.45 | **61.26** | 24.85 | **24.98** / 5.39 / **23.61** |
| Claude Instant | 1-shot | 8192 | 67.5 | 58.18 | 23 | 23.91 / 4.32 / 22.58 |
| Claude Instant | 1-shot | 2048 | 67.49 | 57.76 | 22.84 | 23.78 / 4.23 / 22.43 |
| GPT-4 | 0-shot | 2048 | 69.41 | 58.16 | 23.25 | 22.61 / 3.95 / 21.10 |
| GPT-3.5-Turbo-16K | 0-shot | 8192 | 67.93 | 56.87 | 22.62 | 21.21 / 3.56 / 19.88 |
| GPT-3.5-Turbo-16K | 0-shot | 2048 | 67.80 | 56.52 | 22.32 | 20.99 / 3.46 / 19.74 |
| Claude 2 | 0-shot | 8192 | **70.22** | **62.38** | **25.74** | **25.66** / **6.06** / **24.27** |
| Claude 2 | 0-shot | 2048 | 69.46 | 61.36 | 24.90 | 24.66 / 5.30 / 23.24 |
| Claude Instant | 0-shot | 8192 | 66.94 | 59.02 | 22.95 | 23.16 / 4.04 / 21.82 |
| Claude Instant | 0-shot | 2048 | 66.50 | 58.78 | 22.65 | 22.65 / 3.80 / 21.32 |
| LLaMA-2-70B Chat | 0-shot | 2048 | 66.78 | 53.04 | 19.13 | 18.48 / 3.21 / 17.35 |
| LLaMA-2-13B Chat | 0-shot | 2048 | 67.23 | 55.01 | 20.18 | 19.76 / 3.26 / 18.57 |
| LLaMA-2-7B Chat | 0-shot | 2048 | 63.74 | 40.62 | 11.29 | 10.75 / 1.62 / 10.13 |

## G.4 TEXT CLASSIFICATION

Table 15 shows more detailed results on the text classification datasets including standard deviations across seeds.

### G.4.1 LANGUAGE SPECIFIC RESULTS

Table 16 shows more detailed results on the text classification datasets language specific scores.

SwissBERT, where pretraining tokens were most focused towards the dominant language German seems to have quite even results, with scores in Italian even being the highest. Models trained on CC100 (MiniLM, mDeBERTa, XLM-R and X-MOD) showed mixed results. For all models, German

Table 12: Results of the CVG task split by language. Scores are presented in the order: German, French, and Italian.

| Model | Setup | BERT ↑ | BLEU ↑ | MET ↑ | R1 ↑ | R2 ↑ | RL ↑ |
|---|---|---|---|---|---|---|---|
| GPT-3.5-Turbo | Fine-tuned | 71.89 / 73.01 / 71.17 | 62.29 / 62.15 / 62.29 | 29.02 / 27.43 / 25.38 | 25.68 / 26.97 / 23.48 | 7.31 / 7.49 / 4.64 | 24.60 / 24.94 / 21.77 |
| LLaMA-2-13B-Chat | Fine-tuned | 75.06 / 73.76 / 71.03 | 66.24 / 61.05 / 58.81 | 36.44 / 30.69 / 26.95 | 36.22 / 33.37 / 27.12 | 19.51 / 14.55 / 9.44 | 35.35 / 31.92 / 25.63 |
| GPT-4 | 1-shot | 69.73 / 71.31 / 69.22 | 58.83 / 60.89 / 58.23 | 24.20 / 25.48 / 21.76 | 22.40 / 25.86 / 21.85 | 3.56 / 6.15 / 2.88 | 21.26 / 23.82 / 20.35 |
| Claude-2 | 1-shot | 68.47 / 71.46 / 67.40 | 60.98 / 60.79 / 61.48 | 24.90 / 25.13 / 22.67 | 23.56 / 26.65 / 23.88 | 4.61 / 6.79 / 4.58 | 22.65 / 24.88 / 22.46 |
| GPT-4 | 0-shot | 69.12 / 69.87 / 68.54 | 57.90 / 58.58 / 57.18 | 23.25 / 23.42 / 21.83 | 21.48 / 24.14 / 21.08 | 2.97 / 5.26 / 2.78 | 20.25 / 22.32 / 19.44 |
| Claude-2 | 0-shot | 67.75 / 71.68 / 67.75 | 60.82 / 61.77 / 61.39 | 24.59 / 25.45 / 22.57 | 23.23 / 27.29 / 23.13 | 4.29 / 6.86 / 3.99 | 22.26 / 25.46 / 21.47 |
| LLaMA-2-13B-Chat | 0-shot | 66.72 / 67.94 / 66.65 | 53.20 / 56.73 / 57.61 | 19.86 / 20.60 / 19.98 | 18.22 / 21.50 / 20.44 | 2.18 / 4.59 / 3.03 | 17.39 / 19.92 / 18.95 |

Table 13: Results of Leading Decision Summarization (LDS) task. 'In Len' denotes input length in tokens. **Bold**: best within setup; underlined: best overall. All models, except for the mT5 models, were evaluated on the validation set.

| Model | Setup | In Len ↑ | BERT ↑ | BLEU ↑ | MET ↑ | R1 / R2 / RL ↑ |
|---|---|---|---|---|---|---|
| mT5$_{Large}$ | Fine-tuned | 4096 | *OOM* | *OOM* | *OOM* | *OOM* |
| mT5$_{Large}$ | Fine-tuned | 2048 | 73.10 | 27.21 | 21.88 | 31.47 / 12.22 / 29.94 |
| mT5$_{Large}$ | Fine-tuned | 512 | 70.67 | 26.89 | 18.31 | 24.76 / 6.15 / 23.48 |
| mT5$_{Base}$ | Fine-tuned | 4096 | **73.33** | **30.81** | **23.50** | **32.43** / **12.78** / **30.87** |
| mT5$_{Base}$ | Fine-tuned | 2048 | 72.45 | 30.13 | 21.94 | 30.09 / 10.79 / 28.71 |
| mT5$_{Base}$ | Fine-tuned | 512 | 70.60 | 27.10 | 18.31 | 24.72 / 6.15 / 23.55 |
| mT5$_{Small}$ | Fine-tuned | 4096 | 72.04 | 28.68 | 21.29 | 29.61 / 10.31 / 28.12 |
| mT5$_{Small}$ | Fine-tuned | 2048 | 71.38 | 24.64 | 19.28 | 27.88 / 9.19 / 26.54 |
| mT5$_{Small}$ | Fine-tuned | 512 | 69.66 | 20.73 | 15.95 | 22.91 / 5.36 / 21.85 |
| GPT-4 | 1-shot | 4096 | 73.55 | 47.75 | 34.72 | 30.82 / 9.68 / 28.89 |
| GPT-3.5-Turbo-16K | 1-shot | 8192 | 72.92 | 46.15 | 33.68 | 29.69 / 9.47 / 27.97 |
| GPT-3.5-Turbo-16K | 1-shot | 4096 | 72.89 | 45.21 | 32.76 | 29.69 / 9.25 / 27.94 |
| Claude 2 | 1-shot | 16384 | 73.21 | **49.79** | **35.58** | **31.19** / **9.76** / **29.44** |
| Claude 2 | 1-shot | 4096 | 72.91 | 47.55 | 33.57 | 30.28 / 9.12 / 28.58 |
| Claude Instant | 1-shot | 16384 | 72.23 | 46.29 | 31.05 | 28.26 / 8.78 / 26.42 |
| Claude Instant | 1-shot | 4096 | 72.44 | 44.80 | 30.29 | 27.89 / 8.56 / 26.18 |
| GPT-4 | 0-shot | 4096 | **71.56** | 48.35 | 32.97 | 26.52 / **8.93** / 24.51 |
| GPT-3.5-Turbo-16K | 0-shot | 4096 | 70.28 | 46.08 | 30.60 | 25.18 / 7.58 / 23.59 |
| Claude 2 | 0-shot | 16384 | 71.45 | **49.29** | **33.51** | **28.61** / 8.84 / **26.59** |
| Claude 2 | 0-shot | 4096 | 71.13 | 49.20 | 32.54 | 27.70 / 8.39 / 25.90 |
| Claude Instant | 0-shot | 16384 | 70.64 | 45.19 | 28.92 | 26.23 / 7.68 / 24.28 |
| Claude Instant | 0-shot | 4096 | 71.33 | 45.65 | 29.22 | 26.13 / 8.16 / 24.15 |

performance was very close to French performance. MiniLM, mDeBERTa, and X-MOD showed Italian underperformance whereas XLM-R showed very strong performance in Italian, especially the large variant. Even though the Legal-ch-R models are based on XLM-R, they show underperformance in Italian, but similar performance between French and German. mT5 models performed well in French, the base variant additionally also performed well on Italian. BLOOM was much better in French than in other languages, not surprising given it did not have German and Italian in the pretraining data.

Overall, there only seems to be a weak trend connecting higher percentage of a given language in the pretraining corpus leading to better downstream results in that language.

Table 14: Results of the LDS task split by language. Scores are presented in the order: German, French, and Italian.

| Model | Setup | BERT ↑ | BLEU ↑ | MET ↑ | R1 ↑ | R2 ↑ | RL ↑ |
|---|---|---|---|---|---|---|---|
| GPT-4 | 1-shot | 73.89 / 72.84 / 74.08 | 49.32 / 45.85 / 38.08 | 36.50 / 32.01 / 28.75 | 31.14 / 30.03 / 32.43 | 9.91 / 9.13 / 10.82 | 29.25 / 27.99 / 30.84 |
| GPT-3.5 | 1-shot | 73.26 / 72.21 / 72.50 | 47.44 / 41.50 / 39.08 | 35.23 / 28.48 / 27.37 | 30.64 / 28.22 / 26.11 | 9.46 / 8.87 / 8.96 | 28.90 / 26.46 / 24.34 |
| Claude-2 | 1-shot | 73.17 / 72.34 / 73.69 | 48.62 / 45.50 / 47.68 | 35.07 / 30.69 / 33.91 | 30.13 / 30.30 / 32.57 | 9.29 / 8.52 / 11.50 | 28.39 / 28.71 / 30.86 |
| Claude-Instant | 1-shot | 72.97 / 71.51 / 71.44 | 45.66 / 43.40 / 42.06 | 32.02 / 27.20 / 27.40 | 28.40 / 27.17 / 25.58 | 8.80 / 8.12 / 8.28 | 26.71 / 25.36 / 24.20 |
| GPT-4 | 0-shot | 72.65 / 69.56 / 70.95 | 49.17 / 46.96 / 46.89 | 36.26 / 27.39 / 27.10 | 28.27 / 23.45 / 24.25 | 10.88 / 5.36 / 7.74 | 26.26 / 21.42 / 22.51 |
| GPT-3.5 | 0-shot | 71.25 / 68.39 / 69.84 | 47.39 / 43.68 / 44.17 | 33.31 / 25.51 / 27.91 | 27.11 / 21.73 / 21.86 | 9.22 / 4.64 / 4.75 | 25.47 / 20.22 / 20.48 |
| Claude-2 | 0-shot | 71.74 / 69.81 / 72.77 | 50.21 / 46.94 / 52.35 | 34.72 / 28.09 / 35.60 | 28.93 / 25.02 / 31.04 | 9.27 / 6.38 / 11.55 | 27.11 / 23.22 / 29.45 |
| Claude-Instant | 0-shot | 71.99 / 70.08 / 70.81 | 46.82 / 43.59 / 43.33 | 31.41 / 25.06 / 27.55 | 27.43 / 23.60 / 25.63 | 9.17 / 6.11 / 8.48 | 25.48 / 21.57 / 23.63 |

Table 15: Configuration aggregate scores with standard deviations on the test set. The macro-F1 scores are provided.

| Model | CPB-F | CPB-C | CPC-F | CPC-C | SLAP-F | SLAP-C | JP-F | JP-C | Agg. |
|---|---|---|---|---|---|---|---|---|---|
| MiniLM | $54.7_{+/-1.9}$ | $65.8_{+/-1.6}$ | $9.8_{+/-2.8}$ | $20.8_{+/-3.0}$ | $59.7_{+/-3.8}$ | $61.1_{+/-3.7}$ | $58.1_{+/-0.4}$ | $78.5_{+/-2.3}$ | 32.4 |
| DistilmBERT | $56.2_{+/-0.5}$ | $65.4_{+/-1.7}$ | $19.6_{+/-1.1}$ | $22.1_{+/-0.4}$ | $63.7_{+/-11.7}$ | $65.9_{+/-6.4}$ | $59.9_{+/-0.9}$ | $75.5_{+/-3.3}$ | 42.1 |
| mDeBERTa-v3 | $55.1_{+/-2.0}$ | $69.8_{+/-2.8}$ | $21.0_{+/-3.6}$ | $17.5_{+/-4.4}$ | $63.8_{+/-6.3}$ | $59.3_{+/-7.6}$ | $60.6_{+/-0.9}$ | $77.9_{+/-2.6}$ | 40.2 |
| XLM-R$_{Base}$ | $57.2_{+/-1.5}$ | $65.9_{+/-3.2}$ | $21.3_{+/-1.5}$ | $23.7_{+/-1.9}$ | $67.2_{+/-15.9}$ | $73.4_{+/-2.5}$ | $60.9_{+/-0.6}$ | $79.7_{+/-2.5}$ | 44.6 |
| XLM-R$_{Large}$ | $56.4_{+/-1.8}$ | $67.9_{+/-1.9}$ | $24.4_{+/-7.2}$ | $29.1_{+/-2.7}$ | $65.1_{+/-8.5}$ | $78.9_{+/-4.6}$ | $60.8_{+/-0.6}$ | $80.9_{+/-2.4}$ | 48.6 |
| X-MOD$_{Base}$ | $56.6_{+/-1.8}$ | $67.8_{+/-2.9}$ | $20.0_{+/-3.0}$ | $20.6_{+/-3.5}$ | $63.9_{+/-10.1}$ | $64.4_{+/-7.0}$ | $60.5_{+/-0.6}$ | $79.1_{+/-2.6}$ | 41.9 |
| SwissBERT$_{(xlm-vocab)}$ | $56.9_{+/-0.7}$ | $67.3_{+/-4.7}$ | $25.7_{+/-8.3}$ | $23.0_{+/-4.0}$ | $61.5_{+/-9.5}$ | $73.2_{+/-2.1}$ | $61.4_{+/-0.6}$ | $79.4_{+/-2.5}$ | 46.1 |
| mT5$_{Small}$ | $52.2_{+/-1.9}$ | $62.1_{+/-5.2}$ | $13.2_{+/-2.4}$ | $17.9_{+/-1.7}$ | $53.1_{+/-13.8}$ | $60.9_{+/-15.9}$ | $58.9_{+/-1.0}$ | $74.2_{+/-3.6}$ | 34.4 |
| mT5$_{Base}$ | $52.1_{+/-1.6}$ | $61.5_{+/-3.9}$ | $14.0_{+/-2.8}$ | $19.7_{+/-1.6}$ | $58.4_{+/-17.2}$ | $61.8_{+/-16.8}$ | $54.5_{+/-1.5}$ | $72.0_{+/-3.1}$ | 35.9 |
| BLOOM-560m | $53.0_{+/-1.7}$ | $61.7_{+/-4.1}$ | $10.7_{+/-3.7}$ | $8.0_{+/-3.5}$ | $52.6_{+/-10.7}$ | $53.2_{+/-8.5}$ | $60.5_{+/-0.7}$ | $73.4_{+/-7.2}$ | 24.9 |
| Legal-ch-R$_{Base}$ | $57.7_{+/-1.6}$ | $70.5_{+/-2.3}$ | $16.2_{+/-5.8}$ | $20.1_{+/-5.6}$ | $77.0_{+/-3.6}$ | $79.7_{+/-0.9}$ | $64.0_{+/-1.3}$ | $86.4_{+/-1.9}$ | 40.9 |
| Legal-ch-R$_{Large}$ | $55.9_{+/-2.2}$ | $68.9_{+/-2.1}$ | $25.8_{+/-7.8}$ | $16.3_{+/-8.7}$ | $76.9_{+/-2.3}$ | $84.9_{+/-9.7}$ | $62.8_{+/-0.9}$ | $87.1_{+/-2.2}$ | 43.3 |
| Legal-ch-LF$_{Base}$ | $58.1_{+/-2.1}$ | $70.8_{+/-2.9}$ | $21.4_{+/-2.9}$ | $17.4_{+/-8.6}$ | $80.1_{+/-12.7}$ | $77.1_{+/-4.8}$ | $65.4_{+/-1.7}$ | $86.4_{+/-1.8}$ | 42.5 |

Table 16: Configuration aggregate scores. The macro-F1 scores from the language-specific subsets of the test set are provided.

| Model Languages | CPB-F de / fr / it | CPB-C de / fr / it | CPC-F de / fr / it | CPC-C de / fr / it | SLAP-F de / fr / it | SLAP-C de / fr / it | JP-F de / fr / it | JP-C de / fr / it | Agg. de / fr / it |
|---|---|---|---|---|---|---|---|---|---|
| MiniLM | 57.5 / 53.9 / 52.9 | 68.1 / 65.4 / 64.2 | 12.1 / 13.1 / 6.8 | 24.6 / 21.9 / 17.3 | 55.3 / 60.0 / 64.5 | 57.6 / 60.0 / 66.5 | 57.7 / 58.1 / 58.7 | 77.8 / 81.7 / 76.1 | 36.1 / 36.6 / 26.6 |
| DistilmBERT | 56.3 / 55.6 / 56.8 | 67.8 / 63.9 / 64.7 | 20.2 / 18.2 / 20.7 | 22.6 / 21.6 / 22.2 | 50.9 / 67.2 / 79.6 | 57.8 / 68.8 / 72.9 | 60.5 / 60.7 / 58.6 | 75.6 / 79.8 / 71.7 | 41.4 / 41.3 / 43.6 |
| mDeBERTa-v3 | 57.6 / 55.1 / 52.7 | 73.9 / 68.1 / 67.7 | 25.4 / 22.8 / 16.8 | 22.1 / 21.6 / 12.6 | 59.7 / 60.1 / 73.3 | 59.4 / 69.9 / 51.3 | 59.5 / 61.8 / 60.4 | 78.8 / 80.7 / 74.5 | 44.8 / 43.8 / 33.9 |
| XLM-R$_{Base}$ | 59.4 / 56.3 / 56.0 | 70.2 / 65.4 / 62.5 | 20.0 / 20.6 / 23.5 | 26.5 / 22.1 / 23.1 | 54.5 / 64.6 / 92.2 | 71.5 / 71.9 / 77.1 | 60.9 / 61.6 / 60.2 | 79.9 / 82.8 / 76.7 | 44.5 / 43.3 / 46.2 |
| XLM-R$_{Large}$ | 58.4 / 56.8 / 54.1 | 70.5 / 67.3 / 66.0 | 22.5 / 19.7 / 36.2 | 26.7 / 28.2 / 33.0 | 65.5 / 56.1 / 77.0 | 73.7 / 78.8 / 84.9 | 60.8 / 61.6 / 60.1 | 81.3 / 83.7 / 77.9 | 46.9 / 45.1 / 54.9 |
| X-MOD$_{Base}$ | 59.0 / 56.2 / 54.8 | 71.1 / 68.7 / 64.1 | 19.8 / 17.2 / 24.4 | 23.2 / 24.2 / 16.4 | 55.7 / 61.1 / 79.3 | 63.1 / 74.5 / 57.8 | 60.2 / 61.3 / 60.0 | 79.4 / 82.4 / 76.0 | 42.6 / 42.1 / 40.9 |
| SwissBERT$_{(xlm-vocab)}$ | 57.6 / 55.9 / 57.3 | 72.4 / 69.3 / 61.2 | 23.8 / 20.3 / 39.4 | 28.5 / 24.0 / 18.7 | 50.0 / 66.8 / 72.4 | 71.2 / 72.4 / 76.1 | 61.1 / 62.2 / 60.9 | 79.8 / 82.5 / 76.3 | 46.7 / 44.4 / 47.3 |
| mT5$_{Small}$ | 54.8 / 51.7 / 50.3 | 69.2 / 61.9 / 56.4 | 14.2 / 16.2 / 10.5 | 15.9 / 18.1 / 20.2 | 37.6 / 67.6 / 66.2 | 51.7 / 86.6 / 54.4 | 59.8 / 59.5 / 57.5 | 75.9 / 77.7 / 69.4 | 33.1 / 38.3 / 32.3 |
| mT5$_{Base}$ | 54.1 / 52.1 / 50.3 | 66.4 / 61.9 / 56.8 | 10.6 / 16.3 / 16.9 | 18.7 / 18.7 / 22.1 | 40.4 / 80.8 / 70.6 | 47.2 / 87.9 / 62.7 | 56.2 / 55.7 / 52.6 | 73.4 / 75.3 / 67.9 | 31.0 / 39.0 / 38.9 |
| BLOOM-560m | 55.1 / 53.2 / 50.9 | 64.6 / 65.3 / 56.2 | 12.6 / 16.1 / 7.1 | 9.5 / 13.6 / 5.1 | 39.9 / 61.8 / 63.2 | 42.7 / 61.8 / 59.6 | 59.8 / 61.5 / 60.3 | 68.8 / 84.2 / 69.1 | 26.8 / 34.8 / 18.4 |
| Legal-ch-R$_{Base}$ | 59.3 / 58.4 / 55.5 | 73.8 / 69.4 / 68.6 | 24.3 / 20.5 / 10.5 | 26.2 / 25.3 / 14.0 | 79.8 / 72.1 / 79.6 | 80.8 / 79.6 / 78.6 | 62.5 / 65.8 / 63.9 | 87.6 / 87.9 / 83.7 | 49.4 / 46.3 / 31.7 |
| Legal-ch-R$_{Large}$ | 58.3 / 55.7 / 53.6 | 71.9 / 68.5 / 66.7 | 23.0 / 21.3 / 38.7 | 28.5 / 26.0 / 9.0 | 74.0 / 77.4 / 79.6 | 77.6 / 80.5 / 99.5 | 61.6 / 63.9 / 63.0 | 88.6 / 88.7 / 84.1 | 49.0 / 47.0 / 36.2 |
| Legal-ch-LF$_{Base}$ | 60.7 / 58.3 / 55.5 | 74.8 / 70.0 / 67.8 | 25.3 / 21.5 / 18.4 | 29.2 / 26.7 / 9.9 | 75.9 / 70.3 / 99.7 | 82.2 / 72.6 / 100 | 63.1 / 67.1 / 66.0 | 87.5 / 87.9 / 84.0 | 51.2 / 47.2 / 31.0 |

Table 17: Configuration aggregate scores on the validation set. The macro-F1 scores are provided. The highest values are in bold. It is important to note that the scores presented here are calculated as the harmonic mean over multiple seeds.

| Model | CPB-F | CPB-C | CPC-F | CPC-C | SLAP-F | SLAP-C | JP-F | JP-C | Agg. |
|---|---|---|---|---|---|---|---|---|---|
| MiniLM | 59.1 | 71.0 | 14.9 | 36.9 | 73.8 | 78.9 | 60.9 | 81.4 | 44.4 |
| DistilmBERT | 59.6 | 70.1 | 26.3 | 35.8 | 74.1 | 90.3 | 60.8 | 78.8 | 53.1 |
| mDeBERTa-v3 | 60.1 | 73.0 | 30.4 | 36.0 | 77.4 | 82.0 | 63.3 | 81.1 | 55.5 |
| XLM-R$_{Base}$ | 60.1 | 70.5 | 26.9 | 38.5 | 78.7 | 92.2 | 62.9 | 82.5 | 55.0 |
| XLM-R$_{Large}$ | 60.5 | 71.7 | 27.2 | 39.7 | 74.0 | 96.2 | 63.2 | 83.1 | 55.5 |
| X-MOD$_{Base}$ | 57.1 | 71.0 | 27.0 | 33.4 | 81.3 | 94.4 | 62.5 | 82.1 | 53.5 |
| SwissBERT$_{(xlm-vocab)}$ | 59.0 | 72.1 | 29.4 | 38.8 | 85.6 | 95.5 | 62.6 | 82.3 | 56.8 |
| mT5$_{Small}$ | 54.8 | 66.1 | 26.3 | 32.5 | 84.7 | 88.0 | 60.1 | 77.3 | 51.6 |
| mT5$_{Base}$ | 55.7 | 64.4 | 24.3 | 29.3 | 83.0 | 82.6 | 47.7 | 66.1 | 47.3 |
| BLOOM-560m | 52.2 | 64.3 | 20.1 | 21.8 | 78.5 | 82.4 | 60.5 | 76.7 | 43.3 |
| Legal-ch-R$_{Base}$ | 61.2 | **73.6** | 27.7 | 41.0 | 99.0 | 96.1 | 65.3 | 88.8 | 58.2 |
| Legal-ch-R$_{Large}$ | **61.8** | 73.5 | 29.8 | 32.0 | **99.3** | **98.8** | 65.1 | **89.7** | 56.6 |
| Legal-ch-LF$_{Base}$ | 59.4 | 72.7 | **32.2** | **42.5** | 99.1 | 98.1 | **67.0** | 89.1 | **60.8** |

# H    DETAILED DATA DESCRIPTION

In this section, we provide additional information about the datasets. Table 18 provides additional information about general dataset metadata.

Table 18: Listing of cantons, courts, chambers, law-areas

| Metadata | Number | Examples |
|---|---|---|
| Cantons | 26 (+1) | Aargau (AG), Bern (BE), Basel-Stadt (BS), Solothurn (SO), Ticino (Ti), Vaud (VD),... (+ Federation (CH)) |
| Courts | 184 | Cantonal Bar Supervisory Authority, Supreme Court, administrative authorities, Tax Appeals Commission, Cantonal Court, Federal Administrative Court, ... |
| Chambers | 456 | GR-UPL0-01, AG-VB-002, CH-BGer-011, ZH-OG-001, ZG-VG-004, VS-BZG-009, VD-TC-002, TI-TE-001, ... |
| Law-Areas | 4 | Civil, Criminal, Public, Social |
| Languages | 5 | German, French, Italian, Romansh, English |

## H.1    PRE-TRAINING

### H.1.1    RULINGS AND LEGISLATION

Figures 6 and 8 provide an overview of the distribution of languages and cantons in the rulings dataset respectively. Figure 4 shows the length distribution of the cases.

Figures 7 and 9 provide an overview of the distribution of languages and cantons in the legislation dataset respectively. Figure 5 shows the length distribution of the legislation texts

## H.2    LEADING DECISIONS

Figures 10 and 11 show the length distributions for the facts and considerations of the Leading Decisions dataset.

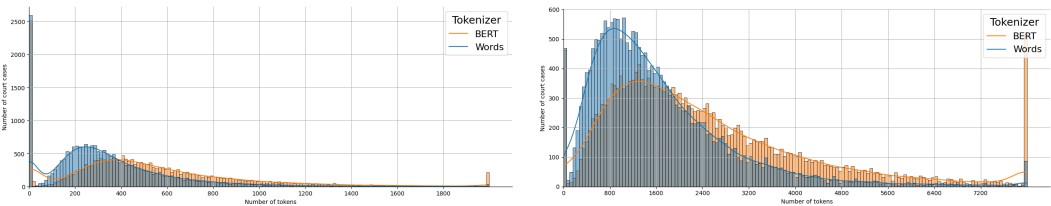

Figure 10: Leading Decisions facts length distribution          Figure 11: Leading Decisions considerations length distribution

## H.3    LAW AREA PREDICTION

Figure 12 shows the length distribution for the facts of the LAP dataset.

## H.4    CRITICALITY PREDICTION

Figures 13 and 14 show the length distributions for the facts and the considerations of the acCP dataset respectively.

## H.5    JUDGMENT PREDICTION

Figure 15 shows the length distribution for the facts of the JP dataset.

## H.6    COURT VIEW GENERATION

Figures 16 and 17 show the length distributions for the facts and the considerations of the CVG dataset respectively.

### H.7 LEADING DECISION SUMMARIZATION

Figures 18 and 19 show the length distributions for the input text and the summary of the LDS dataset respectively

### H.8 INFORMATION RETRIEVAL

Figure 20 shows the length distribution for the facts of the IR dataset. Figure 3 shows the structure of the corpus, queries and qrels for the IR task.

### H.9 CITATION EXTRACTION

Table 26 shows an illustration of the Citation Extraction (CE) task.

Table 19: Illustration of the pre-training corpora

| **Motivation: Pre-training Corpora** |
| --- |
| A large corpus of high quality domain specific text is crucial for training LLMs capable of performing tasks in a given domain. This dataset collects a large part of publicly available legal text relevant for Switzerland. |

**Legislation text:**

Der Grosse Rat des Kantons Aargau, gestützt auf die §§ 72 Abs. 3 und 78 Abs. 1 der Kantonsverfassung, beschliesst:
1. Allgemeine Bestimmungen
§ Gegenstand und Zweck
1 Dieses Gesetz regelt a) die amtliche Information der Öffentlichkeit und den Zugang zu amtlichen Dokumenten [...]
§ 15 Bekanntgabe an Private
Öffentliche Organe geben Privaten Personendaten nur bekannt, wenn
a) sie dazu gesetzlich verpflichtet sind, oder
b) die Bekanntgabe nötig ist, um eine gesetzliche Aufgabe erfüllen zu können [...]

**Metadata:**

UUID: *58450ad4-108d-4e10-b559-a7efece689d7*
Year: *2015*, Language: *German*, Canton: *AG*
Title: *Gesetz über die Information der Öffentlichkeit, den Datenschutz und das Archivwesen*
Abbreviation: *IDAG*, SR Number: *150.700*

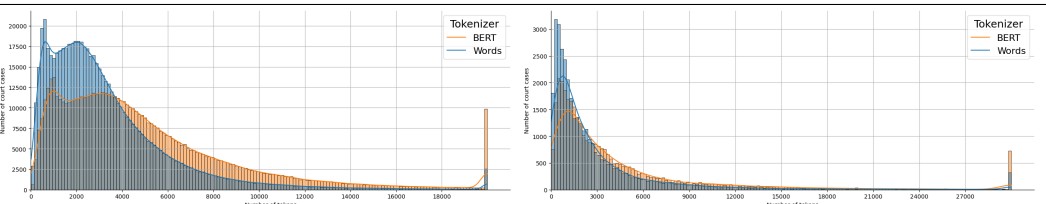

Figure 4: Rulings text length distribution          Figure 5: Legislation text length distribution

Figure 6: Language distribution of rulings texts          Figure 7: Language distribution of legislation texts

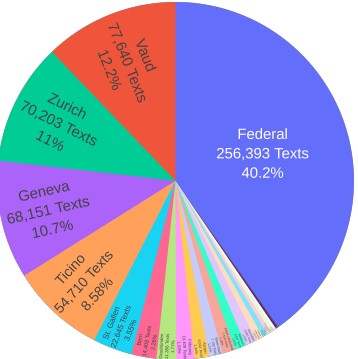
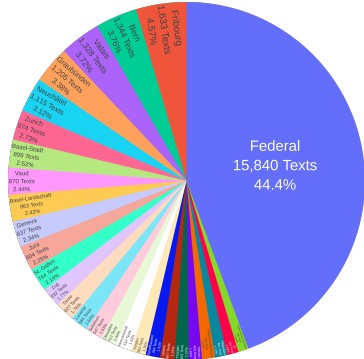

Figure 8: Cantonal distribution of rulings texts          Figure 9: Cantonal distribution of legislation texts

Table 20: Illustration of the Sub Law Area Prediction (SLAP) task

| **Motivation: Sub Law Area Prediction (SLAP)** |
| --- |
| Before the judge even sees a complaint, it is first handled by the court's administrative staff, deciding to which chamber (suborganisation inside the court hearing matters in a specific subpart of the law) the complaints should be routed. For this task, models trained on a dataset like ours could assist by providing a suggestion. |

| **Input** | **Target** |
| --- | --- |
| [Facts]:
I. Faits 1. En date du 11 novembre 2013, l'intimé a déposé à la Commune de Corcelles une demande de permis de construire pour la pose d'un revêtement bitumineux sur l'accès à son immeuble, le prolongement d'un chemin existant et l'installation d'une piscine sur les parcelles n° C._ et D._ du registre foncier de la commune de Corcelles. Les parcelles se situent en zone agricole. Le recourant a formé opposition contre ce projet de construction. Dans sa décision globale du 1er décembre 2014, la Préfecture du Jura bernois a accepté la demande d'octroi du permis de construire.
2. Le 31 décembre 2014, le recourant a déposé un recours contre cette décision auprès de la Direction des travaux publics, des transports et de l'énergie du canton de Berne (TTE). Il fait valoir, en substance, que différentes conditions de la décision globale du 1er décembre 2014 n'auraient pas été respectées. Il fait également valoir que l'intimé aurait réalisé sur son immeuble différents travaux sans autorisation.
3. L'Office juridique, qui dirige les procédures de recours pour la TTE1, a requis le dossier préliminaire et dirigé l'échange des mémoires. Les prises de position de l'intimé, de l'instance précédente et de la commune de Corcelles ont été envoyées le 15 janvier 2015, le 4 février 2015 et le 6 février 2015. Le recourant a déposé deux autres prises de position, le 9 janvier 2015 et le 11 mars 2015. Dans la mesure où cela est important pour la décision, il sera fait référence aux mémoires dans les considérants ci-dessous. L'intimé a vendu son immeuble entre-temps. La présente décision est envoyée au nouveau propriétaire pour information. | Urban Planning and Environmental |

**Possible Labels:**

Tax, Urban Planning and Environmental, Expropriation, Public Administration, Other Fiscal, Rental and Lease, Employment Contract, Bankruptcy, Family, Competition and Antitrust, Intellectual Property, Substantive Criminal, Criminal Procedure

**Metadata:**

Decision ID: *519d0350-6e0e-5551-9bc9-1df033382168*
Year: *2015*, Language: *French*, Law Area: *Public*, Law Sub Area: *Urban Planning and Environmental*
Court: *BE_VB*, Chamber: *BE_VB_001*, Canton: *BE*, Region: *Espace Mittelland*

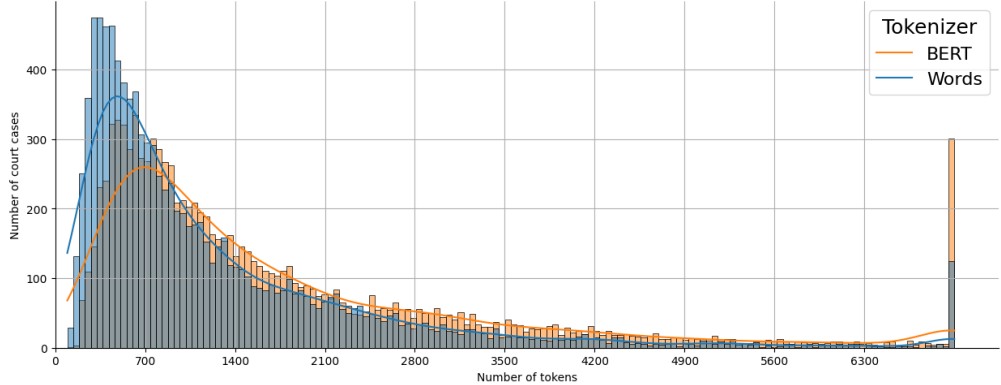

Figure 12: Law Area Prediction facts length distribution

Table 21: Illustration of the Criticality Prediction (CP) task

| **Motivation: Criticality Prediction (CP)** |
|---|
| We see two potential applications of the Criticality Prediction task: Prioritization and Classification. The prioritization task takes as input the facts as a proxy for a complaint and produces a prioritization score judging how critical/important this case is. This prioritization might help decide which cases should be heard earlier or by more experienced judges. In a futuristic scenario where automatic judgment prediction is accepted, cases with low priority could be sent to automated solutions, while high priority cases would be sent to human judges. The classification task takes as input the considerations or the entire ruling and performs a post-hoc analysis comparing it to prior caselaw and judging its potential impact on future jurisprudence. It can be used to designate certain seminal cases that are likely most influenal and for future cases. |

| Input | Target |
|---|---|
| [Consideraions]: Erwägungen: 1. Angefochten ist der in einem kantonal letztinstanzlichen Scheidungsurteil festgesetzte nacheheliche Unterhalt in einem Fr. 30'000.– übersteigenden Umfang; auf die Beschwerde ist somit einzutreten (Art. 72 Abs. 1, Art. 74 Abs. 1 lit. b, Art. 75 Abs. 1 und Art. 90 BGG). 2. Die Parteien pflegten eine klassische Rollenteilung, bei der die Ehefrau die Kinder grosszog und sich um den Haushalt kümmerte. Infolge der Trennung nahm sie im November 2005 wieder eine Arbeitstätigkeit auf und erzielt mit einem 80%-Pensum Fr. 2'955.– netto pro Monat. Beide kantonalen Instanzen haben ihr jedoch auf der Basis einer Vollzeitstelle ein hypothetisches Einkommen von Fr. 3'690.– angerechnet. Das Obergericht hat zwar festgehalten, der Ehefrau sei eine Ausdehnung der Arbeitstätigkeit kaum möglich, gleichzeitig aber erwogen, es sei nicht ersichtlich, weshalb sie nicht einer Vollzeitbeschäftigung nachgehen könne. Ungeachtet dieses Widerspruches wird das Einkommen von Fr. 3'690.– von der Ehefrau ausdrücklich anerkannt, weshalb den nachfolgenden rechtlichen Ausführungen dieser Betrag zugrunde zu legen ist. Der Ehemann verdient unbestrittenermassen Fr. 5'334.– netto pro Monat. [...] | LD label: critical Citation label: critical-1 |

**Possible Labels:**

LD label: critical, non-critical
Citation label: critical-1, critical-2, critical-3, critical-4

**Metadata:**

Decision ID: *65aad3f6-33c2-4de2-91c7-436e8143d6ea*
Year: *2007*, Language: *German*, Law Area: *Civil*, LD Label: *Critical*, Citation Label: *Citation-1*
Court: *CH_BGer*, Chamber: *CH_BGer_005*, Canton: *CH*, Region: *Federation*

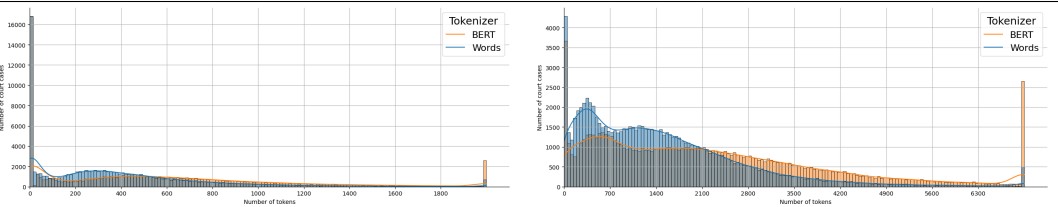

Figure 13: Criticality Prediction facts length distribution    Figure 14: Criticality Prediction considerations length distribution

Table 22: Illustration of the Judgment Prediction (JP) task

---

### Motivation: Judgment Prediction (JP)

Judgment Prediction might be used in the future in jurisdictions that are experiencing extremely high case loads such as US immigration. The legal maxim "justice delayed is justice denied" may provide motivation for judgment prediction being applied in such highly overloaded jurisdictions by giving affected people the opportunity to have their case heard much earlier (in some jurisdictions wait times are years). For example, consider a scenario where a person is held in pretrial detention awaiting their case to be heard. With Judgment Prediction, it may be possible to identify cases where there is a high likelihood of the individual being not guilty. These cases can then be prioritized for judicial review, potentially reducing the time innocent individuals spend in detention due to system backlogs.

---

| Input | Target |
|---|---|
| [Facts]:
En fait : A. Le 5 février 2015 à 21 h 30, à [...],A.H._ a été appréhendé par la police, qui l'a entendu le lendemain vers 0 h 45 comme prévenu notamment d'infraction à la LStup (Loi fédérale sur les stupéfiants ; RS 812.121). L'intéressé a déclaré qu'alors qu'il se trouvait dans un bar à [...], un homme s'était assis à côté de lui et lui avait demandé de la cocaïne. Le prévenu s'était rendu dans l'appartement occupé notamment par B.H._, un compatriote qui l'hébergeait à l'occasion, pour prendre une boulette de cocaïne, qu'il avait ensuite vendue à l'inconnu pour 100 francs. Les policiers lui avaient finalement indiqué que le client en question était en réalité un agent de police en civil. Sur la base d'indications fournies par A.H._ au client lors de cette transaction, l'appartement occupé par B.H._, rue de [...] à [...], avait fait l'objet, la veille vers 22 h 45, d'une perquisition qui avait amené la découverte de 29.8 g de cocaïne et de plusieurs téléphones portables.
Le 6 février 2015 à 13 h 50, le Procureur cantonal Strada a procédé à l'audition d'arrestation de A.H._, lequel a confirmé ses déclarations à la police, en particulier la vente d'une boulette de cocaïne la veille au soir. L'intéressé a également indiqué qu'il avait aidé B.H._ à confectionner des boulettes de cocaïne trois jours plus tôt, qu'il consommait de la cocaïne depuis décembre 2014 et qu'il n'avait pas le droit de demeurer en Suisse, où il était revenu le 23 janvier 2015.
B. Par ordonnance du 7 février 2015, le Tribunal des mesures de contrainte, faisant droit à la requête du Ministère public, a ordonné, en raison du risque de fuite et du risque de collusion, la détention provisoire de A.H._ pour une durée maximale de trois mois, soit au plus tard jusqu'au 5 mai 2015.
C. Par acte du 17 février 2015, A.H._ a interjeté recours devant la Chambre des recours pénale contre cette ordonnance, en concluant, avec suite de frais et de dépens, à sa réforme principalement en ce sens que la demande de détention provisoire soit refusée et la mise en liberté provisoire ordonnée, et à ce que la procédure dirigée contre lui soit classée. Il n'a pas été ordonné d'échanges d'écritures. | Dismissal |

---

### Possible Labels:

Approval, Dismissal

### Metadata:

Decision ID: *0dd2f9f7-872e-4200-9f9c-f1c12520c267*
Year: *2015*, Language: *French*, Law Area: *Penal*, Judgment: *Dismissal*
Court: *VD_TC*, Chamber: *VD_TC_013,* Canton: *VD*, Region: *Région lémanique*

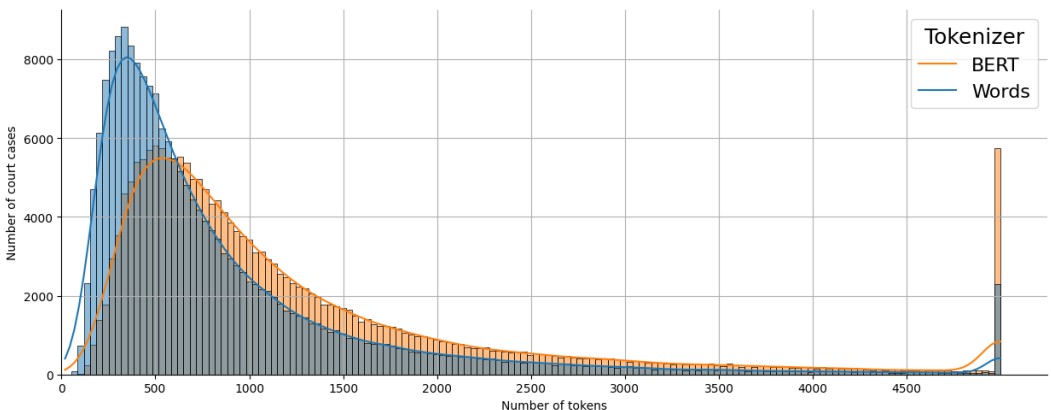

Figure 15: Judgment Prediction facts length distribution

Table 23: Illustration of the Court View Generation (CVG) task

---

**Motivation: Court View Generation (CVG)**

---

Court view generation is arguably one of the most difficult NLP tasks. It requires extensive legal reasoning capabilities, significant experience, and good knowledge of the specific law area a judge is operating in. Machines may assist judges and clerks by suggesting word or sentence continuations while they are typing or even by setting up complete drafts. To solve this task well, the following ingredients are likely necessary: First, a strong retrieval system capable of providing the necessary legal context based on legislation and influential previous rulings. Second, a strong legal reasoning system capable of analyzing the facts, any lower court decisions, and the retrieved documents. This matter is complicated further in our dataset due to lengthy documents in multiple languages.

---

| **Input** | **Target** |
|---|---|
| [Facts]:

Zum Sachverhalt: 1. Am Dienstag, 23. August 2005, um 12.16 Uhr, lenkte X seinen Personenwagen von Bronschhofen her kommend auf der Hauptstrasse in Richtung Wil. Auf der Höhe Hauptstrasse 64 wurde er von der Kantonspolizei St. Gallen anlässlich einer Geschwindigkeitskontrolle innerorts mit einer Geschwindigkeit von 80 km/h gemessen. Nach Abzug der technisch bedingten Sicherheitsmarge von 5 km/h resultierte eine rechtlich relevante Geschwindigkeit von 75 km/h.
2. Mit Strafbescheid des Untersuchungsamtes Gossau wurde der Angeklagte am 10. Mai 2006 wegen grober Verletzung der Verkehrsregeln zu einer Busse von Fr. 610.00 verurteilt. Dagegen erhob er Einsprache. Der Einzelrichter des Kreisgerichtes Alttoggenburg-Wil verurteilte ihn mit Urteil vom 14. September 2006 wegen grober Verkehrsregelverletzung und fällte eine Busse von Fr. 600.00 aus. Für die Löschung im Strafregister wurde eine Probezeit von zwei Jahren angesetzt, die Kosten des Verfahrens wurden dem Angeklagten auferlegt.
3. Dagegen erklärte der Verteidiger fristgerecht Berufung. Er verlangte einen Freispruch von der groben Verkehrsregelverletzung (Art. 90 Ziff. 2 SVG) [...] Die Staatsanwaltschaft trug auf Abweisung der Berufung an. | [Considerations]:

Aus den Erwägungen: 1. Nach Art. 90 Ziff. 2 SVG wird mit Freiheitsstrafe bis zu drei Jahren oder Geldstrafe bestraft, wer [...] Der subjektive Tatbestand der groben Verkehrsregelverletzung ist hier deshalb regelmässig zu bejahen. Eine Ausnahme kommt etwa da in Betracht, wo [...]
2. Der Angeklagte bringt vor, die Vorinstanz habe den Grundsatz in dubio pro reo verletzt, wenn [...] Indem der Angeklagte innerorts mit mindestens 25 km/h zu schnell gefahren ist, hat er den objektiven Tatbestand der groben Verkehrsregelverletzung erfüllt. [...] Aus dem gleichen Grund ist auch der Beweisantrag zur Vornahme eines Augenscheins abzuweisen. III.
1. Der Angeklagte hat eine grobe Verkehrsregelverletzung begangen. Sein Verschulden wiegt schon deshalb nicht mehr leicht, weil [...], so erscheint eine Geldstrafe von 4 Tagessätzen angemessen (Art. 34 i.V.m. Art. 47 StGB). [...] Die Voraussetzungen für den bedingten Strafvollzug sind fraglos erfüllt (Art. 42 StGB). [...]
2. Der Vollzug der Geldstrafe wird unter Ansetzung einer Probezeit von zwei Jahren bedingt aufgeschoben. Bewährt sich der Angeklagte während der Probezeit nicht, so muss die Prognose seines künftigen Legalverhaltens neu gestellt werden und der Angeklagte müsste mit dem Vollzug der Geldstrafe rechnen. [...] |

---

**Possible Labels:**

Text

---

**Metadata:**

Decision ID: *0f86bb1e-ed24-52a1-bec7-e04451485a7f*
Year: *2007*, Language: *German*, Law Area: *Penal*
Court: *SG_KG*, Chamber: *SG_KG_001*, Canton: *SG*, Region: *Eastern Switzerland*

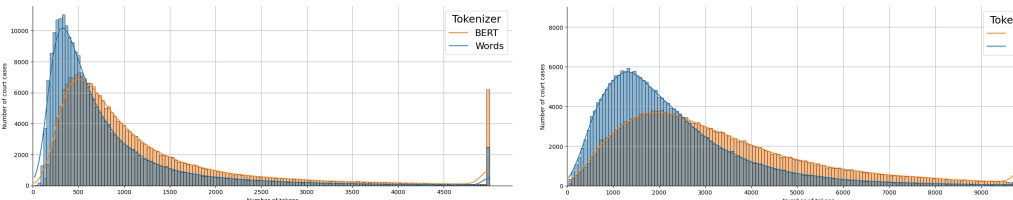

Figure 16: Court View Generation facts length distribution          Figure 17: Court View Generation considerations length distribution

Table 24: Illustration of the Leading Decision Summarization (LDS) task

| Motivation: Leading Decision Summarization (LDS) |
|---|

Summarizing cases is very important for lawyers to absorb the most relevant information in less time. Lawyers need to read many cases during their research. Reducing the time needed to comprehend the gist of a case brings direct economic value.

| Input | Target |
|---|---|
| [Case Text]:
BGE 141 IV 201 S. 201
Dai considerandi:
8.
8.2.1
È stato accertato, senza arbitrio, che la ricorrente ha più volte chiesto a F. di trovare, nel senso di contattare e ingaggiare (avendo precisato che aveva i soldi per pagare), qualcuno che potesse uccidere il marito e che egli rifiutò di fare quello che gli si domandava. 8.2.2
Contrariamente a quanto sostenuto nel gravame, la contestata richiesta risulta tutt'altro che generica: permetteva di ben comprendere sia il genere di infrazione finale prospettata (reato contro la vita) sia la vittima designata sia il comportamento da assumere, ossia reperire e ingaggiare qualcuno allo scopo, atteso che vi era a disposizione denaro. F. non si è risolto a commettere alcunché, motivo per cui si è di fronte solo a un tentativo di istigazione e la questione del nesso causale tra l'atto di persuasione e la decisione dell'istigato di commettere il reato neppure si pone.
[...]
È piuttosto nell'ambito della commisurazione della pena che occorre considerare la gravità reale del tentativo di istigazione, le conseguenze concrete dell'atto commesso e la prossimità del risultato (v. sentenza 6S.44/2007 del 6 giugno 2007 consid. 4.5.5). Nella fattispecie la Corte cantonale ha effettivamente considerato tali aspetti al momento di commisurare la pena. Sicché su questo punto la condanna della ricorrente non viola l' art. 24 cpv. 2 CP ed è conforme al diritto federale. | [Regeste]:
Regeste
Art. 24 Abs. 2 StGB; indirekte Anstiftung (Kettenanstiftung), Versuch.
Auch die versuchte indirekte Anstiftung (Kettenanstiftung) zu einem Verbrechen ist strafbar (E. 8.2.2). |

**Possible Labels:**

Text

**Metadata:**

Decision ID: *91ae0d9f-9aec-4b2b-a7ee-042abc42adaa*
Year: *2015*, Language: *Italian*
Court: *CH_BGE*, Chamber: *CH_BGE_006*, Canton: *CH*, Region: *Federation*

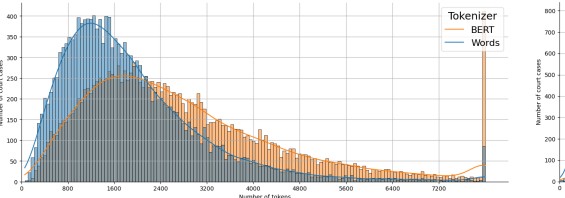
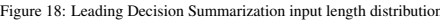

Figure 18: Leading Decision Summarization input length distribution

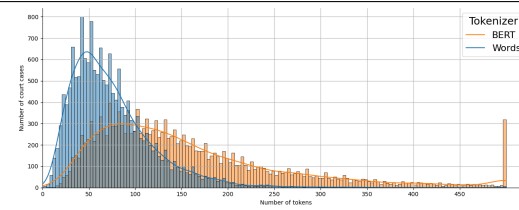

Figure 19: Leading Decision Summarization summary length distribution

Table 25: Illustration of the Multilingual Information Retrieval (MLIR) task

| **Motivation: Multilingual Information Retrieval (MLIR)** |
|---|
| Information retrieval is at the heart of the daily work of lawyers. Much like in scientific writing, lawyers base their arguments on prior caselaw, relevant legislation, and legal analyses. Thus, they spend a large part of their work searching for these documents, motivating the importance of legal IR. Annotating these data at scale is very costly, which is why this dataset is based on the citation graph of Swiss Supreme Court cases. In Switzerland lawyers operate in a trilingual jurisdiction with legislation and caselaw appearing in up to three official languages German, French and Italian. This means that for a complaint written in German, a case written in French might be relevant, leading to Multilingual IR, further complicating the task. |

| Input | Target |
|---|---|
| [Facts]: Fatti: A. Il 9 novembre 2006 G._, nata P._ (1959), ha contratto matrimonio con F._. Dall'unione non sono nati figli. Per contro il marito ha avuto figli (ormai adulti) dal primo matrimonio i quali non hanno però vissuto in economia domestica con G._. Il 26 settembre 2011 è deceduto F._. Con domanda del 4 ottobre 2011 G._ ha chiesto alla cassa di compensazione Medisuisse l'erogazione di una rendita vedovile. Con decisione del 27 ottobre 2011, sostanzialmente confermata il 22 dicembre successivo in seguito all'opposizione dell'interessata, la cassa di compensazione ha respinto la richiesta di prestazione per il motivo che la richiedente non era stata sposata almeno cinque anni con il defunto marito, come invece prescritto dalla legge, bensì "solo" 4 anni 10 mesi e 18 giorni. B. Osservando che il termine di cinque anni non era adempiuto per soli pochi giorni e invocando di conseguenza una applicazione della legge secondo un giudizio di giustizia ed equità, G._ si è aggravata al Tribunale delle assicurazioni del Cantone Ticino e ha chiesto il riconoscimento della rendita. Statuendo per giudice unico, la Corte cantonale ha respinto il ricorso per pronuncia del 29 febbraio 2012. C. L'interessata ha presentato ricorso al Tribunale federale al quale ribadisce la richiesta di prima sede. Dei motivi si dirà, per quanto occorra, nei considerandi. Non sono state chieste osservazioni al gravame. | Laws: 75488867-c001-4eb9-93b9-04264ea91f55 e10ed709-8b11-47e3-8006-88b26d86e498 e6b06567-1236-4210-adb3-e11c26e497d5 2ef9b20e-bb7c-491f-9391-59ac4f74e3c9 b8d4aeef-a8ef-40d9-92a1-090a37538008 1af9b596-92d7-4f80-a38b-876ed88ccfe5 53be6a03-1fd8-4980-aa5c-bd81e9a54d5e 4b5a2135-fee2-4e3b-811e-15ce1c71bddf 6ab38922-6309-4021-83cc-56d776d7a332 

Cited Rulings: 54df6482-97d7-47eb-afb1-1ccb9369cb89 921a799a-9077-4057-8e46-4919fd4f3263 |

| **Possible Labels:** |
|---|
| 10K laws and rulings (see Section 4.5 for more information) |

| **Metadata:** |
|---|
| Decision ID: *6856ac58-5d12-48c4-acef-831d50c79886* 
 Year: *2012*, Language: *Italian* Law Area: *Social* 
 Court: *BE_VB*, Chamber: *CH_BGer_009*, Canton: *BE*, Region: *Federation* |

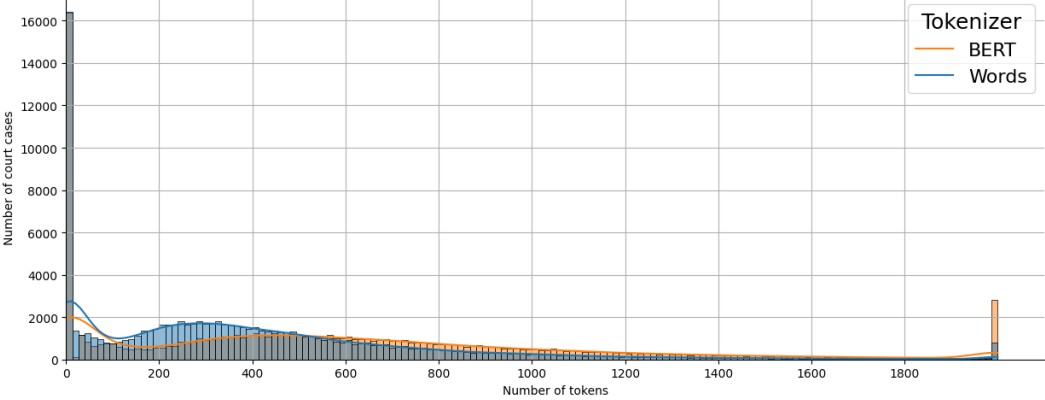

Figure 20: Information Retrieval facts length distribution

Table 26: Illustration of the Citation Extraction (CE) task

**Motivation: Citation Extraction (CE)**

Citation extraction is an important preprocessing step to collect information from legal documents. It enables easy semantic linking to relevant legislation, caselaw and analyses. Due to extensive rulebooks, simple regexes are often insufficient for accurate extraction of legal citations, motivating the need for more complex approaches.

| Input | Target |
|---|---|
| Considerations: ['ergangen', 'ist', 'und', 'sich', 'das', 'Verfahren', 'daher', 'noch', 'nach', 'dem', 'Bundesgesetz', 'über', 'die', 'Organisation', 'der', 'Bundesrechtspflege', '(', 'OG', ')', 'vom', '16', '.', 'Dezember', '1943', 'richtet', '(', 'vgl', '.', 'Art', '.', '132', 'Abs', '.', '1', 'BGG', ';', 'BGE', '132', 'V', '393', 'E', '.', '1', '.', '2', 'S', '.', '395', ')', 'dass', 'das', 'Verfahren', 'nicht', 'die', 'Bewilligung', 'oder', 'Verweigerung', 'von', 'Versicherungsleistungen', 'zum', 'Gegenstand', 'hat', 'und', 'deshalb', 'gemäss', 'Art', '.', '134', 'Satz', '1', 'OG', '[', 'in', 'der', 'von', '1'] | [0, 0, 0, 0, 0, 0, 0, 0, 0, 0, 0, 0, 0, 0, 0, 0, 0, 0, 0, 0, 0, 0, 0, 0, 0, 0, 0, 0, 3, 4, 4, 0, 0, 0, 0, 0, 1, 2, 2, 2, 0, 0, 0, 0, 0, 0, 0, 0, 0, 0, 0, 0, 0, 0, 0, 0, 0, 0, 0, 0, 0, 0, 0, 0, 0, 0, 0, 0, 0, 0, 0, 0, 0, 3, 4, 4, 0, 0, 0, 0, 0, 0, 0, 0] |

**Possible Labels:**

0: O
1: B-CITATION
2: I-CITATION
3: B-LAW
4: I-LAW

**Metadata:**

Decision ID: *1572342e-a20d-4137-9593-47fc43b98af3*
Year: *2007*, Language: *German*, Law Area: *Social*
Court: *CH_BGer*, Chamber: *CH_BGer_009*, Canton: *CH*, Region: *Federation*

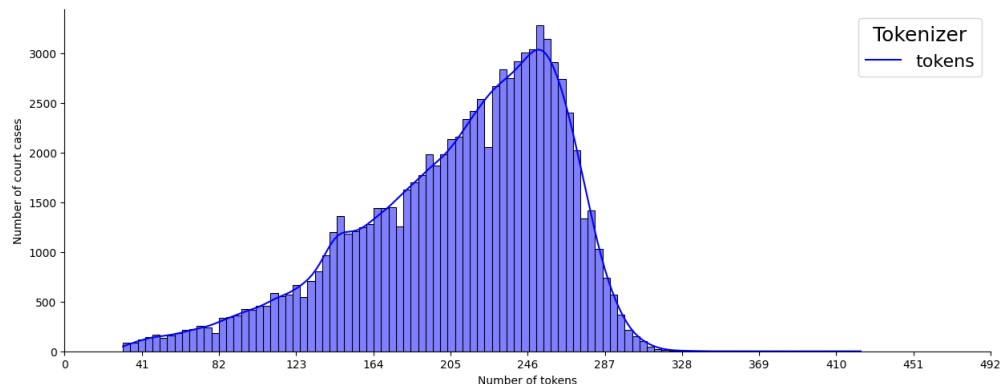

Figure 21: Citation Extraction considerations length distribution

# I ERROR ANALYSIS

## I.1 COURT VIEW GENERATION (CVG)

### I.1.1 GERMAN

Table 27 shows a comparison of generated text in the German language from GPT-3.5-Turbo-16K, GPT-4, Claude Instant and Claude 2. Generally, specific textual constructs, such as those related to asylum applications, appear frequently in a similar manner so fine-tuned models are able to proficiently predict these patterns, whereas zero-shot models face challenges. *LLaMA-2-13B-Chat (0-shot)* switches to English after a few sentences, and the German segments contain linguistic and grammatical errors. A possible reason could be that LLaMA-2 was predominantly trained in English rather than other languages. It also made an unsupported claim regarding the appellant's age, which wasn't mentioned in the input. Zero-shot models tend to center more on the primary content, while fine-tuned models are tailored to predict formalities, mirroring the target. The fine-tuned *LLaMA-2-13B-Chat* references 'BFM' instead of the 'SEM' (State Secretariat for Migration), deviating from both *GPT-3.5*'s outputs and the original input. To note, SEM emerged in 2004 from a merger between the BFM (Federal Office for Refugees) and the IMES (Federal Office for Immigration, Integration and Emigration), nevertheless our sample was from 2016. As outlined in Section 5.3, constraints in model availability and computational resources led to truncation of the target output during fine-tuning and experimentation, thus accentuating formal aspects like court jurisdiction and the legitimacy of appeals. This causes zero-shot iterations to receive lower scores, even when they remain contextually accurate. In terms of content, both zero-shot models were on the right track, tending more towards rejection. Furthermore, zero-shot predictions provide clearer insights by minimizing emphasis on formal nuances.

### I.1.2 FRENCH

Table 28 shows a comparison of generated text in the French language from GPT-3.5-Turbo-16K, GPT-4, Claude Instant and Claude 2. Remarkably, the LLaMA zero-shot model doesn't revert to English when handling the French text, unlike its behavior in the German scenario.

## I.2 LEADING DECISION SUMMARIZATION (LDS)

Table 29 shows a comparison of generated text from GPT-3.5-Turbo-16K, GPT-4, Claude Instant and Claude 2.

In the side-by-side analysis, all models demonstrated an authentic stylistic representation. GPT-3.5-Turbo 16K provides a more narrative-style summary, while the other models stick more closely to a classic 'Regeste' format. However, when it came to the factual accuracy, GPT-3.5-Turbo-16K faltered by negating a crucial element (*"wurde entschieden, dass 'kein' hypothetisches Einkommen.."*). Claude-2 cited an outdated legal provision, Article 137 of the ZGB, which has been inactive since 2011, even though the case decision was from 2017 and no such reference was present in the input. Despite this, the article cited remains relevant to the context. On the other hand, GPT-4's reference to BGE 137 III 118 and GPT-3.5's mention of BGE 143 III 233 S. 235 were accurate and existed in the provided input.

Table 27: Comparison of the generated considerations (CVG task) from four LLMs in different settings. The scores of each example is stated below in the following order: BERTScore, BLEU-Score, METEOR, ROUGE-1 / ROUGE-2 / ROUGE-L. The first row contains the facts of the case which were given as input, the second row contains the the considerations, the target of this task.

## Generated Considerations - German

Sachverhalt: A. Der Beschwerdeführer ersuchte am 14. August 2015 in der Schweiz um Asyl. Am 27. August 2015 wurde er zur Person befragt (BzP) und am 10. September 2015 fand – im Beisein einer Vertrauensperson – die Anhö- rung zu seinen Asylgründen statt. B. Der aus Albanien stammende Beschwerdeführer gab an, er sei am (...) geboren und somit noch minderjährig. Zur Begründung seines Asylge- suchs führte er aus, er habe eine Beziehung zu einer jungen Frau gehabt, worauf deren Vater, nachdem er von der Beziehung mit seiner Tochter er- fahren gehabt habe, von ihm verlangt habe, diese zu heiraten. Da er aber das Mädchen nicht liebe, habe er weder eine Ehe mit ihr eingehen noch eine Beziehung mit ihr weiterführen wollen. Der Vater der jungen Frau habe sich mit seinen Eltern in Verbindung gesetzt und ihnen erklärt, dass ihr Sohn seine Tochter zu heiraten habe, da er mit ihr eine Beziehung geführt habe. Seine Eltern seien damit einverstanden gewesen. Seither habe ihn sein Vater täglich ermahnt, dass er dieses Mädchen nun heiraten müsse. Weil er sich jeden Tag dasselbe von seinem Vater habe anhören müssen und er dies nicht mehr ertragen habe, habe er sich zum Verlassen seines Heimatlandes entschlossen. Bei einer Rückkehr nach Albanien hätte er mit Schwierigkeiten zu rechnen, da man ihn zur Heirat zwingen würde. Zum Beleg seiner Personalien reichte der Beschwerdeführer eine Identi- tätskarte ein. C. Mit Verfügung vom 5. Oktober 2015 – dem Beschwerdeführer und seiner Vertrauensperson eröffnet am darauffolgenden Tag – stellte die Vorinstanz fest, der Beschwerdeführer erfülle die Flüchtlingseigenschaft nicht, lehnte das Asylgesuch ab und verfügte die Wegweisung aus der Schweiz. Den zuständigen Kanton beauftragte sie mit dem Vollzug der Wegweisung. D. Mit Eingabe vom 13. Oktober 2015 (Poststempel) erhob der Beschwerde- führer gegen diesen Entscheid beim Bundesverwaltungsgericht Be- schwerde und beantragte die Aufhebung der angefochtenen Verfügung, sodann sei ihm Asyl zu gewähren, eventualiter sei die Unzumutbarkeit des Wegweisungsvollzugs festzustellen und ihm die vorläufige Aufnahme zu D-6530/2015 Seite 3 gewähren, subeventualiter sei die Sache an die Vorinstanz zurückzuwei- sen und diese anzuweisen, eine allfällige Wegweisung des Beschwerde- führers pflichtgemäss abzuklären, dies unter Würdigung seiner Minderjäh- rigkeit. In prozessualer Hinsicht beantragte er die unentgeltliche Prozess- führung im Sinne von Art. 65 Abs. 1 VwVG sowie den Verzicht auf die Er- hebung eines Kostenvorschusses.

Das Bundesverwaltungsgericht zieht in Erwägung: 1. 1.1. Gemäss Art. 31 VGG beurteilt das Bundesverwaltungsgericht Be- schwerden gegen Verfügungen nach Art. 5 VwVG. Das SEM gehört zu den Behörden nach Art. 33 VGG und ist daher eine Vorinstanz des Bundesver- waltungsgerichts. Eine das Sachgebiet betreffende Ausnahme im Sinne von Art. 32 VGG liegt nicht vor. Das Bundesverwaltungsgericht ist daher zuständig für die Beurteilung der vorliegenden Beschwerde und entschei- det auf dem Gebiet des Asyls in der Regel – und so auch vorliegend – endgültig (Art. 105 AsylG [SR 142.31]; Art. 83 Bst. d Ziff. 1 BGG). 2. Die Kognition des Bundesverwaltungsgerichts und die zulässigen Rü- gen richten sich im Asylbereich nach Art. 106 Abs. 1 AsylG und im Bereich des Ausländerrechts nach Art. 49 VwVG (vgl. BVGE 2014/26 E. 5), 2.1. Mit Beschwerde kann die Verletzung von Bundesrecht (einschliesslich Missbrauch und Überschreiten des Ermessens) sowie die unrichtige und unvollständige Feststellung des rechtserheblichen Sachverhalts gerügt werden (Art. 106 Abs. 1 AsylG). 3. 3.1. Vorab ist die Frage der Prozessfähigkeit als Sachurteilsvoraussetzung von Amtes wegen zu prüfen (FRITZ GYGI, Bundesverwaltungsrechtspflege, 2. Aufl. 1983, S. 73). Als verfahrensrechtliches Korrelat der Handlungsfä- higkeit ist sie nach den einsch

Decision ID: *26c65bde-9976-569d-920e-a3781b01536f*, Year: *2016*, Language: German, Court: *CH_BVGE*

| **GPT-3.5-Turbo** (Fine-tuned) | **LLaMA-2-13B Chat** (Fine-tuned) |
|---|---|
| Das Bundesverwaltungsgericht zieht in Erwägung: 1. 1.1 Gemäss Art. 31 VGG beurteilt das Bundesverwaltungsgericht Be- schwerden gegen Verfügungen nach Art. 5 VwVG. Das SEM gehört zu den Behörden nach Art. 33 VGG und ist daher eine Vorinstanz des Bundesver- waltungsgerichts. Eine das Sachgebiet betreffende Ausnahme im Sinne von Art. 32 VGG liegt nicht vor. Das Bundesverwaltungsgericht ist daher zuständig für die Beurteilung der vorliegenden Beschwerde, soweit diese auf dem Gebiet des Asyls (Art. 105 AsylG) zulässig (Art. 83 BGG) ist. 1.2 Beurteilt das Bundesverwaltungsgericht Beschwerden nach Art. 31 VGG, so wendet es – unter Vorbehalt der Anwendung von staatsvertra- glichen Rechten – materielles Bundesrecht an (Art. 37 Abs. 1 VGG). 1.3 Mit Beschwerde kann die Verletzung von Bundesrecht (einschliesslich die unrichtige oder unvollständige Feststellung des rechtserheblichen Sach- verhalts [Art. 97 Abs. 1 BGG] und die Unangemessenheit [Art. 105 Abs. 2 BGG]), die Rechtsverletzung nach Art. 95 BGG und die unrichtige oder un- zweckmässige Sachverhaltsfeststellung (Art. 97 Abs. 1 BGG) gerügt werden. Das Bundesverwaltungsgericht wendet nunmehr auch das Beschwerde- recht nach Art. 105 AsylG sowie die Staatsvertragsvorbehalte an, denen Rechnung zu tragen ist (Art. 37 Abs. 3 VGG). D-6530/2015 Seite 4 2. 2.1 Zur Beschwerde gegen Entscheide über Asylgesuche ist berechtigt, wer durch den angefochtenen Entscheid besonders berührt ist und ein rechtlich geschütztes Inter | [Erwägungen]: 1. 1.1 Gemäss Art. 31 VGG beurteilt das Bundesverwaltungsgericht Be- schwerden gegen Verfügungen nach Art. 5 VwVG. Das BFM gehört zu den Behörden nach Art. 33 VGG und ist daher eine Vorinstanz des Bundesver- tungsgerichts. Eine das Sachgebiet betreffende Ausnahme im Sinne von Art. 32 VGG liegt nicht vor. Das Bundesverwaltungsgericht ist daher zuständig für die Beurteilung der vorliegenden Beschwerde und entschei- det auf dem Gebiet des Asyls endgültig, ausser bei Vorliegen eines Auslie- ferungsersuchens des Staates, vor welchem die beschwerdeführende Per- son Schutz sucht (Art. 105 AsylG [SR 142.31]; Art. 83 Bst. d Ziff. 1 BGG). Eine solche Ausnahme im Sinne von Art. 83 Bst. d Ziff. 1 BGG liegt nicht vor, weshalb das Bundesverwaltungsgericht endgültig entscheidet. 1.2 Das Verfahren richtet sich nach dem VwVG, dem VGG und dem BGG, soweit das AsylG nichts anderes bestimmt (Art. 37 VGG und Art. 6 AsylG). 1.3 Die Beschwerde ist frist- und formgerecht eingereicht. Der Beschwerdeführer hat am Verfahren vor der Vorinstanz teilgenommen, ist durch die angefochtene Verfügung besonders berührt und hat ein schutzwürdiges Interesse an deren Aufhe- bung beziehungsweise Ände- rung. Er ist daher zur Einreichung der Beschwerde legitimiert (Art. 105 und Art. 108 Abs. 1 AsylG; Art. 48 Abs. 1 sowie Art. 52 Abs. 1 VwVG). Auf die Beschwerde ist einzutreten. 2. Mit Beschwerde kann die Verletzung von Bundesrecht |
| 85.15, 80.08, 50.36, 57.45/41.46/57.45 | 85.07, 78.41, 47.00, 60.56/51.46/60.56 |

| **Claude 2** (0-shot) | **LLaMA-2-13B Chat** (0-shot) |
|---|---|
| Hier meine Erwägungen zum vorliegenden Sachverhalt: 1. Prozessvoraussetzungen Die sachliche Zuständigkeit des Bundesverwaltungsgerichts ergibt sich aus Art. 31 VGG. Gemäss Art. 31 Abs. 1 VGG beurteilt das Bundesverwaltungsgericht Beschw- erden gegen Verfügungen nach Art. 5 AsylG. Die Vorinstanz hat am 5. Oktober 2015 eine Verfügung nach Art. 5 AsylG erlassen. Gegen diese Verfügung wurde frist- und formgerecht Beschwerde erhoben. Die Prozessvoraussetzungen sind damit erfüllt. 2. Materielle Beurteilung 2.1 Flüchtlingseigenschaft Gemäss Art. 3 AsylG gilt als Flüchtling eine Person, die in ihrem Heimatstaat wegen ihrer Rasse, Religion, Nationalität, Zugehörigkeit zu einer bestimmten sozialen Gruppe oder wegen ihrer politischen Anschauungen ernsthaften Nachteilen ausgesetzt ist oder begründete Furcht hat, solchen Nachteilen ausgesetzt zu werden. Der Beschwerdeführer macht geltend, er werde in seinem Heimatland Albanien zur Ehe mit einer Frau gezwungen, die er nicht heiraten möchte. Eine Zwangsheirat kann eine Verletzung der Men- schenrechte darstellen. Allerdings ist im vorliegenden Fall nicht ersichtlich, dass der Beschwerdeführer aufgrund eines in Art. 3 AsylG genannten Merkmals von Zwangsheirat bedroht wäre. Vielmehr scheint es sich um einen familiären Konflikt zwischen den Eltern des Beschwerdeführers und der Familie der jungen Frau zu handeln. Aus den Akten ergeben sich keine Hinweise, dass der Beschwerdeführer bei einer Rückkehr nach Albanien staatlichen Verfolgungshandlungen ausgesetzt wäre. Auch liegen keine Beweise dafür vor, dass er durch nichtstaat | Erwägungen Das Bundesverwaltungsgericht hat den Beschwerdeführer auf seine Er- wägungen hin gehört und bewertet die behaupteten Asylgründe. 1. Minderjährigkeit Der Beschwerdeführer gibt an, geboren zu sein und minderjährig zu sein. Die Vorin- stanz hat jedoch festgestellt, dass er nicht minderjährig ist. Es ist jedoch unumstritten, dass der Beschwerdeführer eine Beziehung zu einer jungen Frau hatte, die von ihrem Vater verlangt wurde, diese zu heiraten. Der Beschwerdeführer hat jedoch keine Ehe mit ihr eingegangen und will nicht weitere Beziehung mit ihr haben. 2. Asylgründe Der Beschwerdeführer behauptet, dass er aus Albanien geflohen ist, weil er sich dort Schwierigkeiten because of his refusal to marry the young woman would have faced. However, the evidence presented does not support this claim. There is no indication that the young woman's father would have subjected the applicant to persecution or violence if he had not married his daughter. 3. Flüchtlingseigenschaft The applicant does not meet the criteria for refugee status under the Geneva Convention. He has not demonstrated that he is fleeing persecution or that his life or freedom would be threatened if he were to return to Albania. 4. Personalien The applicant has provided an identity card as proof of his personal details. However, there is no independent evidence to support his claims. The Vorinstanz has found that the applicant does not meet the criteria for refugee status, and therefore, his asylum application should be rejected. 5. Unzumutbarkeit des Wegweisungsvollzugs The applicant has requested that the court consider the unreasonableness of the deportation order. However, the court has found that the order is reasonable and in accordance with the law. 6. Allfäl- lige Wegweisung The court has considered the possibility of a conditional release of the applicant. However, given the lack of evidence supporting the applicant's claims and the fact that he does not meet the criteria for refugee status, the court has decided not to grant conditional release. 7 |
| 66.96, 60.71, 26.59, 27.80/6.31/27.12 | 62.23, 38.56, 11.97, 14.33/0.42/13.71 |

Table 28: Comparison of the generated considerations (CVG task) from four LLMs in different settings. The scores of each example is stated below in the following order: BERTScore, BLEU-Score, METEOR, ROUGE-1 / ROUGE-2 / ROUGE-L. The first row contains the facts of the case which were given as input, the second row contains the the considerations, the target of this task.

## Generated Considerations - French

considérant en fait A. A._ et l'entreprise individuelle C._ ont conclu un contrat de travail d'une durée indéterminée, ayant pour objet l'engagement de A._ en qualité de collaboratrice administrative, à un taux d'activité de 40%, dès le 10 février 2014. Financée principalement par l'assurance chômage, respectivement par le SECO, C._ a pour but d'apporter son soutien aux jeunes en difficulté d'insertion professionnelle au moyen de cours, d'ateliers et de coaching et a notamment été mandatée par l'association D._ pour apporter son soutien aux adolescents de langue allemande. Dès le 1er mars 2014, A._ était occupée à 40% en tant que collaboratrice administrative et à 10% en tant qu'enseignante. Le contrat de travail, signé le 10 juin 2014, prévoyait un salaire mensuel brut de CHF 2'733.80 pour 16 heures d'activité administrative hebdomadaire ainsi qu'un salaire horaire brut de CHF 50.20 pour 4 heures d'enseignement par semaine. Dès le 1er septembre 2014, A._ a augmenté son taux d'enseignement à 40%, à l'essai. En août 2014, C._ a été transformée en la société B._ GmbH, dont E._ est l'associée gérante avec signature individuelle. B._ GmbH a repris tous les contrats de travail et a été inscrite au registre du commerce le 7 août 2014. B. Le 13 octobre 2014 a eu lieu une séance entre E._, A._, l'enseignante F._ et l'apprenti G._. Au cours de celle-ci, E._ a signalé que A._ était manifestement débordée par l'activité supplémentaire d'enseignement, que l'essai n'avait ainsi pas été probant, qu'il fallait donc l'arrêter et qu'il fallait engager quelqu'un pour assumer ce volet de travail. À partir du 15 octobre 2014, A._ a été mise en arrêt de travail à 100 % jusqu'au 31 octobre 2014 pour des raisons de santé. [...] En date du 30 octobre 2014, A._ a informé E._ qu'elle reprendrait son activité le 3 novembre 2014. Néanmoins, le 2 novembre 2014, A._ a été hospitalisée pour des douleurs dorsales. Elle a alors été mise en arrêt de travail à 100% du 2 au 4 novembre 2014. Le 4 novembre 2014, A._ a indiqué à son employeur qu'elle se présenterait à son travail le lendemain, mais qu'un rendez-vous chez son médecin traitant était prévu le même jour. De plus, elle a demandé à E._ de lui consacrer un peu de temps pour un entretien. Tribunal cantonal TC Page 3 de 9 E._ et A._ se sont entretenues le 5 novembre 2014. Au cours de cet entretien, A._ a déclaré son souhait de poursuivre son activité au sein de l'entreprise. [...]

en droit 1. a) La décision attaquée constitue une décision finale de première instance au sens des art. 308 al. 1 et 236 CPC. La voie de droit ouverte contre une telle décision est l'appel (art. 308 al. 1 let. a CPC), sauf si la valeur litigieuse est inférieure à CHF 10'000.- (art. 308 al. 2 CPC), auquel cas la décision ne peut faire l'objet que d'un recours (art. 319 let. a CPC). Dans le cas d'espèce, la valeur litigieuse au dernier état des conclusions est supérieure à CHF 10'000.- (cf. demande, ch. 1 et 2; réponse, ad. 1), de sorte que la voie de l'appel est ouverte. La valeur litigieuse devant la Cour est néanmoins inférieure à CHF 15'000.- (art. 74 al. 1 let. a et art. 51 al. 1 let. a LTF). Par conséquent, cet arrêt ne peut faire l'objet que d'un recours constitutionnel subsidiaire (art. 113 ss LTF). b) L'appel du 21 octobre 2016 a été déposé en temps utile, soit dans le délai légal de 30 jours à compter de la notification de la décision attaquée survenue le 21 septembre 2016 (art. 311 al. 1 CPC). Il en va de même de l'appel joint déposé le 13 décembre 2016, soit dans le délai légal de 30 jours à compter de la notification de l'appel survenue le 14 novembre 2016 (art. 312 al. 2 CPC). c) Dûment motivés et dotés de conclusions, l'appel et l'appel joint sont pour le surplus recevables en la forme. d) La cognition de la Cour est pleine et entière en fait comme en droit (art. 310 CPC). 2. Dans son appel, l'appelante reproche aux premiers juges de ne pas avoir reconnu le caractère abusif de la résiliation intervenue le 15 décembre 201

---

Decision ID: *8bb15591-be0b-5bc3-933a-e6f856a46fec*, Year: *2017*, Language: French, Court: *FR_TC*

---

| **GPT-3.5-Turbo** (Fine-tuned) | **LLaMA-2-13B Chat** (Fine-tuned) |
|---|---|
| en droit 1. En vertu de l'art. 57C al. 3 let. b de la loi du 11 avril 1991 sur l'organisation judiciaire (LOJ ; RS 173.01), le Tribunal cantonal connaît seul des causes portant sur la légalité de certaines décisions des tribunaux cantonaux. Dans le cas particulier, interjeté contre une décision finale prise par la juridiction inférieure sur la base de l'art. 64A al. 1 let. b de la loi sur la procédure civile du 19 novembre 2010 (CPC ; RS 270.11), le recours, déposé dans le délai (art. 48A al. 3 CPC) et la forme prévus par la loi (art. 79 al. 1 LOJ et 321-327 CPC), n'est en principe pas recevable, en vertu de l'art. 57C al. 2 LOJ, sauf s'il doit être traité comme un recours de droit public (cf. art. 57D LOJ). 2. Déposé dans le délai et les formes prescrits par la loi à l'encontre d'une décision finale prise par un tribunal cantonal, le recours doit être traité comme un recours de droit public, étant dès lors admissible sur le principe (art. 57C al. 4 LOJ). 3. En droit public fribourgeois, le TRPC apprécie librement la recevabilité et la wellfondation des recours de droit public cantonal, interprète et applique le droit cantonal de manière indépendant. En cas de violation du droit fédéral, le recours est, quant à lui, soumis au Tribunal fédéral. Celui-ci contrôle la violation du droit fédéral (selon les principes applicables au recours en matière de droit public, cf. à ce propos les art. 82 ss LTF), le droit constitutionnel et, le cas échéant, l'usage du pouvoir d'appréciation cantonal selon la maxime de l'arbitraire (Art. 49 al. 1 Cst. et 106 al. 2 LTF | [Considérations]: En droit, les premiers juges ont considéré qu'au vu des faits retenus, il fallait admettre le licenciement abusif. En effet, le lien de confiance qui unissait les parties avait été irrémédiablement rompu dès le 5 novembre 2014. A._ avait refusé de signer le avertissement et la lettre de résiliation de son contrat de travail, ce qui avait par conséquence qu'elle ne pouvait plus être tenue de respecter les conditions de la résiliation. Par ailleurs, le fait que la demanderesse ait été mise en arrêt de travail à 100 % du 2 au 11 novembre 2014, puis à 100 % dès le 12 novembre 2014 jusqu'au 30 novembre 2014, puis à 70 % du 1er au 14 décembre 2014, puis à 50 % du 15 décembre 2014 au 15 janvier 2015 et enfin à 25 % du 16 au 29 janvier 2015 ne pouvait être interprété comme une résiliation. En effet, la demanderesse ne pouvait pas, en tant que collaboratrice à 40%, assumer son travail à 100% dès le 2 novembre 2014, ce qu'elle aurait dû faire en tant qu'enseignante si l'on tient compte de la situation médicale de cette dernière. Par ailleurs, le fait que l'employeur ait licencié la demanderesse le 7 novembre 2014, soit durant une période de protection, ne pouvait pas être considéré comme un licenciement abusif. En effet, l'employeur n'avait pas connaissance du fait que la demanderesse était en arrêt de travail à 100 % du 2 au 11 novembre 2014. Enfin, l'employeur avait mis un terme au contrat de travail par courrier du 10 décembre 2014, soit à la fin de la période de protection, ce qui était régulier. |
| 73.33, 64.17, 28.71, 24.58/6.34/24.58 | 63.70, 57.75, 22.96, 23.57/2.20/22.14 |

| **Claude 2** (0-shot) | **LLaMA-2-13B Chat** (0-shot) |
|---|---|
| I. Conditions de procédure
1. Selon l'art. 343 al. 1 ch. 1 CPC, le tribunal établit sa compétence d'office et examine s'il a la compétence matérielle, fonctionnelle et locale pour connaître de la demande.
En l'espèce, le litige porte sur une demande en paiement pour licenciement abusif fondée sur un contrat de travail. Il s'agit donc d'un litige relevant du droit du travail, pour lequel le Tribunal des prud'hommes de l'arrondissement de la Sarine est compétent ratione materiae (art. 8 CPC). De plus, la compétence territoriale est établie à l'art. 34 CPC. La compétence fonctionnelle n'est pas contestée.
Les conditions de procédure sont ainsi remplies.
II. En droit 2. La demanderesse a été engagée par l'entreprise individuelle C._ à un taux d'activité de 40% dès le 10 février 2014. Suite à la transformation de C._ en la société B._ GmbH en août 2014, cette dernière a repris le contrat de travail. La demanderesse a augmenté son taux d'activité à 50% dès le 1er septembre 2014, à l'essai.
3. Le 15 octobre 2014, la demanderesse a été mise en arrêt maladie jusqu'au 31 octobre 2014. Le même jour, elle a adressé un courrier au Service public de l'emploi dans lequel elle mettait en cause le professionnalisme de l'intimée.
4. Le 30 octobre 2014, la demanderesse a informé l'intimée de sa reprise de travail le 3 novembre 2014. Toutefois, elle a été hospitalisée du 2 au 4 novembre 2014 et mise à nouveau en arrêt maladie. Le 5 novembre 2014, l'intimée lui a signifié un avertissement ainsi qu'une résiliation ordinaire de son | Considérant les faits allégués et les demandes des parties, le tribunal est appelé à évaluer les dispositions juridiques pertinentes et à apprécier les preuves pour déterminer si le licenciement d'A._ a été abusif.
Tout d'abord, il est important de noter que le contrat de travail entre A._ et B._ GmbH a été résilié pendant une période de protection, ce qui signifie que la résiliation était nulle et non avenue. A._ a également signalé des manquements de l'entreprise et a exprimé des critiques quant à la gestion d'E._, ce qui peut être considéré comme une raison valable pour mettre fin au contrat. Cependant, il est important de noter que ces manquements et ces critiques n'ont pas été formellement étayés par des preuves solides.
Ensuite, il est important de considérer le lien de confiance qui existe entre A._ et E._. Bien que E._ ait signalé que le lien de confiance avait été rompu en raison des critiques formulées par A._, il est possible de considérer que le lien de confiance n'a pas été rompu de manière formelle et que les rapports de travail pouvaient continuer. En outre, il est important de prendre en compte la situation de santé d'A._, qui a été hospitalisée pour des douleurs dorsales et a été mise en arrêt de travail à plusieurs reprises. Bien que les préjudices pour la santé d'A._ soient acknowledgeables, il est important de noter que ces préjudices n'ont pas été directement causés par le licenciement d'A._.
Enfin, il est important de considérer les dispositions juridiques pertinentes. Le Code des obligations suisse (CO) prévoit que le licenciement d'un employé doit être justifié et que l'employeur doit fournir des preuves solides pour étayer la résiliation. Le CO également prévoit que le licenciement abusif est a ground for dismissal.
En résumé, le tribunal doit évaluer les dispositions jur |
| 70.04, 61.20, 23.83, 24.75/2.92/24.75 | 63.92, 49.30, 13.62, 14.63/1.26/13.94 |

Table 29: Comparison of the generated regeste (LDS task) from four LLMs in different settings. The scores of each example is stated below in the following order: BERTScore, BLEU-Score, METEOR, ROUGE-1 / ROUGE-2 / ROUGE-L. Due to the size of the input, only the target is displayed.

### Generated Regeste

Regeste Abänderung des Unterhaltsbeitrages für die Dauer des Scheidungsverfahrens infolge veränderter Verhältnisse; Rechtsmissbrauch ( Art. 179 Abs. 1, Art. 2 Abs. 2 ZGB ).
Vermindert der Unterhaltspflichtige sein Einkommen in Schädigungsabsicht, ist eine Abänderung der Unterhaltsleistung selbst dann ausgeschlossen, wenn die Einkommensreduktion nicht mehr rückgängig gemacht werden kann (Änderung der Rechtsprechung; E. 3).

Decision ID: *3e70603c-bbcd-47de-a066-23124945fcc1*, Year: *2017*, Language: German, Court: *CH_BGE*

| **GPT-4** (1-shot) | **Claude 2** (1-shot) |
|---|---|
| Regeste
Abänderung von Unterhaltsbeiträgen bei Reduktion des Einkommens (BGE 137 III 118).
Bei der Bemessung des Unterhaltsbeitrages ist grundsätzlich vom tatsächlich erzielten Einkommen des Unterhaltspflichtigen auszugehen. Eine Abänderung der Unterhaltsleistung ist jedoch auszuschliessen, wenn der Unterhaltspflichtige sein Einkommen in Schädigungsabsicht vermindert hat, selbst dann wenn die Einkommensverminderung nicht mehr rückgängig gemacht werden kann (E. 3.4). | Regeste
Art. 137 ZGB (Unterhaltsabänderung). Abänderung des Ehegattenunterhalts bei veränderter Einkommenslage. Bei freiwilliger Schädigung ist eine Abänderung selbst dann ausgeschlossen, wenn die Einkommensverminderung nicht mehr rückgängig gemacht werden kann (E. 3). |
| 79.82, 68.93, 55.47, 45.16/28.04/43.01 | 83.88, 47.53, 42.44, 53.33/29.27/53.33 |
| **GPT-3.5-Turbo-16K** (0-shot) | **Claude Instant** (0-shot) |
| BGE 143 III 233 S. 235 - Ehegattenunterhalt: Abänderung aufgrund veränderter Einkommenslage des Unterhaltspflichtigen; Anrechnung eines hypothetischen Einkommens; Rechtsmissbräuchliches Verhalten des Unterhaltspflichtigen. Der unterhaltspflichtige Ehemann beantragte eine Abänderung des Ehegattenunterhalts aufgrund seiner veränderten Einkommenslage. Das Gericht stellte fest, dass eine Abänderung ausgeschlossen ist, wenn die veränderte Einkommenslage durch eigenmächtiges, widerrechtliches oder rechtsmissbräuchliches Verhalten des Unterhaltspflichtigen herbeigeführt wurde. Es wurde entschieden, dass in diesem Fall kein hypothetisches Einkommen angerechnet werden kann, selbst wenn die Einkommensverminderung nicht mehr rückgängig gemacht werden kann. | Regeste:
a) Art. 107ff ZGB, 128 III 4 E. 4
b) Ehegattenunterhalt, Abänderungsverfahren, hypothetisches Einkommen
c) Das Bundesgericht hielt fest, dass dem Unterhaltspflichtigen, der sein Einkommen in Schädigungsabsicht vermindert hat, selbst dann ein hypothetisches Einkommen anzurechnen ist, wenn die Einkommensverminderung nicht mehr rückgängig gemacht werden kann. Somit ist eine Abänderung der Unterhaltsleistung in einem solchen Fall ausgeschlossen. |
| 72.21, 43.71, 44.39, 27.52/12.60/23.85 | 76.91, 67.20, 49.20, 41.67/24.30/41.67 |

# J PROMPTS

In the following, we provide the prompts that we used for experiments with LLMs over 2B parameters.

## J.1 TEXT CLASSIFICATION

We created our prompts in a style similar to that used in Chalkidis (2023). First, we introduce the document, e.g., the court decision. Then, we provide an explanation of the labels and the task in general. Finally, we present the labels from which the model can choose. To limit the model from producing excessive output, we conclude the prompt with a cloze-style sentence, such as 'The relevant label in this case is:', which the model simply needs to complete.

### J.1.1 CRITICALITY PREDICTION

**Criticality Prediction (CP) LD Facts**
Given the facts from the following Swiss Federal Supreme Court Decision:
{INPUT FROM THE VALIDATION SET}
Federal Supreme Court Decisions in Switzerland that are published additionally get the label critical, those Federal Supreme Court Decisions that are not published additionally, get the label non-critical. Therefore, there are two labels to choose from:
- critical
- non-critical
The relevant label in this case is:

**Criticality Prediction (CP) LD Considerations**
Given the considerations from the following Swiss Federal Supreme Court Decision:
{INPUT FROM THE VALIDATION SET}
Federal Supreme Court Decisions in Switzerland that are published additionally get the label critical, those Federal Supreme Court Decisions that are not published additionally, get the label non-critical. Therefore, there are two labels to choose from:
- critical
- non-critical
The relevant label in this case is:

**Criticality Prediction (CP) Citation Facts**
Given the facts from the following Swiss Federal Supreme Court Decision:
{INPUT FROM THE VALIDATION SET}
How likely is it that this Swiss Federal Supreme Court Decision gets cited. Choose between one of the following labels (a bigger number in the label means that the court decision is more likely to be cited):
- critical-1
- critical-2
- critical-3
- critical-4
The relevant label in this case is:

**Criticality Prediction (CP) Citation Considerations**
Given the considerations from the following Swiss Federal Supreme Court Decision:
{INPUT FROM THE VALIDATION SET}
How likely is it that this Swiss Federal Supreme Court Decision gets cited. Choose between one of the following labels (a bigger number in the label means that the court decision is more likely to be cited): - critical-1
- critical-2
- critical-3
- critical-4
The relevant label in this case is:

### J.1.2 JUDGMENT PREDICTION

**Judgment Prediction (JP) Facts**
Given the facts from the following court decision:
{INPUT FROM THE VALIDATION SET}
Will this court decision get approved or dismissed? There are two labels to choose from:
- dismissal
- approval
The relevant label in this case is:

**Judgment Prediction (JP) Considerations**
Given the considerations from the following court decision:
{INPUT FROM THE VALIDATION SET}
Will this court decision get approved or dismissed? There are two labels to choose from:
- dismissal
- approval
The relevant label in this case is:

### J.1.3 (SUB) LAW AREA PREDICTION

**Law Area Prediction (LAP) Facts**
Given the facts from the following court decision:
{INPUT FROM THE VALIDATION SET}
Which topic/law area is relevant out of the following options:
- Civil
- Public
- Criminal
- Social
The relevant option is:

**Law Area Prediction (LAP) Considerations**
Given the considerations from the following court decision:
{INPUT FROM THE VALIDATION SET}
Which topic/law area is relevant out of the following options:
- Civil
- Public
- Criminal
- Social
The relevant option is:

**Sub Law Area Prediction (SLAP) Facts**
Given the facts from the following court decision:
{INPUT FROM THE VALIDATION SET}
Which topic/law area is relevant out of the following options:
- Rental and Lease
- Employment Contract
- Bankruptcy
- Family
- Competition and Antitrust
- Intellectual Property
- Substantive Criminal
- Criminal Procedure
- Tax
- Urban Planning and Environmental
- Expropriation
- Public Administration
- Other Fiscal
The relevant option is:

**Sub Law Area Prediction (SLAP) Considerations**
Given the considerations from the following court decision:

{INPUT FROM THE VALIDATION SET}
Which topic/law area is relevant out of the following options:
- Rental and Lease
- Employment Contract
- Bankruptcy
- Family
- Competition and Antitrust
- Intellectual Property
- Substantive Criminal
- Criminal Procedure
- Tax
- Urban Planning and Environmental
- Expropriation
- Public Administration
- Other Fiscal
The relevant option is:

## J.2 TEXT GENERATION

In the subsequent sections, we detail the prompts employed for the 0-shot setup. For the 1-shot experiments, we consistently appended the same example in the respective language to the 0-shot instruction.

### J.2.1 COURT VIEW GENERATION (CVG)

**Court View Generation (CVG) in German**
'Ziel: Generiere Erwägungen basierend auf dem gegebenen Sachverhalt eines Schweizer Gerichtsurteils.
Hintergrund: Ein Schweizer Gerichtsurteil besteht aus Rubrum, Sachverhalt, Erwägungen, Dispositiv (Urteilsformel) und Unterschrift. Die Erwägungen sind die rechtliche Würdigung des Geschehens durch das Gericht.
Anweisung:
-Sachverhalt Verstehen: Der gegebene Sachverhalt enthält bestrittene und unbestrittene Fakten, die Begehren der Parteien, das Beweisverfahren und die Prozessgeschichte.
-Beginne mit Prozessvoraussetzungen: Prüfe zunächst, ob die Prozessvoraussetzungen (z.B. Zuständigkeit des Gerichts) erfüllt sind. Wenn nicht strittig, reicht es aus zu bestätigen, dass die Voraussetzungen erfüllt sind.
-Rechtliche Würdigung: Eruiere relevante Rechtssätze basierend auf den behaupteten und rechtlich relevanten Tatsachen.
-Setze dich mit den rechtlichen Standpunkten der Parteien auseinander.
-Beachte die Beweislastverteilung und würdige die Beweise frei, aber berücksichtige relevante gesetzliche Beweisregeln.
-Iura novit curia: Deine rechtliche Würdigung muss nicht zwangsläufig dem rechtlichen Vorbringen der Parteien entsprechen. Berücksichtige andere mögliche Argumentationslinien.
-Zusammenfassung: Fasse am Ende deine Erwägungen, das Ergebnis Ihrer rechtlichen Würdigung, zusammen.
-Output: Der generierte Text sollte strukturiert, klar und in der Form von typischen Erwägungen eines Schweizer Gerichtsurteils sein.

{Sachverhalt des Schweizer Gerichtsurteils}:
{INPUT FROM THE VALIDATION SET}

{Erwägungen}:

**Court View Generation (CVG) in French**
But: Génère des considérations basées sur les faits donnés d'un jugement suisse.
Contexte: Un jugement suisse est composé du rubrum, des faits, des considérations, du dispositif (formule du jugement) et de la signature. Les considérations sont l'appréciation juridique des

événements par le tribunal.
Instructions:
- Comprends les faits: Les faits donnés contiennent des faits contestés et non contestés, les demandes des parties, la procédure de preuve et l'historique du procès.
- Commence par les conditions de procédure: Vérifie d'abord si les conditions de procédure (par exemple, la compétence du tribunal) sont remplies. Si cela n'est pas contesté, il suffit de confirmer que les conditions sont remplies.
- Appréciation juridique: Évalue les dispositions juridiques pertinentes basées sur les faits allégués et juridiquement pertinents.
- Confronte-toi aux points de vue juridiques des parties.
- Tiens compte de la répartition de la charge de la preuve et évalue les preuves librement, mais tiens compte des règles légales de preuve pertinentes.
- Iura novit curia: Ton appréciation juridique ne doit pas nécessairement correspondre aux arguments juridiques présentés par les parties. Considère d'autres lignes d'argumentation possibles.
- Résumé: Résume à la fin de tes considérations le résultat de ton appréciation juridique.
- Résultat: Le texte généré devrait être structuré, clair et sous la forme de considérations typiques d'un jugement suisse.

{Faits du jugement suisse}:
{INPUT FROM THE VALIDATION SET}

{Considérations}:

**Court View Generation (CVG) in Italian**
Obiettivo: Genera considerazioni basate sui fatti presentati in una sentenza svizzera.
Contesto: Una sentenza svizzera si compone di rubrum, fatti, considerazioni, dispositivo (formula della sentenza) e firma. Le considerazioni rappresentano la valutazione giuridica degli eventi da parte del tribunale.
Istruzioni:
- Comprendi i fatti: I fatti presentati includono fatti contestati e non contestati, le richieste delle parti, la procedura probatoria e la storia del processo.
- Inizia con le condizioni processuali: Verifica prima di tutto se le condizioni processuali (ad es. la competenza del tribunale) sono soddisfatte. Se non contestate, basta confermare che le condizioni sono state soddisfatte.
- Valutazione giuridica: Valuta le norme giuridiche rilevanti in base ai fatti affermati e giuridicamente rilevanti.
- Confrontati con i punti di vista giuridici delle parti.
- Tieni conto della distribuzione dell'onere della prova e valuta le prove liberamente, ma considera le regole di prova legalmente rilevanti. - Iura novit curia: La tua valutazione giuridica non deve necessariamente corrispondere alle argomentazioni giuridiche delle parti. Considera altre possibili linee di argomentazione.
- Riassunto: Riassumi alla fine delle tue considerazioni il risultato della tua valutazione giuridica.
- Risultato: Il testo generato dovrebbe essere strutturato, chiaro e nella forma di considerazioni tipiche di una sentenza svizzera."

{Fatti della sentenza svizzera}:
{INPUT FROM THE VALIDATION SET}

{Considerazioni}:

J.2.2  LEADING DECISION SUMMARIZATION (LDS)

In the LDS task, we used only one German prompt because the output (the regeste) is always in German in our dataset. Despite the input being multilingual, models tend to generate in the prompt's language, regardless of the input's language.

**Leading Decision Summarization (LDS)**
Ziel: Generiere eine Regeste basierend auf einem Schweizer Gerichtsurteils.
Hintergrund: Ein Schweizer Gerichtsurteil setzt sich aus Sachverhalt, Erwägungen und Dispositiv

zusammen. Die Regeste dient als Kurzzusammenfassung und beinhaltet Leitsätze des Urteils. Nur Leitentscheide haben eine Regeste.

Anweisung:

1. Sachverhalt: Lies und verstehe den gegebenen Sachverhalt.

2. Erwägungen: Analysiere die Erwägungen, um die Hauptargumente und Gründe zu identifizieren.

3. Dispositiv: Beachte das Dispositiv, da es das endgültige Urteil enthält.

4. Erstelle die Regeste: Die Regeste sollte aus drei sehr kurzen Teilen bestehen: a. Zitiere die wichtigsten relevanten Artikelziffern (ohne den Artikeltitel). b. Nenne kurze, relevante, deskriptive Keywords, über die Thematik des Falls. c. Formuliere einen sehr kurzen Fliesstext, der die wichtigsten Erwägungen zitiert und kurz zusammenfasst.

Output: Die Regeste sollte eine klare und strukturierte Kurzzusammenfassung des Urteils bieten, die aus zitierten Artikeln, Keywords und einem sehr kurzen Fliesstext besteht.

{Gegebener Sachverhalt, Erwägungen und Dispositiv}:
{INPUT FROM THE VALIDATION SET}

{Regeste auf Deutsch}:

## K   EXAMPLE GENERATIONS

Tables 38 and 39 show excerpts of examples produced by the best model for the CVG and LDS tasks, respectively.

Table 30: Nine examples of generated considerations by mT5$_{Large}$ with input length 2048 across three languages, showcasing high, average, and low scored outputs in CVG Task

| Target | considerations: Erwägungen: 1. 1.1 Der angefochtene Entscheid ist in Anwendung von Sozialversicherungsrecht ergangen. Die Sozialversicherungsrechtliche Abteilung des Verwaltungsgerichts beurteilt gemäss Art. 57 des Bundesgesetzes vom 6. Oktober 2000 über den Allgemeinen Teil des Sozialversicherungsrechts (ATSG; SR 830.1) i.V.m. Art. 54 Abs. 1 lit. a des kantonalen Gesetzes vom 11. Juni 2009 über die Organisation der Gerichtsbehörden und der Staatsanwaltschaft (GSOG; BSG 161.1) Beschwerden gegen solche Entscheide. Die Beschwerdeführerin ist im vorinstanzlichen Verfahren mit ihren Anträgen nicht durchgedrungen, durch den angefochtenen Entscheid berührt und hat ein schutzwürdiges Interesse an dessen Aufhebung, weshalb sie zur Beschwerde befugt ist (Art. 59 ATSG). Die örtliche Zuständigkeit ist gegeben (Art. 58 ATSG). Da auch die Bestimmungen über Frist (Art. 60 ATSG) sowie Form (Art. 61 lit. b ATSG; Art. 81 Abs. 1 i.V.m. Art. 32 des kantonalen Gesetzes vom 23. Mai 1989 Urteil des Verwaltungsgerichts des Kantons Bern vom 6. Dez. 2022, UV/21/844, Seite 4 über die Verwaltungsrechtspflege [VRPG; BSG 155.21]) eingehalten sind, ist auf die Beschwerde einzutreten. 1.2 Anfechtungsobjekt bildet der Einspracheentscheid vom 5. November 2021 (act. II 45). Streitig und zu prüfen ist, ob die Beschwerdegegnerin ihre Leistungen im Zusammenhang mit dem Ereignis vom 19. Juni 2020 zu Recht per 30. September 2020 eingestellt hat. 1.3 Die Abteilungen urteilen gewöhnlich in einer Kammer bestehend aus drei Richterinnen oder Richtern (Art. 56 Abs. 1 GSOG). 1.4 Das Gericht überprüft den angefochtenen Entscheid frei und ist an die Begehren der Parteien nicht gebunden (Art. 61 lit. c und d ATSG; Art. 80 lit. c Ziff. 1 und Art. 84 Abs. 3 VRPG). 2. 2.1 Die Zusprechung von Leistungen der obligatorischen Unfallversicherung setzt grundsätzlich das Vorliegen eines Berufsunfalles, eines Nichtberufsunfalles oder einer Berufskrankheit (Art. 6 Abs |
|---|---|
| Predicted | considerations: Erwägungen: 1. 1.1 Der angefochtene Entscheid ist in Anwendung von Sozialversicherungsrecht ergangen. Die Sozialversicherungsrechtliche Abteilung des Verwaltungsgerichts beurteilt gemäss Art. 57 des Bundesgesetzes über den Allgemeinen Teil des Sozialversicherungsrechts vom 6. Oktober 2000 (ATSG; SR 830.1) i.V.m. Art. 54 Abs. 1 lit. a des kantonalen Gesetzes über die Organisation der Gerichtsbehörden und der Staatsanwaltschaft vom 11. Juni 2009 (GSOG; BSG 161.1) Beschwerden gegen solche Entscheide. Die Beschwerdeführerin ist im vorinstanzlichen Verfahren mit ihren Anträgen nicht durchgedrungen, durch den angefochtenen Entscheid berührt und hat ein schutzwürdiges Interesse an dessen Aufhebung, weshalb sie zur Beschwerde befugt ist (Art. 59 ATSG). Die örtliche Zuständigkeit ist gegeben (Art. 69 Abs. 1 lit. a des Bundesgesetzes über die Unfallversicherung vom 20. Dezember 1946 [UVG; SR 832.20]). Da auch die Bestimmungen über Frist (Art. 60 ATSG) sowie Form (Art. 61 lit. b ATSG; Art. 81 Abs. 1 i.V.m. Art. 32 des kantonalen Gesetzes über die Verwaltungsrechtspflege vom 23. Mai 1989 [VRPG; BSG 155.21]) eingehalten sind, ist auf die Beschwerde einzutreten. 1.2 Das Gericht überprüft den angefochtenen Entscheid frei und ist an die Begehren der Parteien nicht gebunden (Art. 61 lit. c und d ATSG; Art. 80 lit. c Ziff. 1 und Art. 84 Abs. 3 VRPG). 1.3 Der Beschwerdeführer hat am vorinstanzlichen Verfahren teilgenommen, ist durch den angefochtenen Entscheid besonders berührt und hat ein schutzwürdiges Interesse an dessen Aufhebung oder Änderung. Auf die Beschwerde ist einzutreten. 2. 2.1 Die Beschwerdegegnerin hat die Bestimmungen auf den Anspruch auf Heilbehandlung und Taggeld in Form von Heilbehandlung zufolge ihrer im August 2021 (act. II 42 und 46) eingereichten medizinischen Berichte zutreffend wiedergegeben. Darauf wird verwiesen. 2.2 Nach der Rechtsprechung ist der Beschwerdeführer durch den angefochtenen Entscheid berührt und haben ein schutzwürdiges |
| Scores | BERT: 91.69, BLEU: 84.17, METEOR: 62.43, ROUGE-1/2/L: 75.49 / 65.78 / 75.49 |

| Target | considerations: EN DROIT 1. Conformément à l'art. 134 al. 1 let. a ch. 2 de la loi sur l'organisation judiciaire, du 26 septembre 2010 (LOJ - E 2 05 ) en vigueur dès le 1 er janvier 2011, la chambre des assurances sociales de la Cour de justice connaît, en instance unique, des contestations prévues à l'art. 56 de la loi fédérale sur la partie générale du droit des assurances sociales, du 6 octobre 2000 (LPGA - RS 830.1 ) relatives à la loi fédérale sur l'assurance-invalidité du 19 juin 1959 (LAI - RS 831.20 ).![endif]>![if> Sa compétence pour juger du cas d'espèce est ainsi établie. 2. À teneur de l'art. 1 al. 1 LAI, les dispositions de la LPGA s'appliquent à l'assurance-invalidité (ci-après : AI), à moins que la loi n'y déroge expressément.![endif]>![if> Toutefois, les modifications légales contenues dans la LPGA constituent, en règle générale, une version formalisée dans la loi de la jurisprudence relative aux notions correspondantes avant l'entrée en vigueur de la LPGA ; il n'en découle aucune modification du point de vue de leur contenu, de sorte que la jurisprudence développée à leur propos peut être reprise et appliquée (ATF 130 V 343 consid. 3). 3. Les modifications de la loi fédérale sur l'assurance-invalidité du 21 mars 2003 (4 ème révision), du 6 octobre 2006 (5 ème révision) et du 18 mars 2011 (révision 6a), entrées en vigueur le 1 er janvier 2004, respectivement, le 1 er janvier 2008 et le 1 er janvier 2012, entraînent la modification de nombreuses dispositions légales dans le domaine de l'assurance-invalidité. |
|---|---|
| Predicted | considerations: EN DROIT 1. Conformément à l'art. 134 al. 1 let. a ch. 4 de la loi sur l'organisation judiciaire, du 26 septembre 2010 (LOJ; RS E 2 05 ) en vigueur dès le 1 er janvier 2011, la Chambre des assurances sociales de la Cour de justice connaît, en instance unique, des contestations prévues à l'art. 56 de la loi fédérale sur la partie générale du droit des assurances sociales, du 6 octobre 2000 (LPGA; RS 830.1 ) relatives à la loi fédérale sur l'assurance-invalidité du 19 juin 1959 (LAI; RS 831.20 ).![endif]>![if> Sa compétence pour juger du cas d'espèce est ainsi établie.![endif]>![if> 2. Les dispositions de la LPGA s'appliquent à l'assurance-invalidité, à moins que la loi n'y déroge expressément.![endif]>![if> Toutefois, les modifications légales contenues dans la LPGA constituent, en règle générale, une version formalisée dans la loi de la jurisprudence relative aux notions correspondantes avant l'entrée en vigueur de la LPGA; il n'en découle aucune modification du point de vue de leur contenu, de sorte que la jurisprudence développée à leur propos peut être reprise et appliquée (ATF 130 V 230 consid. 1.1; 335 consid. 1.2; ATF 129 V 4 consid. 1.2; ATF 129 V 4 consid. 1.2; ATF 127 V 467 consid. 1, 126 V 136 consid. 4b et les références). 3. A teneur de l'art. 17 al. 1 de la loi fédérale sur la partie générale du droit des assurances sociales, du 6 octobre 2000 (LPGA; RS 830.1 ), les modifications légales contenues dans la LPGA constituent, en règle générale, une version formalisée dans la LAI. |
| Scores | BERT: 91.27, BLEU: 85.8, METEOR: 68.12, ROUGE-1/2/L: 78.29 / 66.82 / 78.29 |

| Target | considerations: in diritto In ordine 2.1. La presente vertenza non pone questioni giuridiche di principio e non è di rilevante importanza (ad esempio per la difficoltà dell'istruttoria o della valutazione delle prove). Il TCA può dunque decidere nella composizione di un Giudice unico ai sensi dell'articolo 49 cpv. 2 LOG (STF 9C_699/2014 del 31 agosto 2015; 8C_855/2010 dell'11 luglio 2011; 9C_211/2010 del 18 febbraio 2011). Nel merito 2.2. Oggetto del contendere è sapere se a ragione l'Ufficio AI ha soppresso all'assicurato il diritto ad una rendita intera con effetto dal 1. settembre 2015 (cfr. consid. 1.3). L'insorgente postula l'annullamento della decisione impugnata del 10 maggio 2017 e quindi il mantenimento del diritto alla rendita intera. 2.3. Secondo l'art. 4 cpv. 1 LAI in relazione con gli art. 7 e 8 della LPGA, con invalidità s'intende l'incapacità al guadagno presunta permanente o di rilevante durata, cagionata da un danno alla salute fisica o psichica, conseguente ad infermità congenita, malattia o infortunio. Gli elementi fondamentali dell'invalidità, secondo la surriferita definizione, sono quindi un danno alla salute fisica o psichica conseguente a infermità congenita, malattia o infortunio, e la conseguente incapacità di guadagno. Occorre quindi che il danno alla salute abbia cagionato una diminuzione della capacità di guadagno, perché il caso possa essere sottoposto all'assicurazione per l'invalidità ( Duc, L'assurance invalidité, in: Meyer (Hrsg.), Schweizerisches Bundesverwaltungsrecht, Band XIV, Soziale Sicherheit, 2a ed., Basilea/Ginevra/Monaco di Baviera 2007, pag. 1411, n. 46 ). Giusta l'art. 28 cpv. |
|---|---|
| Predicted | considerations: in diritto In ordine 2.1. La presente vertenza non pone questioni giuridiche di principio e non è di rilevante importanza (ad esempio per la difficoltà dell'istruttoria o della valutazione delle prove). Il TCA può dunque decidere nella composizione di un Giudice unico ai sensi degli articoli 26 c cpv. 2 della Legge organica giudiziaria civile e penale e 2 cpv. 1 della Legge di procedura per le cause davanti al Tribunale delle assicurazioni (cfr. STF 9C_211/2010 del 18 febbraio 2011; STF 9C_792/2007 del 7 novembre 2008; STF H 180/06 e H 183/06 del 21 dicembre 2007; STFA I 707/00 del 21 luglio 2003; STFA H 335/00 del 18 febbraio 2002; STFA U 347/98 del 10 ottobre 2001, pubblicata in RDAT I-2002 pag. 190 seg.; STFA H 304/99 del 22 dicembre 2000; STFA I 623/98 del 26 ottobre 1999). Nel merito 2.2. Oggetto della lite è sapere se l'Ufficio AI ha diritto alla rendita intera intera dal 1. settembre 2015 (doc. AI 122/445-450), a causa dell'invalidità della persona assicurata. Secondo l'art. 4 cpv. 1 LAI in relazione con gli art. 7 e 8 della LPGA, con invalidità s'intende l'incapacità al guadagno presunta permanente o di rilevante durata, cagionata da un danno alla salute fisica o psichica, conseguente a infermità congenita, malattia o infortunio. Gli elementi fondamentali dell'invalidità, secondo la surriferita definizione, sono quindi un danno alla salute fisica o psichica conseguente a infermità congenita, malattia o infortunio, e la conseguente incapacità di guadagno. Occorre quindi che il danno alla salute abbia cagionato una diminuzione della capacità di guadagno, perché il caso possa |
| Scores | BERT: 86.57, BLEU: 81.39, METEOR: 65.76, ROUGE-1/2/L: 68.54 / 58.87 / 66.85 |

| Target | considerations: Das Bundesverwaltungsgericht zieht in Erwägung: 1. Nach Art. 53 Abs. 1 KVG kann gegen Beschlüsse der Kantonsregierungen nach Art. 39 KVG beim Bundesverwaltungsgericht Beschwerde geführt werden. Der angefochtene Beschluss Nr. 617/2019 des Regierungsrats des Kantons Zürich vom 26. Juni 2019 wurde gestützt auf Art. 39 KVG er- lassen. Das Bundesverwaltungsgericht ist deshalb zur Beurteilung der Be- schwerde zuständig (vgl. auch Art. 90a Abs. 2 KVG). Die Beschwerdefüh- rerin hat am vorinstanzlichen Verfahren teilgenommen, ist als Adressatin durch den angefochtenen Regierungsratsbeschluss besonders berührt und hat insoweit an dessen Aufhebung beziehungsweise Abänderung ein schutzwürdiges Interesse (Art. 48 Abs. 1 VwVG). Sie ist daher zur Be- schwerde legitimiert. Auf die frist- und formgerecht erhobene Beschwerde ist, nachdem auch der Kostenvorschuss rechtzeitig geleistet wurde, einzu- treten (Art. 50 Abs. 1, Art. 52 Abs. 1 und Art. 63 Abs. 4 VwVG). Das Haupt- rechtsbegehren der Beschwerdeführerin richtet sich auf Aufhebung der Zif- fer I des Dispositivs des angefochtenen Beschlusses. Soweit sie dabei Be- zug auf «C. Neue Regelung der Verlegungen ab Geburtshaus» nimmt, ist das im Lichte der Begründung der Beschwerde nicht so zu verstehen, dass sie auch die Aufhebung eines Teils der Begründung des angefochtenen Beschlusses beantragt. Der Antrag der Vorinstanz, auf die Beschwerde sei teilweise nicht einzutreten, weil sie sich auch gegen die Begründung des angefochtenen RRB richte, ist daher unbegründet. 2. 2.1 Mit Beschwerde gegen einen Spitallistenentscheid im Sinne von Art. 39 Abs. 1 KVG kann die Verletzung von Bundesrecht unter Einschluss des Missbrauchs oder der Überschreitung des Ermessens und die unrichtige oder unvollständige Feststellung des rechtserheblichen Sachverhalts ge- rügt werden. Nicht zulässig ist hingegen die Rüge der |
|---|---|
| Predicted | considerations: Das Bundesverwaltungsgericht zieht in Erwägung: 1. 1.1 Gemäss Art. 31 des Verwaltungsgerichtsgesetzes vom 17. Juni 2005 (VGG, SR 173.32) beurteilt das Bundesverwaltungsgericht Beschwerden gegen Verfügungen nach Art. 5 des Verwaltungsverfahrensgesetzes vom 20. Dezember 1968 (VwVG, SR 172.021), sofern keine Ausnahme nach Art. 32 VGG vorliegt. Als Vorinstanzen gelten die in Art. 33 und 34 VGG genannten Behörden. Dazu gehören u.a. Verfügungen im Sinne von Art. 5 VwVG. Das Bundesverwaltungsgericht ist somit zuständig für die Beurteilung der vorliegenden Beschwerde und entscheidet auf dem Gebiet des Gesundheits- schutzes endgültig, ausser bei Eintreten auf einen Entscheid, der eine Verfügung im Sinne von Art. 5 VwVG betrifft. Eine Ausnahme nach Art. 32 VGG liegt nicht vor. 1.2 Das Bundesverwaltungsgericht ist zur Beurteilung der vorliegenden Beschwerde zuständig. 1.3. Gegen Entscheide des Regierungsrats ist ein Beschwerdeverfahren zulässig, wenn dieses eine Rechtsmittelinstanz ist und grundsätzlich der Rechts- schutzinteresse der Beschwerde- berechtigten gegeben ist (vgl. art. 32 VGG; BGE 138 II 274 E. 5.1 S. 280). 1.4 Das Bundesverwaltungsgericht wendet im Beschwerde- verfahren das Bundesrecht von Amtes wegen an (Art. 106 Abs. 1 des Bundesgesetzes vom 20. Dezember 1968 über das Verwaltungsverfahren [VwVG, SR 172.021]). Es wird gemäss Art. 62 Abs. 4 VwVG zur Beurteilung von Beschwerden in Zivilsachen gemäss Art. 82 Abs. 1 und Art. 86 Abs. 1 Bst. d bis Bst. e VwVG grundsätzlich nur geprüft, wenn diese vor Bundesgericht nicht mehr vorgetragen wurden und keine Verletzung des Op- fers gegeben wurde. Das Bundesverwal- tungsgericht wendet im Beschwerdeverfahren das Bundesrecht von Amtes wegen an. Es ist gemäss Art. 62 Abs. 4 an die Begründung der Begehren nicht gebunden und kann die Beschwerde auch aus anderen als den geltend gemachten Gründen gutheissen |
| Scores | BERT: 75.72, BLEU: 67.17, METEOR: 31.26, ROUGE-1/2/L: 26.43 / 8.84 / 25.23 |

| Target | considerations: Considérant en droit: 1. Selon l'art. 5 al. 1 des dispositions du RPAC relatives à la Commission de recours individuel, la décision rendue par cette autorité peut faire l'objet d'un recours de droit administratif auprès du Tribunal cantonal dans les trente jours suivant la communication de la décision motivée, conformément à la loi cantonale du 28 octobre 2008 sur la procédure administrative (LPA-VD; BLV 173.36). D'après l'art. 92 al. 1 LPA-VD, le Tribunal cantonal connaît des recours contre les décisions et décisions sur recours rendues par les autorités administratives, lorsque la loi ne prévoit aucune autre autorité pour en connaître. En l'espèce, le recours a été déposé en temps utile (art. 5 al. 1 RPAC) et il satisfait aux autres conditions formelles de recevabilité (art. 75 al. 1 let. a, 79 al. 1 et 99 LPA-VD, applicables par renvoi de l'art. 5 al. 1 RPAC), si bien qu'il convient d'entrer en matière. 2. a) La classification salariale du poste qu'occupe le recourant est litigieuse en l'occurrence. b) L'organisation de l'administration fait partie des tâches propres des autorités communales (art. 2 de la loi vaudoise du 28 février 1956 sur les communes [LC; BLV 175.11]). Selon cette loi, il incombe au Conseil général ou communal de définir le statut des collaborateurs communaux et la base de leur rémunération (art. 4 al. 1 ch. 9 LC), la municipalité ayant la compétence de nommer les collaborateurs et employés de la commune, de fixer leur traitement et d'exercer le pouvoir disciplinaire (art. 42 al. 1 ch. 3 LC). Conformément à l'art. 33 al. 1 RPAC, le traitement du fonctionnaire comprend le traitement de base (let. a), les allocations |
|---|---|
| Predicted | considerations: Considérant en droit: 1. 1.1 Le personnel doit maintenir, par contrat de droit privé, une commune, un partenaire public, pour assurer la bonne exécution de leurs missions spéciales, ainsi qu'un minimum d'administration (art. 10 al. 1 du RPAC). L'art. 10 al. 2 RPAC prévoit que les fonctionnaires dans toute la ville de Lausanne sont soumis à la qualification communale et leur remplacement a lieu par des fonctions communales cantonales, comme applicable en la matière. Elle constitue en revanche une décision sujette à recours auprès de la juridiction compétente (art. 20 al. 2 et art. 30 al. 1 de la loi sur le personnel des communes du 15 novembre 1976 (LPAC). 1.2 Selon la jurisprudence relative aux dispositions transitoires applicables au moment de la notification de la décision pour connaître du jugement du service, la décision de rémunération fait apparaître un changement de système de rémunération selon le RPAC. Cette disposition, qui comprend ainsi des critères objectifs et suffisants, permet au fonctionnaire de connaître de tous les aspects de la classification (Tappy, CPC, n. 1 ad art. 10 LPAC), ainsi que par le service de réduire son salaire. En l'espèce, l'autorité intimée a constaté que l'établissement des faits importants du dossier et de la satisfaction exigée sur le plan d'administration devait être complété dans la mesure où cela n'était pas possible, ce qui constituait effectivement un changement de système (Tappy, op. cit., n. 13 ad art. 10 LPAC). Il a constaté que le Service du personnel avait fixé le niveau de fonction attribué au fonctionnaire pour son titre de travail. Il a en outre estimé que cette classification était une décision de mise en œuvre déléguée à celui qui a entrepris une modification légale ou réglementaire du service. Parcon |
| Scores | BERT: 75.74, BLEU: 66.19, METEOR: 31.69, ROUGE-1/2/L: 28.32 / 4.21 / 26.01 |

| Target | considerations: Diritto: 1. Il ricorso in materia di diritto pubblico può essere presentato per violazione del diritto, così come stabilito dagli art. 95 e 96 LTF. Il Tribunale federale fonda la sua sentenza sui fatti accertati dall'autorità inferiore (art. 105 cpv. 1 LTF). Può scostarsi da questo accertamento solo qualora esso sia avvenuto in modo manifestamente inesatto, ovvero arbitrario (DTF 134 V 53 consid. 4.3 pag. 62), oppure in violazione del diritto ai sensi dell'art. 95 LTF (art. 105 cpv. 2 LTF). Salvo i casi in cui tale inesattezza sia lampante, la parte ricorrente che intende contestare i fatti accertati dall'autorità inferiore deve spiegare, in maniera circostanziata, per quale motivo ritiene che le condizioni di una delle eccezioni previste dall'art. 105 cpv. 2 LTF sarebbero realizzate (cfr. DTF 133 II 249 consid. 1.4.3 pag. 254 con riferimento). 2. Oggetto del contendere è sapere se sia lesivo del diritto federale il giudizio cantonale, che conferma la decisione su opposizione, la quale a sua volta nega all'assicurato le indennità per insolvenza per non avere rivendicato tempestivamente le proprie pretese salariali. 3. 3.1. Il Tribunale cantonale delle assicurazioni, esposto lo svolgimento del processo, le norme e la prassi ritenute applicabili, ha accertato che il ricorrente è stato legato da un contratto di lavoro con la ditta B._ SA, dichiarata fallita il 15 maggio 2018, dal 15 aprile 2017 al 18 settembre 2017 come informatico e con un salario di fr. 3'360.- mensili. L'ultimo giorno di lavoro effettuato è stato il 1° settembre 2017. Il salario è stato saldato dal 15 al 30 aprile 2017. Per contro, non è stato pagato per i mesi di maggio, giugno, luglio e agosto, nonché per il 1° settembre 2017. Il |
|---|---|
| Predicted | considerations: Diritto: 1. 1.1. La procedura ricorsuale in materia di diritto pubblico è disciplinata dal diritto federale (art. 82 lett. a LTF) e dalla legge federale, in vigore dal 1° gennaio 2003, sull'assicurazione contro la disoccupazione: si tratta infatti di una causa di diritto pubblico contro una decisione finale adottata dalla Corte cantonale delle assicurazioni, entrata in vigore il 1° gennaio 2003. Il ricorso presentato dall'assoggettato al diritto di ricorrere (art. 89 cpv. 1 LTF) è di massima ammissibile dinanzi al Tribunale federale (art. 89 cpv. 1 LTF). 1.2. Il ricorso in materia di diritto pubblico può essere presentato per violazione del diritto, così come stabilito dagli art. 95 e 96 LTF, oltre che per violazione del diritto ai sensi dell'art. 95 LTF. Il Tribunale federale applica il diritto federale (art. 106 cpv. 1 LTF), senza essere vincolato né dai motivi addotti nel ricorso (art. 106 cpv. 2 LTF). Per contro, nel ricorso in materia di diritto pubblico il Tribunale federale esamina d'ufficio e con piena cognizione l'ammissibilità dei gravami che gli vengono sottoposti (DTF 133 III 439 consid. 1.3). 1.3. La critica del giudizio impugnato esplica degli effetti (art. 105 cpv. 1 LTF). Non è ammissibile che i ricorsi in materia di diritto pubblico possano essere decisi in base al diritto federale, ai sensi dell'art. 95 LTF, senza istruttoria (art. 97 cpv. 1 LTF, Art. 105 cpv. 2 LTF). 1.3. Il Tribunale federale esamina d'ufficio e con piena cognizione l'ammissibilità dei ricorsi che gli vengono sottoposti (DTF 133 III 439 consid. 1.3). 1.4. Con il ricorso in materia di pubblico contro la decisione di primo grado, il Tribunale cantonale ha emesso |
| Scores | BERT: 75.79, BLEU: 66.28, METEOR: 30.29, ROUGE-1/2/L: 37.74 / 20.43 / 36.48 |

| | |
|---|---|
| Target | considerations: Das Versicherungsgericht zieht in Erwägung: 1. Streitig und zu prüfen ist der Rentenanspruch der Beschwerdeführerin. - 3 - 2. Am 1. Januar 2022 sind die Änderungen betreffend Weiterentwicklung der IV (WEIV) in Kraft getreten. Weder dem IVG noch der IVV sind besondere Übergangsbestimmungen betreffend die Anwendbarkeit dieser Änderun- gen im Hinblick auf nach dem 1. Januar 2022 beurteilte mögliche Ansprü- che des Zeitraums bis zum 31. Dezember 2021 zu entnehmen. Es sind daher nach den allgemeinen übergangsrechtlichen Grundsätzen jene Be- stimmungen anzuwenden, die bei der Erfüllung des rechtlich zu ordnenden oder zu Rechtsfolgen führenden Tatbestands Geltung haben beziehungs- weise hatten (vgl. Urteil des Bundesgerichts 8C_136/2021 vom 7. April 2022 E. 3.2.1 mit Hinweis unter anderem auf BGE 144 V 210 E. 4.3.1 S. 213). Da vorliegend Leistungen mit Anspruchsbeginn vor dem 1. Januar 2022 streitig sind, ist für deren Beurteilung die bis zum 31. Dezember 2021 geltende Rechtslage massgebend. 3. 3.1. Die Beschwerdegegnerin stützte sich in der Verfügung vom 14. Juni 2022 (Vernehmlassungsbeilage [VB] 162) in medizinischer Hinsicht im Wesent- lichen auf das bidisziplinäre Gutachten der SMAB vom 3. August 2021 (VB 139; Fachdiszipinen: Neurologie und Psychiatrie). 3.2. Die SMAB-Gutachterinnen stellten in der interdisziplinären Gesamtbeurtei- lung die folgenden Diagnosen mit Auswirkungen auf die Arbeitsfähigkeit (VB 139.1 S. 6): "1. Neuropathische Schmerzen an der Innenseite des Mittelfingers links und der Handinnenfläche links (Schnittverletzung am Mittelfinger links mit initialer Nervenverletzung 19.11.2014, epineurale und weitere mehrfache operative Eingriffe 2014-2018), 2. Chronische Schmerzstörung mit somatischen und psychischen (ICD-10: F45.41)". Ohne Auswirkung auf die Arbeitsfähigkeit sei eine rezidivierende depres- sive Störung, gegenwärtig remittiert. Die Gutachterinnen hielten fest, ins- gesamt bestehe aufgrund der psychiatrischen und neurologi |
| Predicted | considerations: Das Gericht zieht in Erwägung: 1. Gemäss Art. 87 Abs. 2 des Bundesgesetzes über die Unfallversicherung (UVG) besteht Anspruch auf eine ganze Rente, wenn die versicherte Person mindestens zu 66 2/3 Prozent, auf eine halbe Rente, wenn sie mindestens zu 50 Prozent oder auf eine Viertelsrente, wenn sie mindestens zu 40 Prozent invalid ist. In Härtefällen besteht gemäss Art. 87 Abs. 2 UVG auch ein Anspruch auf eine halbe Rente, wenn sie mindestens zu 50 Prozent, wenn sie mindestens zu 40 Prozent invalid ist (Art. 8 Abs. 1 UVG). Anspruch auf eine Rente haben gemäss Art. 28 Abs. 2 des Bundes- gesetzes über die Invalidenversicherung (IVG) Versicherte, die im Sinne von Art. 16 Abs. 1 IVG in Verbindung mit Art. 28 bis Abs. 1 IVG in Verbindung mit Abs. 1 und 2 IVG auch dann Anspruch auf eine Viertelsrente, wenn sie mindestens zu 40 Prozent invalid sind (Art. 28 Abs. 2 IVG i.V.m. Art. 16 ATSG). 2. 2.1 Anspruch auf eine Rente haben gemäss Art. 28 Abs. 1 IVG Versicherte, die: a. ihre Erwerbsfähigkeit oder die Fähigkeit, sich im Aufgabenbereich zu betä- tigen, nicht durch zumutbare Eingliederungsmassnahmen wieder herstellen, erhalten oder verbessern können; b. während eines Jahres ohne wesentlichen Unterbruch durchschnittlich min- destens 40 Prozent arbeitsunfähig (Art. 6 ATSG) gewesen sind; und c. nach Ablauf dieses Jahres zu mindestens 40 Prozent invalid (Art. 8 ATSG) sind. 2.2 Die Beschwerdeführerin bringt vor, die ärztlich eingeholten ärztlichen Berichte seien als diagnostisch zu qualifizieren. Das trifft vorliegend nicht zu. Ihr Gesundheitszustand sei gemäss Abklärungen vom RAD mit einer Invalidität von mindestens 40 Prozent zu vereinbaren. Die Leistungsfähigkeit sei in Art. 16 ATSG eingetreten. 2.2 Die Leistungsfähigkeit sei in Art. 16 ATSG i.V.m. Art. 28 Abs. 1 IVG |
| Scores | BERT: 62.42, BLEU: 51.67, METEOR: 19.69, ROUGE-1/2/L: 17.58 / 2.56 / 17.58 |

| Target | considerations: le conseil de B_ a rappelé que celle-ci contestait que son frère aurait été en droit de loger gratuitement au 1_ dès le mois d'avril 2015. Afin d'éviter la notification d'une poursuite interruptive de la prescription, A_ était invité à signer une déclaration de renonciation à la prescription concernant les loyers qui, selon B_, étaient dus à la succession de leurs défunts parents. Etait annexée à ce courrier une déclaration de renonciation à invoquer la prescription libellée en ces termes : " Par la présente, A_ déclare renoncer à invoquer toute exception de prescription vis-à-vis des successions de [feu D_ et feu C_], eu égard à toutes prétentions éventuelles de loyers et charges alors qu'il logeait chemin 1_ à _ Fribourg. Cette déclaration de renonciation est valable jusqu'au 31 décembre 2020. La présente déclaration ne vaut en aucun cas reconnaissance de dette ou de responsabilité quelconque. La présente déclaration n'a d'effet que si la prescription n'est pas déjà acquise à ce jour ". A_ n'a pas donné suite à ce courrier. f. Par réquisition de poursuite du 7 avril 2020, reçue le lendemain par l'Office cantonal des poursuites de Genève (ci-après : l'Office), Me H_, agissant en qualité de représentant du créancier, à savoir " Pour l'hoirie de feu C_, décédée le _ 2019, Madame B_, Rte 2_, _ Fribourg ", a requis la poursuite de A_, domicilié avenue 3_, _ Genève, pour la somme de 112'500 fr. avec intérêts à 5% dès le 1 er mai 2017, réclamée à titre de " Loyers et charges dus pour le logement du 1_ à _ Fribourg d'avril 2015 à mai 2019 (50 x |
|---|---|
| Predicted | considerations: EN DROIT 1. 1.1 Le litige porte sur la désignation d'un représentant de la communauté héréditaire et la désignation d'un administrateur de la régie G_ SA (art. 602 al. 3 CC). Le motif de la désignation du représentant est incontestable. Selon l'art. 602 al. 5 CC, la désignation d'un représentant de la communauté héréditaire est donc une décision de dernière instance cantonale au sens des art. 80 al. 1 et 82 al. 1 CPC, qui doit être notifiée à l'unique partie plaignante qui a pris part à la procédure (ATF 110 IV 92 consid. 1a; arrêt du Tribunal fédéral X. du 27 juillet 2004 ; ATF 99 III 58 consid. 1; ATF 118 Ia 397 consid. 1b; Tappy, CPC-VD, n. 3 ad art. 602 CC ; Replin, Le représentant de la communauté héréditaire, 5 ème éd., Lausanne 2013, p. 569; ATF 118 IV 286 consid. 2a; TF 6B_211/2007 du 29 janvier 2008, consid. 5.3; ATF 117 IV 29 consid. 3b; TF 8B_44/2007 du 15 août 2007, consid. 3.2; Tappy, Procédure civile, tome II, ad art. 602; Piquerez, in : Kuhn/Jeanneret [éd.], Basler Kommentar, n. 6 ad art. 602; TF 7B_51/2007 du 1 er janvier 2008, consid. 3.2; ATF 130 III 136 consid. 1.2.1; TF 9C_438/2007 du 30 septembre 2007, consid. 3.1; ATF 134 III 102 consid. 3.1; TF 6B_71/2007 du 24 août 2007, consid. 5b; TF 9C_792/2007 du 11 août 2007, consid. 4.2). 1.2 La désignation d'un représentant et de l'administration régulière de l'ensemble de la succession ont été rejetées. 2. 2.1. L'art. 602 al. 3 CC ouvre un recours au Tribunal cantonal, sans être lié par l |
| Scores | BERT: 63.90, BLEU: 46.65, METEOR: 19.12, ROUGE-1/2/L: 11.52 / 1.64 / 10.91 |

| | |
|---|---|
| Target | considerations: A.c. Con progetto di decisione dell'8 gennaio 2014 l'UAI ha concluso, in via di riconsiderazione, per la soppressione della mezza rendita d'invalidità con effetto retroattivo al 30 aprile 2011. In sede di audizione, l'UAI ha esperito nuovi accertamenti, in particolare la perizia pluridisciplinare dell'8 aprile 2016. Con decisione del 14 luglio 2016 l'UAI ha riconosciuto che al momento dell'assegnazione della mezza rendita d'invalidità è incorso in un errore manifesto avendo applicato il "Prozentvergleich" in luogo del metodo ordinario di confronto dei redditi, da cui risulta un grado d'invalidità del 26 %. Ritenuta l'importanza della sua correzione, l'UAI, riconsiderando le decisioni 14 dicembre 2006 e 14 maggio 2007 nonché la comunicazione del 20 marzo 2009, ha predisposto la soppressione del diritto alla mezza rendita invalidità con effetto retroattivo al 30 aprile 2011. B. A._ si è aggravato il 13 settembre 2016 al Tribunale delle assicurazioni del Cantone Ticino, che con giudizio del 17 maggio 2017 ha respinto il gravame. C. Il 19 giugno 2017 (timbro postale) A._ inoltra un ricorso in materia di diritto pubblico al Tribunale federale, cui chiede di ammettere il ricorso e riformare il giudizio cantonale nel senso di confermare il diritto ad almeno una mezza rendita d'invalidità e ordinare all'UAI di versare tutte le prestazioni trattenute dal momento della revisione ad oggi. L'UAI e l'Ufficio federale delle assicurazioni sociali rinunciano a determinarsi. Diritto: 1. Il ricorso in materia di diritto pubblico può essere presentato per violazione del diritto così come stabilito dagli art. 95 e 96 LTF. Il Tribunale federale esamina d'ufficio l'applicazione de l diritto federale (art. 106 cpv. 1 L |
| Predicted | considerations: Diritto: 1. Il Tribunale delle assicurazioni esamina d'ufficio e con piena cognizione l'ammissibilità dei gravami che gli vengono sottoposti (DTF 134 V 131 consid. 1.1 pag. 131; 133 V 477 consid. 1.1 pag. 481). In particolare, i gravami inoltrati già in tempo utile contro le decisioni di prima istanza rese dall'UAI in materia di assicurazioni sociali possono essere decisi sulla scorta degli atti, senza istruttoria (DTF 133 V 477 consid. 1.2 pag. 481). Il giudizio può essere reso sulla base degli atti, senza istruttoria (DTF 133 V 477 consid. 1.2 pag. 481; 133 V 477 consid. 1.2 pag. 481). 2. Il Tribunale delle assicurazioni esamina d'ufficio e con piena cognizione l'ammissibilità dei gravami che gli vengono sottoposti (DTF 133 V 477 consid. 1.2 pag. 481). 3. 3.1. Giusta l'art. 4 cpv. 1 LAI in relazione con gli art. 7 e 8 della LPGA, con invalidità s'intende l'incapacità al guadagno presunta permanente o di rilevante durata, cagionata da un danno alla salute fisica o psichica, conseguente a infermità congenita, malattia o infortunio. Gli elementi fondamentali dell'invalidità, secondo la surriferita definizione, sono quindi un danno alla salute fisica o psichica conseguente a infermità congenita, malattia o infortunio, e la conseguente incapacità di guadagno. Occorre quindi che il danno alla salute abbia cagionato una diminuzione della capacità di guadagno, perché il caso possa essere sottoposto all'assicurazione per l'invalidità (DTF 134 V 329 consid. 3.2.3 pag. 331; 133 V 477 consid. 3.3.1 pag. 481; 129 V 4 consid. 1.2 pag. 5 con rinvii). 3.2. Giusta |
| Scores | BERT: 67.86, BLEU: 55.76, METEOR: 25.41, ROUGE-1/2/L: 24.32 / 5.59 / 23.65 |

Table 39: Three examples of generated summaries by mT5Base with input length 4096, showcasing high, average, and low scored outputs in LDS Task

| Target | regeste: Regeste Teilklage und negative Feststellungswiderklage ( Art. 86 und 224 ZPO ). Die Ausnahme vom Erfordernis der gleichen Verfahrensart gemäss Art. 224 Abs. 1 ZPO für negative Feststellungswiderklagen ist nicht auf den Fall beschränkt, dass es sich bei der Hauptklage um eine sogenannte echte Teilklage handelt, sondern gilt allgemein dann, wenn die Teilklage eine Ungewissheit zur Folge hat, die es rechtfertigt, die Feststellung des Nichtbestands einer Forderung oder eines Rechtsverhältnisses zu verlangen (E. 2). |
|---|---|
| Predicted | regeste: Regeste Voraussetzungen, unter denen eine Widerklage erhoben werden darf, wenn eine echte Teilklage beurteilt wird (E. 2.1). Zur Anwendung von Art. 224 Abs. 1 ZPO ist die Ausnahme vom Erfordernis der gleichen Verfahrensart gemäss Art. 224 Abs. 1 ZPO nicht auf diesen Fall beschränkt, sondern gilt allgemein dann, wenn die Teilklage eine Ungewissheit zur Folge hat, die es rechtfertigt, im Sinne von Art. 88 ZPO die Feststellung des Nichtbestands einer Forderung oder eines Rechtsverhältnisses zu verlangen (E. 2.2). |
| Scores | BERT: 88.03, BLEU: 30.45, METEOR: 43.78, ROUGE-1/2/L: 72.44 / 54.05 / 70.87 |
| Target | regeste: Regeste Uferschutzplanung Wohlensee; Gesamtinteressenabwägung. Die Behörden nehmen bei der Genehmigung einer Sondernutzungsplanung eine umfassende Interessenabwägung vor (vgl. Art. 3 RPV ; E. 3.2). Würdigung des Gutachtens der Vogelwarte Sempach und der Stellungnahme des BAFU als Fachbehörde (E. 5.5, 6.3 und 6.4). Im zu beurteilenden Fall wird das öffentliche Interesse an einer ufernahen Wegführung (vgl. Art. 3 Abs. 2 lit. c RPG ) relativiert, weil ein durchgehender Sichtschutz zum Schutz der im Gebiet vorkommenden Vogelarten erforderlich wäre, was Spaziergängern das Erleben der Uferlandschaft nur sehr beschränkt ermöglichen würde (E. 6.5 sowie 3.3). Das Gebiet des Wohlensees stellt eines von insgesamt 25 Wasser- und Zugvogelreservaten von nationaler Bedeutung dar (vgl. Art. 11 Abs. 2 JSG, Art. 2 Abs. 1 i.V.m. Anhang 1 WZVV). |
| Predicted | regeste: Regeste Art. 3 Abs. 2 lit. c RPG ; schweizerische Vogelwarte Sempach; Umweltschutz. Gutachten im Bereich des Umweltrechts, insbesondere von dem UVG/USG ergangenen Streitwert (E. 3.1). Ein öffentlicher Weg, der über Privatgrundstücke führt und mit dem Enteignungsrecht zugunsten des Gemeinwesens verbunden ist, stellt eine öffentlich-rechtliche Eigentumsbeschränkung dar (E. 3.3). Interessenabwägungen sind grundsätzlich frei (E. 3.4 und 3.5). Umfang des geplanten Uferweges; ein ausweisenes öffentliches Interesse an einem direkt dem Ufer entlangführenden Weg sowie an einer zumindest ufernahen Wegführung (E. 3.5). |
| Scores | BERT: 73.91, BLEU: 47.21, METEOR: 30.91, ROUGE-1/2/L: 32.75 / 10.63 / 32.75 |
| Target | regeste: Regeste a Schwere Bedrängnis, grosse seelische Belastung und heftige Gemütsbewegung; Art. 48 lit. a Ziff. 2 und Art. 48 lit. c StGB. Abgrenzung dieser Strafmilderungsgründe untereinander (E. 2.1-2.3). Regeste b Mord, heftige Gemütsbewegung und grosse seelische Belastung; Art. 48 lit. c und Art. 112 StGB. Die für den Mord typische Skrupellosigkeit schliesst die Bejahung der Strafmilderungsgründe der heftigen Gemütsbewegung und der grossen seelischen Belastung aus (E. 2.5). |
| Predicted | regeste: Regeste Art. 112 und 49 Abs. 2 StGB ; tiefe Gewalt, Schwere Gewalt, Mord. Die Beeinträchtigung oder der schwere Gewalt ist einen psychischen Klumpen im Sinne des Art. 48 StGB, wenn beide Personen einen Beziehungskonflikt bilden und sich ein bestimmtes Verhältnis zwischen einseitigen Umständen des Tatvergehens und der Begründung der Tatbegehren haben (E. 2 und 3). |
| Scores | BERT: 69.3, BLEU: 12.14, METEOR: 14.72, ROUGE-1/2/L: 30.43 / 10.26 / 30.43 |

