# OpenReview forum: "SCALE: Scaling up the Complexity for Advanced Language Model Evaluation"
_ICLR.cc/2024/Conference — ICLR 2024 Conference Withdrawn Submission_

### Official Review · Reviewer_cCJK · 2023-10-13

**Soundness:** 3 good
**Presentation:** 2 fair
**Contribution:** 2 fair
**Rating:** 3
**Confidence:** 3

**Summary:**

This work introduces a benchmark called SCALE, which consists of seven legal tasks sourced from the Swiss legal system.  On this benchmark, the authors evaluate a wide range of LLMs, including black-box models, open-source models, and the one tuned via in-domain data.

**Strengths:**

1. the dataset is unique and definitely interesting to the LLM community and people in related domains.
2. it includes multiple distinct and challenging tasks
3. The experiments are extensive and cover many recent models

**Weaknesses:**

1. The title is misleading, as a dataset for a specialized domain, the title should state its scope clearly. I think it is of great importance for a serious research paper to have appropriate title.
2. Related to the above scope issue, in the abstract (as well as the main body of the paper), the authors state that we need more challenging tasks for LLM, then why legal tasks specifically? To accomplish the goal of proposing more challenging tasks, why not use data from domains like finance, medical, etc?
3. This work includes extensive experiments, but it could be better to include more analysis, discussion and takeaways

**Questions:**

Citation format in the first sentence of section 5 is incorrect.

---

> ### Author Response · Authors · 2023-11-20
>
> We extend our sincere appreciation for the time and effort you dedicated to reviewing our paper. Your insightful comments have been immensely valuable, and we are particularly grateful for your acknowledgment of the strengths highlighted in your review.
>
> Unique and Interesting Dataset: Your recognition of the uniqueness and value of our dataset to the Large Language Model (LLM) community and related domains resonates deeply with our intentions behind its creation. We aimed to provide a resource that stands out and contributes meaningfully to the research landscape.
>
> Distinct and Challenging Tasks: Your observation regarding the inclusion of multiple distinct and challenging tasks affirms our commitment to creating a benchmark that pushes the boundaries of current LLM capabilities. Diversifying tasks was a deliberate choice to ensure a comprehensive evaluation.
>
> Extensive Experiments with Recent Models: We appreciate your positive assessment of the extensive experiments covering a spectrum of recent models. Your acknowledgment reinforces our dedication to a thorough evaluation, aiming to contribute relevant insights to the LLM research community.
>
> We appreciate the constructive feedback provided by the reviewer and acknowledge the importance of refining our work based on insightful observations. In response to the raised points of critique, we are committed to enhancing the clarity, focus, and depth of our paper.
>
> **Weakness:** “The title is misleading, as a dataset for a specialized domain, the title should state its scope clearly. I think it is of great importance for a serious research paper to have an appropriate title.”
> **Response:** While we appreciate the feedback, we acknowledge the importance of a clear title. We are considering alternative titles that more precisely convey the specialized nature of the dataset.
>
> **Weakness:** “Related to the above scope issue, in the abstract (as well as the main body of the paper), the authors state that we need more challenging tasks for LLM, then why legal tasks specifically? To accomplish the goal of proposing more challenging tasks, why not use data from domains like finance, medical, etc?”
> Response: We recognize the concern and will refine our focus to emphasize a legal benchmark. This adjustment aims to align the paper more closely with the specific challenges posed by legal tasks for LLMs.
>
> **Weakness:** “This work includes extensive experiments, but it could be better to include more analysis, discussion and takeaways.”
> Response: We share the reviewer's perspective on the importance of in-depth analysis and discussion. While space constraints limit the depth in the main body, we are considering relocating supplementary material to the appendix to provide more comprehensive insights.
>
> **Question/Weakness:** “Citation format in the first sentence of section 5 is incorrect.”
> Response: Thank you for pointing the mistake out. We will remedy that.

---

### Official Review · Reviewer_pqFy · 2023-10-31

**Soundness:** 2 fair
**Presentation:** 2 fair
**Contribution:** 2 fair
**Rating:** 3
**Confidence:** 4

**Summary:**

This paper proposes to introduce a benchmark of model performance across classification, text generation & information retrieval tasks on legal datasets. The datasets consist of long documents that are multilingual in nature. The models benchmarked in the study include LLMs and some (smaller)  models such as mT5 and XLM-R that were fine-tuned on domain-specific data. The results demonstrate the variability in performance that LLMs yield for different tasks.

**Strengths:**

The paper is clearly written and highlights the variability in performance of LLMs across tasks, especially when applied to domains that they may not have been exposed to during training. It is interesting to see, for example, that smaller fine-tuned models (XLM-R or RoBERTa based models) outperform off-the-shelf LLMs for text classification tasks across all the legal datasets used in this paper.
Additionally, the experiments were detailed and clearly explained.
Moreover, the inclusion of the observant ethics statement is highly commendable.

**Weaknesses:**

Although construction of a benchmark for better evaluation of LLMs along specific dimensions is of importance, I was unable to determine what the novelty of this work is, w.r.t. other already existing NLP benchmarks. Benchmarks for LLM evaluation already exist both for the legal domain [e.g. LEXTREME (https://arxiv.org/pdf/2301.13126.pdf), LexGLUE (https://aclanthology.org/2022.acl-long.297.pdf), LegalBench (https://arxiv.org/pdf/2308.11462v1.pdf) etc.] and otherwise [BigBench (https://arxiv.org/pdf/2206.04615.pdf), HELM (https://arxiv.org/pdf/2211.09110.pdf), etc]. The current work expands on these benchmarks in terms of including legal datasets specific to the Swiss legal system, which do not meet the standards of an ICLR paper, in my opinion.
Further, if the focus is on evaluating LLMs, it would be important to include more LLMs in the zero-shot & one-shot settings. These could include Falcon, Flan-T5 XXL, Alpaca, Vicuna etc. This would allow for a wider coverage of the behavior of LLMs on the tasks at hand.

**Questions:**

It would be great if you could highlight the key, novel contributions of the paper in comparison to the already existing benchmarks for LLMs.

**Details Of Ethics Concerns:**

The ethics section is very well-written in the paper already.

---

> ### Author Response · Authors · 2023-11-20
>
> We appreciate your thorough review and valuable feedback on our paper. Your comments highlighted our work's strengths, contributing significantly to the field. Key aspects you acknowledged include:
>
> Clarity of Presentation: Your acknowledgement of our paper's clarity is gratifying. Our aim for precise communication helps readers comprehend our methodology and findings.
>
> Variability in LLM Performance: You noted the variability in Large Language Models (LLMs) across tasks, especially in new domains, validating the significance of our findings.
>
> Effectiveness of Smaller Fine-Tuned Models: Your interest in our experiments with models like XLM-R and RoBERTa highlights the effectiveness of tailored approaches.
>
> Detailed and Clear Experiments: We appreciate your positive feedback on our experiment's clarity and detail, underscoring our commitment to methodological transparency.
>
> Ethics Statement: We are thankful for your recognition of our ethics statement, reflecting our dedication to ethical standards in research
>
> Your feedback reinforces our belief in our contributions. We now address your constructive suggestions to further improve our paper.
>
>
> **Weakness:** “Although the construction of a benchmark for better evaluation of LLMs along specific dimensions is of importance, I was unable to determine what the novelty of this work is, w.r.t. other already existing NLP benchmarks.“
> **Response:** The distinctive aspect lies in the novel framework of modeling judicial processes from start to finish with concrete datasets. Additionally, we present novel and extremely challenging cross-lingual datasets not present in existing legal benchmarks. However, we acknowledge the importance of clarity regarding the novelty of our work and will make it clearer.
>
> **Weakness:** „Benchmarks for LLM evaluation already exist both for the legal domain [e.g. LEXTREME, LexGLUE, LegalBench etc.] and otherwise [BigBench, HELM, etc].“
> **Response:** We appreciate the reviewer's diligence in identifying existing benchmarks. While we acknowledge their existence, our work expands on these benchmarks by incorporating two challenging text generation tasks, an information retrieval task and by coining the novel criticality prediction tasks, providing a more diverse and challenging evaluation paradigm. None of these three task categories are part of the existing legal benchmarks to our knowledge. We will emphasize these points more clearly in the revised manuscript.
>
> **Weakness:** “The current work expands on these benchmarks in terms of including legal datasets specific to the Swiss legal system, which do not meet the standards of an ICLR paper, in my opinion.”
> **Response:** The Swiss legal system is uniquely positioned to serve as a testing ground for evaluation of the most complex challenges in legal NLP: 1) it is non-English (most models are much better in English). 2) it is crosslingual (three languages), (e.g. requiring retrieval of French legal documents from a German query, an extremely difficult task). 3) It has exceptional public availability of caselaw and legislation per capita (e.g., over 600K decisions for just over 8M people). Additionally, as exemplified by a huge wave in legal AI startups over the last few months (e.g., Harvey raising 20M, CaseText being acquired for 650M), this space is very important. However, we understand the concern about the specificity of the legal datasets and will provide a more nuanced discussion in the revised manuscript.
>
> **Weakness:** „Further, if the focus is on evaluating LLMs, it would be important to include more LLMs in the zero-shot & one-shot settings. These could include Falcon, Flan-T5 XXL, Alpaca, Vicuna, etc. This would allow for a wider coverage of the behavior of LLMs on the tasks at hand.“
> **Response:** As clarified in the introduction of Section 5, the mentioned models were not included due to their lack of multilingual capabilities. We will make this rationale more explicit in the revised manuscript.
>
> **Question:**:“It would be great if you could highlight the key, novel contributions of the paper in comparison to the already existing benchmarks for LLMs.“
> ** Response:** We appreciate the reviewer's suggestion. In our revised manuscript, we will explicitly highlight the key and novel contributions of our work in comparison to existing benchmarks for LLMs. We will delve into specific aspects that set our benchmark apart, such as the inclusion of text generation tasks and the unique challenges posed by the Swiss legal system. This clarification aims to provide a more nuanced understanding of the distinctive features of our work.

---

> ### Comment · Reviewer_pqFy · 2023-11-23
> **Acknowledgement of author response**
>
> I thank the authors for their detailed response, and for addressing the weaknesses raised. I would increase my rating by 1 as a result.

---

### Official Review · Reviewer_xb8g · 2023-11-01

**Soundness:** 2 fair
**Presentation:** 3 good
**Contribution:** 2 fair
**Rating:** 5
**Confidence:** 4

**Summary:**

This study introduces a comprehensive evaluation dataset for large language models (LLMs) focusing on the legal domain. The dataset is sourced from Swiss legal documents and comprises seven multilingual datasets covering four key dimensions: long documents, specificity to the Swiss legal domain, multilinguality, and multitasking. The authors further conduct in-domain pretraining to develop Legal-Swiss-RoBERT and Legal-Swiss-LF models specifically tailored for this domain. The proposed evaluation includes seven tasks (LAP, CP, IR, CVG, JP, CP, and LDS) forming a testbed to assess the performance of existing LLMs in the legal domain.

**Strengths:**

1. This study introduces an evaluation dataset specifically designed to assess the performance of large language models (LLMs) in the legal domain. The dataset emphasizes 4 challenging dimensions that pose difficulties for LLMs, thereby providing a comprehensive and rigorous evaluation for LLMs operating within legal fields.

2. The quality of the research is substantiated by rigorous experimental design. The authors conducted experiments to examine and analyzed the performance of existing pre-trained language models (LMs) in legal fields. Furthermore, the authors showcased the significance and worth of their collected dataset by fine-tuning LMs on it. The multilingual nature of the dataset adds an additional layer of complexity to the evaluation of language models.

**Weaknesses:**

About the experimental setup: it is advisable to expand the inclusion of additional existing large language models (LLMs) in the experiments. Given the lengthy nature of legal documents, it would be beneficial to evaluate the performance of LLMs specifically pre-trained for handling long contexts. It is recommended to conduct more comparisons with LLMs specifically designed to handle long contexts. Furthermore, it appears inequitable to compare models with input length restrictions imposed by fixed-sized tokens. It is also desirable to extend the proposed method to encompass a wider range of LLMs.

**Questions:**

1. Please address the weaknesses above.

2. In text classification tasks, the models are provided with facts and considerations explicitly written by legal professionals such as lawyers or judges. This simplification reduces the evaluation complexity of language models (LMs). What are the underlying reasons for adopting this simplified approach and reducing the complexity in LM evaluation?

3. In Table 3, the majority of models exhibit superior performance on the "-C" datasets compared to the "-F" datasets. However, BLOOM, Legal-Swiss-RoBERTa_{Large}, and Legal-Swiss-LF_{Base} demonstrate relatively poorer performance. What could be the potential reasons behind the clearly weaker performance of these models?

---

> ### Author Response · Authors · 2023-11-20
>
> We express our sincere gratitude to the esteemed reviewer for providing insightful feedback on our paper. We are pleased to acknowledge the reviewer's recognition of the strengths inherent in our work, which forms the foundation of this rebuttal. Specifically, we appreciate the acknowledgment of our study's contribution in introducing a tailored evaluation dataset crafted for assessing large language models (LLMs) in the intricate landscape of the legal domain. The emphasis on four challenging dimensions within the dataset, strategically designed to pose difficulties for LLMs, underlines our commitment to providing a comprehensive and rigorous evaluation framework.
>
> Furthermore, we welcome the reviewer's positive remarks regarding the quality of our research, highlighting the rigorous experimental design employed in examining and analyzing the performance of existing pre-trained language models in the legal domain. The reviewer aptly acknowledges the significance and worth of our collected dataset, particularly underscored by our approach of fine-tuning language models on this distinctive legal corpus. The recognition of the multilingual nature of our dataset, adding an extra layer of complexity to the evaluation, reinforces the depth and breadth of our contributions.
>
> In the subsequent sections, we address the concerns and queries raised by the reviewer, aiming to provide clarification and additional context where needed. We value the constructive dialogue with the reviewer, and we are committed to refining and strengthening our work based on the valuable insights received.
>
> **Weakness:** "About the experimental setup: it is advisable to expand the inclusion of additional existing large language models (LLMs) in the experiments."
> **Response:** As detailed in Section 5, we have already assessed most state-of-the-art multilingual models available.
>
> **Weakness:** "Given the lengthy nature of legal documents, it would be beneficial to evaluate the performance of LLMs specifically pre-trained for handling long contexts. It is recommended to conduct more comparisons with LLMs specifically designed to handle long contexts."
> **Response:** We have already included Claude in our evaluation, which can effectively handle 100K tokens. Additionally, we are considering the inclusion of GPT-4 turbo for further assessment.
>
> **Weakness:** "Furthermore, it appears inequitable to compare models with input length restrictions imposed by fixed-sized tokens. It is also desirable to extend the proposed method to encompass a wider range of LLMs."
> **Response:** Our primary focus is on dataset creation, and we have already conducted evaluations with a diverse set of models (over 20 different models in total!).
>
> **Question:** “In text classification tasks, the models are provided with facts and considerations explicitly written by legal professionals such as lawyers or judges. This simplification reduces the evaluation complexity of language models (LMs). What are the underlying reasons for adopting this simplified approach and reducing the complexity in LM evaluation?”
> **Response:** Due to the non-public availability of complaints, adopting a simplified approach with explicit information from legal professionals becomes necessary. Nevertheless, as demonstrated in our experiments in Table 3, most models still struggle greatly not exceeding 50 macro-F1 aggregated over all text classification tasks!
>
> **Question:** "In Table 3, the majority of models exhibit superior performance on the '-C' datasets compared to the '-F' datasets. However, BLOOM, Legal-Swiss-RoBERTa_{Large}, and Legal-Swiss-LF_{Base} demonstrate relatively poorer performance. What could be the potential reasons behind the clearly weaker performance of these models?"
> **Response:** Thank you for highlighting this observation. BLOOM and Legal-Swiss-RoBERTa_{Large} only perform worse in CPC-C. Legal-Swiss-LF_{Base} performs worse in CPC-C and SLAP-C. However, also other models such as PaLM-2 and GPT-3.5 perform poorer in the CPC-C task compared to CPC-F. The CPC task, where models need to predict how many times a decision will be cited in the future, is very difficult in general. It seems that some models find more correlation between the facts and the citation count and others between the considerations and the citation count. We will investigate this matter more and add more discussion around it.

---

> > ### Comment · Reviewer_xb8g · 2023-11-22
> >
> > Thank you for the response. Regrettably, the elucidations provided have not fully assuaged my reservations pertaining to question 3. I have no further questions and keep the rating unchanged.

---

### Official Review · Reviewer_x9cb · 2023-11-04

**Soundness:** 4 excellent
**Presentation:** 3 good
**Contribution:** 3 good
**Rating:** 6
**Confidence:** 4

**Summary:**

This paper introduces a new multilingual legal NLP benchmark dataset, named SCALE. The characteristics of this dataset are multilingual (German, French, Italian, Romansh, and English), long documents, and multitasking. The origin of SCALE is Swiss legal documents, and the authors arrange the raw data into several text classification and generation tasks. The authors also show the performance of large language models and it shows that the benchmark is still challenging for the NLP community.

**Strengths:**

The primary strength of this paper lies in proposing a challenging and extensible benchmark dataset. It is helpful for AI and NLP researchers who are interested in the tasks.

**Weaknesses:**

The benchmark is quite interesting and sound. But I have some curiosity which I mention in the Questions. I hope to listen to the author's responses.

**Questions:**

- For each task, what is human (legal experts and non-experts) performance, especially NLG tasks? Other benchmark tasks show the human performance for the tasks that can be the upper bound or challenge for AI models.
- How each task helps in the legal domain. For example, if you have an AI model that is good at law area prediction, how would it help?
- Presentation: it would be better to enlarge the font size of captions. Now it is hard to read.

**Details Of Ethics Concerns:**

In the Judgement documents, it might contain PII.

---

> ### Author Response · Authors · 2023-11-20
>
> We express our sincere gratitude for your thoughtful and insightful review of our paper. Your constructive feedback has been invaluable in refining our work. We particularly appreciate your recognition of the primary strength of our paper – the proposal of a challenging and extensible benchmark dataset. We are thrilled to know that you find our dataset valuable for AI and NLP researchers engaged in tasks related to the legal domain. In this rebuttal, we aim to address your valuable comments and provide further clarification on certain aspects of our work.
>
> **Question:** "For each task, what is human (legal experts and non-experts) performance, especially NLG tasks? Other benchmark tasks show the human performance for the tasks that can be the upper bound or challenge for AI models."
>
> **Response:** Non-expert performance is not considered relevant or useful in this context. The annotations in our datasets are already annotated by experts, namely highly trained clerks and judges in the Swiss judicial system.
>
> ‘’Question:** "How does each task help in the legal domain? For example, if you have an AI model that is good at law area prediction, how would it help?"
>
> **Response:** We elaborate on this aspect in the appendix on page 44, detailing how the tasks contribute to routing complaints to the correct legal departments inside a court.
>
> **Question:** "Presentation: it would be better to enlarge the font size of captions. Now it is hard to read."
>
> **Response:** We adhered to the style sheet of the venue for the presentation, but we'll review and consider adjustments for better readability.

---

> > ### Comment · Reviewer_x9cb · 2023-11-22
> > **Thank you for your response.**
> >
> > Thank you for your response. Initially, I liked this paper for its focus on a multilingual dataset, particularly within a specific domain. I find the specificity beneficial as it allows for in-depth analysis and domain-specific task exploration.
> >
> > However, upon reading other reviews, I am curious about the necessity of the multilingual aspect. As someone not well-versed in legal matters and residing in a non-multilingual country, it is challenging to envision the practical utility and challenges of a multilingual legal document dataset.
> >
> > - Are they parallel corpus such as European Parliament Proceedings Parallel Corpus? I think it is not since the proportion of languages are different (Figure 6 and 7).
> >
> > - Do Swiss courts generate multilingual documents due to the presence of four national languages in Switzerland? Consequently, could this dataset be used to train models for tasks like classifying documents written in any of the five languages?
> >
> > https://en.wikipedia.org/wiki/Languages_of_Switzerland
> >
> > - Given the unbalanced language distributions, is dealing with documents in Romansh or English, which are low-resource languages in SCALE, one of the challenges?
> >
> > While I appreciate the dataset's positive aspects - long documents, domain specificity, multilinguality, and multitasking (as the authors highlighted) - it seems that, due to various highlights, a more profound understanding of each aspect might be limited within the scope of this paper.
> >
> > Yours,

---

> > > ### Author Response · Authors · 2023-11-23
> > >
> > > Thank you for your message!
> > >
> > > Our datasets are partially parallel: The federal legislation is parallel, and some of the cantonal (regional) legislative documents are too. Regarding caselaw, the leading decisions are parallel, but all other decisions are not.
> > >
> > > The Swiss courts usually generate the document in the language that the complaint comes in. So if a complaint from Geneva comes in, the decision will be written in French. If there is a complaint from Zurich, the case will be handled in German. Yes, our ten text classification tasks are tackling exactly that problem: Cases in different languages come in, and they have to be classified, for example into a specific law area for routing to the correct court department.
> > >
> > > Yes, exactly. Dealing with cases in low-resource languages is definitely a challenge. To promote fairness, we average our scores across languages, thus giving equal weight to each language-specific subset of the data.
> > >
> > > We hope that our comments were clarifying and are happy to answer any remaining questions.